# Thunder-DDA-PASEF enables high-coverage immunopeptidomics and is boosted by MS²Rescore with MS²PIP timsTOF fragmentation prediction model

David Gomez-Zepeda [1,2,3] ✉, Danielle Arnold-Schild[1], Julian Beyrle [1,2,3], Arthur Declercq[4,5], Ralf Gabriels [4,5], Elena Kumm[1], Annica Preikschat [1], Mateusz Krzysztof Łącki[1], Aurélie Hirschler [6], Jeewan Babu Rijal [6], Christine Carapito [6], Lennart Martens [4,5], Ute Distler [1,7], Hansjörg Schild[1,7] & Stefan Tenzer [1,2,3,7] ✉

Human leukocyte antigen (HLA) class I peptide ligands (HLAIps) are key targets for developing vaccines and immunotherapies against infectious pathogens or cancer cells. Identifying HLAIps is challenging due to their high diversity, low abundance, and patient individuality. Here, we develop a highly sensitive method for identifying HLAIps using liquid chromatography-ion mobility-tandem mass spectrometry (LC-IMS-MS/MS). In addition, we train a timsTOF-specific peak intensity MS²PIP model for tryptic and non-tryptic peptides and implement it in MS²Rescore (v3) together with the CCS predictor from ionmob. The optimized method, Thunder-DDA-PASEF, semi-selectively fragments singly and multiply charged HLAIps based on their IMS and m/z. Moreover, the method employs the high sensitivity mode and extended IMS resolution with fewer MS/MS frames (300 ms TIMS ramp, 3 MS/MS frames), doubling the coverage of immunopeptidomics analyses, compared to the proteomics-tailored DDA-PASEF (100 ms TIMS ramp, 10 MS/MS frames). Additionally, rescoring boosts the HLAIps identification by 41.7% to 33%, resulting in 5738 HLAIps from as little as one million JY cell equivalents, and 14,516 HLAIps from 20 million. This enables in-depth profiling of HLAIps from diverse human cell lines and human plasma. Finally, profiling JY and Raji cells transfected to express the SARS-CoV-2 spike protein results in 16 spike HLAIps, thirteen of which have been reported to elicit immune responses in human patients.

Identifying ligands of the major histocompatibility complex (MHC) or human leukocyte antigen (HLA), also called immunopeptides, is key for developing vaccines and immunotherapies (extensively reviewed in refs. 1–3). Human HLA class-I complexes bind peptides (HLAIps) of typically 9–12 amino acids generated by a multi-step process called antigen processing, which involves multiple proteolytic events by the proteasome and aminopeptidases[4–8]. Loaded HLA complexes are then displayed on the cell surface, where CD8+ T-cells scrutinize them.

Detection of a non-self antigen, e.g., HLAIps derived from viral proteins or mutated cancer-related proteins, leads to the efficient elimination of the presenting cell by cytotoxic T lymphocytes. Thus, non-self HLAIps constitute key targets for developing peptide or mRNA vaccines in the context of personalized immunotherapies, or diagnostic tools. Various in silico tools have been developed to predict HLA-binding peptides from genomic, transcriptomic, or riboSeq data. Still, most predictors are primarily based on HLA binding affinity, thus not fully considering the antigen processing and presentation mechanisms, resulting in discrepancies between predicted and presented HLAIps[9,10]. Therefore, liquid chromatography mass spectrometry (LC-MS)-based immunopeptidomics is essential for directly identifying HLA class I presented peptides from cells, tissues, and biofluids[9,11].

However, LC-MS immunopeptidomics faces different challenges than bottom-up proteomics, where proteins are usually digested using trypsin (reviewed in refs. 3, 12). HLAIps are generated by a complex multi-step process, including various proteolytic events[13,14]. This results in peptides with restricted size and sequence patterns imprinted by the specificities of TAP transport and HLA binding. While these motifs differ between individual HLA alleles, they restrict the length and sequence space presented by a single allele. Thus, immunopeptidomics samples are more likely to contain isobaric peptides, potentially co-eluting from the LC, than enzyme-digested samples[2]. Since tryptic peptides are usually multi-charged, typical bottom-up proteomics workflows often omit the fragmentation and identification of singly-charged ions, which are more challenging to identify. In addition, singly-charged peptides are often masked by chemical noise, and their fragmentation generates many uncharged segments not detected by the MS[2]. Moreover, individual HLAIps are low abundant, and the sample preparation recovery yields are low (~0.5–3%[15]). These factors demand tailored and high-sensitivity LC-MS methods and have major implications in database searches. The unspecific cleavage of HLAIps increases the search space by up to 2 orders of magnitude compared to tryptic digests. This impairs the discrimination of false positive from true positive peptide-spectrum matches (PSMs), negatively impacting peptide identification yield and confidence[16].

Coupling ion mobility separation (IMS) to LC-MS provides an extra dimension of separation, resolving ions in the gas phase by their collisional cross section (CCS), which is defined by their size and shape. As a result, the signal-to-noise ratio increases and isobaric ions may be resolved, thus increasing the sensitivity, number, and confidence of peptide identifications. Field asymmetric waveform ion mobility spectrometry (FAIMS) has been combined with LC-MS to increase peptide coverage in immunopeptidomics experiments[17]. However, FAIMS acts as a gas-phase fractionation device, filtering ions in function of their mobility in the electric field. Since only a population of ions can be analyzed simultaneously, profiling peptides with different mobilities such as multiply and singly-charged peptides requires dividing the cycle time within an LC-MS run between different populations of ions or performing multiple injections per sample[17]. In contrast, the timsTOF Pro instruments use a dual trapped ion mobility spectrometry (TIMS) analyzer to perform a parallel accumulation-serial fragmentation (PASEF) of ions. In brief, a package of ions is trapped in the first analyzer while the previous ion package is separated across the IMS range before ions are fragmented and detected. This allows for a high duty cycle and low ion losses, resulting in higher sensitivity for data-dependent acquisition (DDA-PASEF)[18,19].

During the Covid-19 pandemic, there have been significant efforts to identify SARS-CoV-2 HLAIps, mainly focusing on characterizing the immunogenicity in vitro or in vivo of large libraries of synthetic peptides of in silico predicted HLA-binders (25 studies reviewed in ref. 20). This has provided important insights into possible immunodominant regions in the viral proteome, HLA allele-dependent responses to

SARS-CoV-2, and the protection capabilities of vaccines (reviewed in refs. 20–22). More than 2000 putative HLA-binding peptides have been predicted from the SARS-CoV-2 genome[23]. However, only a few SARS-CoV-2 HLAIps have been detected by LC-MS until now[24–26], including less than ten HLAIps for the spike glycoprotein refs. 24–26, the main target of vaccines and diagnostic tests. This emphasizes the challenges of LC-MS immunopeptidomics and the need for more sensitive and robust methods.

Here, we present Thunder-DDA-PASEF, an optimized LC-IMS-MS method for immunopeptidomics, and its application in discovering SARS-CoV-2 spike HLAIps. The optimized method uses an extended TIMS separation time (300 ms) to improve IMS resolution and sensitivity[18,27]. To include singly charged peptides while efficiently using instrument cycle time, precursors are selected using a tailored isolation polygon for semi-selectively fragmenting potential HLAIps. Compared to the Standard method, Thunder-DDA-PASEF doubled (on average) the HLAIps identifications across samples with diverse HLA alleles. In addition, to increase the number and confidence of peptides identified, we trained a specialized MS²PIP model[28] to predict peptide fragmentation in timsTOF instruments. Using this model to rescore the results with MS²Rescore (v3)[16,29] boosted HLAIp identifications by 41.7% to 33%, resulting in 5738 HLAIps detected from as little as one million JY cell equivalents, and 14,516 HLAIps from 20 million. Subsequently, we employed Thunder-DDA-PASEF to study the HLA-I ligandome repertoire of two cell lines recombinantly expressing the segments of the spike protein of SARS-CoV-2. This resulted in 16 HLAIps derived from the SARS-COV-2 spike protein. Notably, 13 of these peptides have been previously reported to elicit immune responses in human patients, confirming the potential of our improved method for efficient epitope discovery. In conclusion, the optimized Thunder-DDA-PASEF boosted by MS²Rescore achieved deep profiling of the HLA class I ligandome even from low sample inputs.

## Results
### General workflow for LC-IMS-MS immunopeptidomics
For our immunopeptidomics experiments, we followed the procedure shown in Fig. 1 and described in Material and Methods. The LC-MS methods and data processing settings are detailed in Supplementary Data S1. The ready-to-use MS method for timsTOF Pro instruments is included in Supplementary Data S2. Briefly, we enriched HLAIps from JY cells by immunoprecipitation using W6/32 antibody, and analyzed them by nanoLC-IMS-MS on a nanoElute coupled to timsTOF-Pro-2 in DDA-PASEF mode, using PEAKS XPro for subsequent peptide identification. After training an MS²PIP model[28] to predict peptide fragmentation for timsTOF data, we used it in an updated version of MS²Rescore[16,29] (v3.0.0b4) to improve the identifications for the dataset exploring the SARS-Cov-2 spike immunopeptidome. To evaluate the identification of possible HLA class I ligands, we predicted the peptide binding to the respective HLA alleles of each sample using NetMHCpan-4.1[30] via MhcVizPipe[31]. Here, we refer to the predicted HLAI peptide binders ($rank \leq 2\%$) as HLAIps to distinguish them from the total peptides identified, and both terms indicate the stripped sequences unless otherwise specified. We performed several iterations to optimize our LC-IMS-MS method for identifying HLA class I ligands, as described in the following sections.

### An HLAIp-tailored DDA-PASEF fragmentation scheme including singly-charged ions efficiently identified possible HLAIps
Contrarily to tryptic peptides, HLAIps originate from a large diversity of antigen processing events[13,14] and do not necessarily contain basic amino acid residues[2]. Thus, many HLAIps can only be detected as singly-charged ions in LC-MS since only their N-terminal residue can carry a positive charge (H⁺). For this reason, HLAIp-immunopeptidomics workflows have recently incorporated the

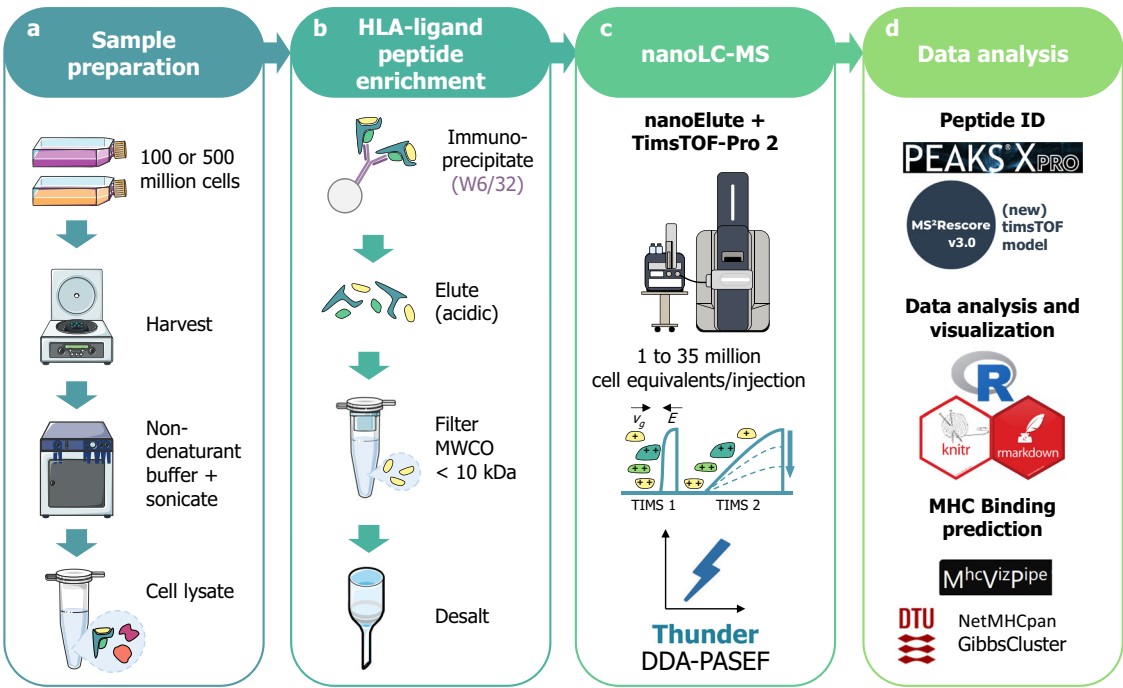

**Fig. 1 | Immunopeptidomics workflow using Thunder-DDA-PASEF. a** Sample preparation: 100 or 500 million cells of diverse cell lines were harvested, then lysed by sonication in 1% CHAPS in PBS buffer (m/v). Alternatively, 4 mL of unfractionated plasma were processed. **b** HLA-ligand peptide enrichment: was performed by immunoaffinity using the W6/32 anti-HLA-A, B, C antibody coupled to CNBr-activated agarose beads; after overnight incubation and several washes, peptides were eluted with 0.2% trifluoro-acetic acid (v/v), ultrafiltered on molecular weight cutoff filters (MWCO, 10 kDa cutoff) and desalted in HLB plates (Waters Corp.).

**c** NanoLC-MS: analysis was performed using a nanoElute coupled to timsTOF-Pro-2 in DDA-PASEF[18] with different parameters to optimize the MS acquisition. **d** Data analysis: Database search was performed in PEAKS XPro using unspecific cleavage. After training a MS²PIP[28] timsTOF fragmentation prediction model, peptide identification was rescored using MS²Rescore (MS²R, v3.0.0b4)[16,29]. Data analysis was performed in R and predicted MHC-binding affinity was evaluated using NetMHCpan-4.1[30] and GibbsCluster-2.0[64] through MhcVizPipe (v0.7.9)[31].

fragmentation of singly-charged ions (with 2$^+$ and 3$^+$) within the m/z range of possible HLAIps[17,32–36]. In addition, HLAIps have a restricted size range of typically 9–12 amino acids (AAs)[2], but between 8 to 13 AAs in some instances[37,38]. Therefore, LC-MS immunopeptidomics methods require instrument-specific adaptations that differ to standard proteomics methods. Since this had not been fully studied in timsTOF instruments, we aimed to specifically optimize a DDA-PASEF method for immunopeptidomics.

First, we tested the Standard-DDA-PASEF method for proteomics[18] to analyze JY HLAIps samples (Fig. 2a, d, g). DDA-PASEF takes advantage of the charge-state-dependent mobility separation to selectively fragment ions detected within an isolation polygon on the inverse reduced ion mobility (1/K$_0$) vs. m/z space. Since it was designed for tryptic peptides, the Standard isolation polygon covers the multiply-charged ion cloud, which is clearly separated from singly-charged ions (Fig. 2a). This resulted in almost 5000 unique peptides from three injection replicates of JY HLAIps, mainly comprising doubly-charged ions (89%, Fig. 2d, g) and almost 77% of 8–13-mers (Fig. 2g, j). As expected, most singly-charged ions were excluded from fragmentation, and only a few were identified due to IMS peak tailing into the isolation polygon. This was similar to the results of a previous study where DDA-PASEF with the Standard polygon was used to profile the JY immunopeptidome[39] and the TIMS range was limited to 0.6–1.3 1/K$_0$, effectively excluding the singly charged peptides (Supplementary Fig. S1 and Supplementary Notes).

To enable the identification of singly charged peptides, our next step was to remove the isolation polygon (Fig. 2b) and extend the TIMS range to 1.75 1/K$_0$. This resulted in the fragmentation of singly-charged peptides, representing more than half (54.5%) of all the peptides identified and 59.6% of the 8-13-mers (Fig. 2e, h). Furthermore, the proportion of peptides with 8 to 13 AAs was 12.4% higher than in the

Standard-polygon (Fig. 2k, j), corresponding to 72% more 8-13-mers identified on average ($p = 8.1 \times 10^{-5}$, Fig. 2m). However, without an isolation polygon, many low m/z singly-charged ions and high mass multiply-charged ions were fragmented (Fig. 2b, e).

Therefore, we designed fragmentation isolation polygons covering the singly-charged and multiply-charged 8-13-mer peptides (Fig. 2c, f, i; Supplementary Data S1, tab isolation_polygon_thunder). Since the resulting isolation polygon resembles a lightning or thunder icon, we termed it the "Thunder" polygon. This HLAIp-tailored scheme identified peptides within the isolation polygon (Fig. 2c, f, i), roughly maintaining the charge distribution of peptides identified, and marginally increasing the proportion of 8-13-mers compared to no-polygon (by 2.6%) (Fig. 2l, k, respectively). As a result, the Thunder polygon increased the identification of 8–13-mers on average by 70.6% relative to the Standard ($p = 7.7 \times 10^{-5}$, Fig. 2m)). Compared to no-polygon, the Thunder polygon resulted in 24% fewer MS2 scans ($p = 5.9 \times 10^{-4}$, Fig. 2n), but a similar yield of 8-13-mers identified (Fig. 2m). This 18% increase in the identification rate shows that the Thunder polygon used the cycle time more efficiently to fragment 8-13-mers. In contrast, without an isolation polygon, a large proportion of the cycle time was used inefficiently to fragment ions that are not of interest for HLAIp profiling. These may include non-peptidic small ions or larger peptides (Fig. 2b and Supplementary Fig. S2a, b) originating from the degradation of HLA proteins, the antibody, or other co-enriched proteins (Supplementary Fig. S2c). Results Section 3 will provide additional insight into the specifics of the thunder-shaped region.

**Optimizing TIMS and MS settings further enhanced the identification of 8-13-mers**

Once having established the capabilities of DDA-PASEF with the Thunder isolation scheme for immunopeptidomics, we thoroughly

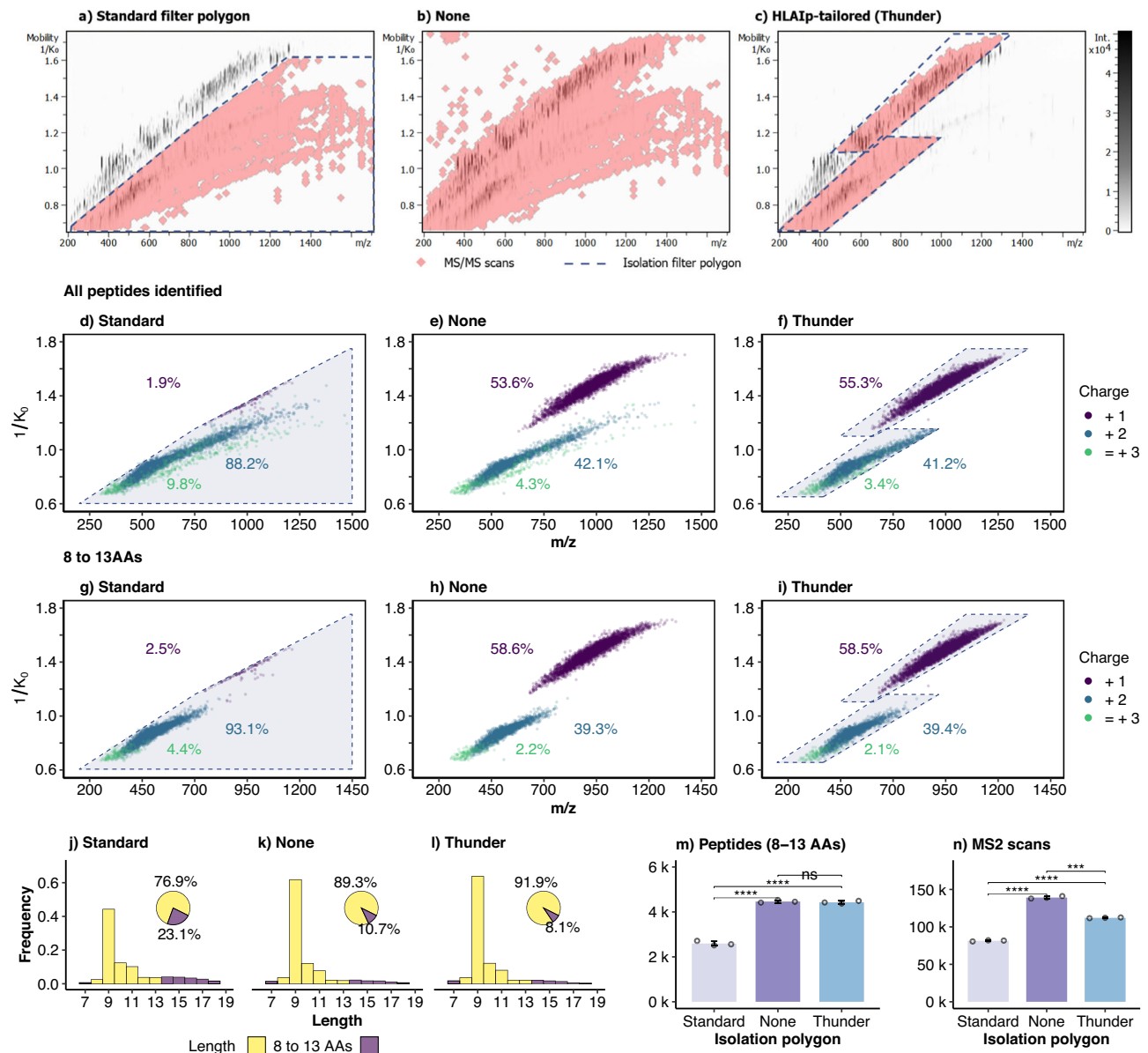

**Fig. 2 | Evaluation of the different fragmentation isolation filters: Standard, None and HLAIp-tailored (Thunder).** **a**–**c** Exemplary heatmaps of ion intensities (gray-scale) across the inversed ion mobility (1/K₀) vs m/z dimensions showing fragmentation events (red rhombi). **d**–**i** Corresponding peptides identified across the 1/K₀ vs m/z dimensions colored by charge state, including all peptides (**d**–**f**) or only those with 8–13 amino acids (AAs) (**g**–**i**); the dotted lines delimit the perimeter of the respective fragmentation isolation polygons. **j**–**l** Length distribution and percentage of peptides (pie-charts) with 8 to 13 AAs or other lengths; cut-off at 20 AAs dropping

5.4%, 1.6%, and 0.26% of peptides identified for Standard, None and HLAIp-tailored, respectively. **m** Unique peptides (with 8–13 AAs) identified per injection in each method ($p_{Standard,None} = 8.1 \times 10^{-5}$, $p_{Standard,Thunder} = 7.7 \times 10^{-5}$, $p_{None,Thunder} = 0.49$). **n** Number of MS2 scans triggered per injection in each method. In m, n points represent individual injections (3 replicates); bars and error bars show *mean* ± *sd*, $p_{Standard,None} = 3.0 \times 10^{-5}$, $p_{Standard,Thunder} = 2.5 \times 10^{-6}$, $p_{None,Thunder} = 5.9 \times 10^{-4}$). Two-sided *t*-test, ns: $p > 0.05$, *$p \leq 0.05$, **$p \leq 0.01$, ***$p \leq 0.001$, ****$p \leq 0.0001$.

optimized other parameters of the MS method aiming to increase both the number of peptides identified and the reproducibility. In PASEF, each analysis cycle comprises several frames where the trapping TIMS tunnel accumulates a package of ions. Simultaneously, the second TIMS resolves the previous package of ions by ramping down the elution voltage. Increasing TIMS times enhances IMS resolution and accommodates more fragmentation events per MS2 frame while preserving the sensitivity[18]. Raising the TIMS time from 100 to 300 ms resulted in an significant increase of 80% in peptide identification, while no substantial increase was observed between 400 ms and 300 ms (< 5% increase, non-significant) (Fig. 3a). The data completeness also increased slightly between 100 ms (50.2%) and 300 ms (55.6%) (Fig. 3b). However, the longer cycle times resulted in five-fold fewer

MS1 frames and doubled the median coefficient of variation (CV) at 400 ms compared to 100 ms (Fig. 3c–e). To compensate for this and to make the method compatible with 1 h LC gradients, we decreased the number of MS2 frames per cycle from 10 to 3, and the MS2 cycle overlap from 4 to 1 (Fig. 3f–j). This resulted in a 1.2 s cycle time, without significantly impacting the number of peptides identified but improved data completeness by 5% and reduced the median peak area CV from 19.3% to 10.3% associated with an increase in MS1 frames (Fig. 3j, f–i, respectively), thus improving the reproducibility overall. We selected the configuration with 300 TIMS and 3 MS2 frames for further analyses since it provided the optimal balance between peptide identification and reproducibility in terms of data completeness and peak area.

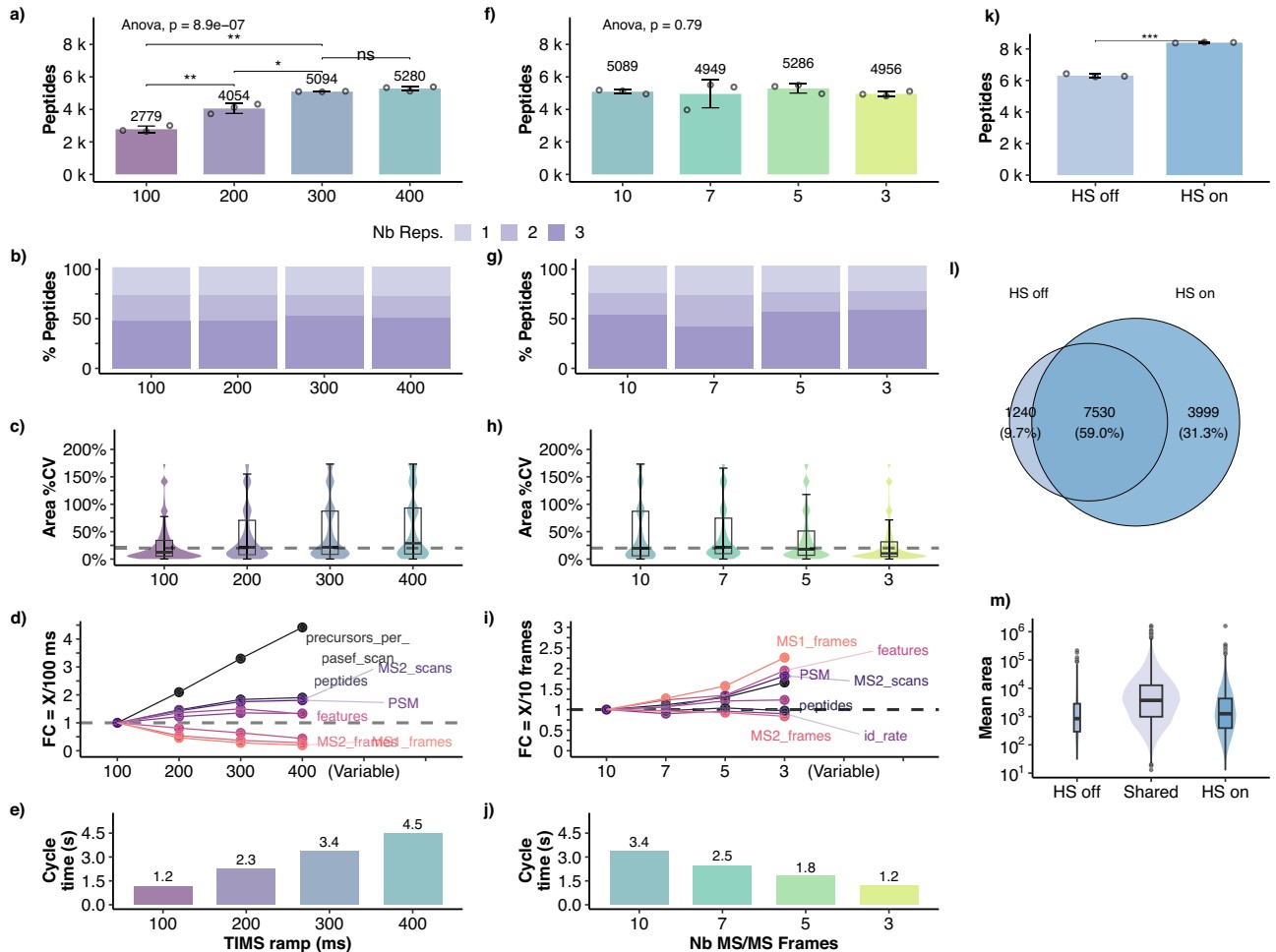

**Fig. 3 | TIMS and MS optimization. a–i** Effect of the TIMS ramp duration (a to d) and number of MS2 frames (**f–i**) on peptide identification (3 injection replicates each). (a, f): Unique peptides identified per injection (points represent individual injections; bars and error bars show *mean ± sd*). **a** ANOVA followed by two-sided *t*-test (ns: $p > 0.05$, $*p \leq 0.05$, $**p \leq 0.01$; $p_{100,200} = 0.0058$, $p_{200,300} = 0.027$, $p_{300,400} = 0.15$, $p_{100,300} = 0.0023$). shown only for the neighbouring methods and to compare the starting and final settings, 100 and 300 ms. **f** ANOVA indicating no significant differences between the methods. **b**, **g** Data completeness shown as the proportion of peptides identified in one, two, or all the three replicates. **c**, **h** Violin and boxplots of the peak area coefficient of variation (CV) of peptides identified in at least 2 replicates (center line, median; box limits, upper and lower quartiles; whiskers, 1.5x interquartile

range). **d**, **i** Spaghetti plot of the log2 fold change (log2FC) of identification parameters for each condition as compared against the initial (100 ms or 10 MS2 frames, respectively). **e**, **j** Cycle time (seconds) for either varying TIMS ramp durations (**d**) or number of MS2 frames per cycle (**h**). **k–m** Effect of the high sensitivity (HS) mode on peptide identification and dynamic range (3 injection replicates). **k** Unique peptides identified per injection in each method (points represent individual injections; bars and error bars show *mean ± sd*; two-sided *t*-test, $***p = 4.4 \times 10^{-4} < 0.001$). **l** Venn plot showing the overlap between HS mode on or off. **m** Peptide area (mean of three replicates) distribution for peptides identified only with HS off, only with HS on or shared by both, with widths scaled in function of the number of peptides (center line, median; box limits, upper and lower quartiles; whiskers, 1.5x interquartile range).

The timsTOF Pro-2 has a high-sensitivity mode (HS), in which the detector voltages are optimized for low sample amounts. To evaluate its effect on immunopeptidomics samples, we activated the HS mode in the method with 300 ms TIMS and 3 MS2 frames. As a result, peptide identification significantly increased by 33% (Fig. 3i). From the 12,769 peptides identified in total in this experiment, 49% were shared between HS on or off, only 9.7% were exclusive to HS off, and 31.3% were exclusively identified with HS on (Fig. 3j). The distribution and median of the area of the additional identifications were half an order of magnitude lower than for the shared peptides (Fig. 3k). Thus, peptides with low intensity benefited the most from the HS mode, but it also resulted in more identifications in the upper intensity range.

In summary, the optimized method resolves ions using a 300-ms TIMS ramp, fragmenting mainly ions with 1⁺, 2⁺, and 3⁺ charges in 3 MS2 frames per MS1 frame within a 1.2 s cycle time and takes advantage of the high-sensitivity mode. In contrast, the Standard-DDA-PASEF designed for proteomics samples uses 100 ms ramps and selectively

fragments multiply-charged ions in 10 MS2 frames per MS1 frame within a 1.2 s cycle time. For simplicity, in the following sections we refer to the methods with the Standard and optimized settings as 100 ms and 300 ms, respectively, and to the fully optimized method including the HLAIp-tailored isolation polygon as Thunder-DDA-PASEF.

**The optimized Thunder-DDA-PASEF doubled the identification of HLA class I ligands across diverse cell lines and human plasma**
The high polymorphism of HLA across the population results in a large variety of diverse peptide sequence motifs with distinct anchor amino acids required to bind the protein complexes depending on the HLA allotypes. To better understand the effects of the HLAIp-tailored Thunder isolation polygon and the DDA-PASEF optimization across distinct HLA-binding peptide motifs, we analyzed samples with diverse alleles using methods without and with the HLAIp-tailored isolation polygon (Thunder), combined with the Standard (100ms_None and 100ms_Thunder) and optimized DDA-PASEF settings (300ms_None and 300ms_Thunder) (Fig. 4 and Supplementary Data S3 and S4). We

used HLAIps enriched from cell lines covering 17 distinct HLA binding motifs, JY (HLA-A0201, B0702, C0702), HeLa (HLA-A68:02, A03:19, B15:03, C12:03), and SK-MEL-37 (HLA-A0201, A1101, B1501, B5601, C0102), and commercial human plasma (predicted as HLA-B0702, B1501, A0101, A2601 using MHCMotifDecon (ref. 40).

First, we focused on the effect of the isolation polygon on peptide and, more specifically, HLAIp identification, as shown in Fig. 4a for the combined datasets, and in Supplementary Figs. S3 (all peptides) and S4 (HLAIps only) for each sample. As expected, the Thunder isolation polygon limited the fragmentation and identification of peptides to the area within its boundaries, while no polygon resulted in a more extensive distribution of peptides across the $1/K_O$ vs. m/z dimensions. However, even in the acquisitions without polygon, 99.9% of the predicted HLAI binders (HLAIps) were detected within the defined boundaries of the Thunder polygon, showing that it does not exclude peptides from any of the HLA motifs screened.

For all the samples, the methods employing the Thunder polygon resulted in higher identification of total peptides than the analogous methods without polygon, and a similar effect was observed for HLAIps (Fig. 4b, c). At 100 ms, using the isolation polygon increased the HLAIp identification by 7.3% for JY, 6.3% for HeLa, and 8.8% for SK-MEL-37 but did not affect numbers in plasma. At 300 ms, the method with the isolation polygon increased the identifications by 16.5, 13.1, 9.4, and 5.1% for JY, plasma, HeLa, and SK-MEL-37, respectively. For each sample type, the peptide charge distribution across all peptides (Fig. 4d), the proportion of 8–13-mers (Fig. 4e), and the proportion of 8-13-mers predicted to bind the corresponding HLA alleles (Fig. 4f) were roughly similar without and with the polygon. The differences were below 1.5% in most cases. Similarly, the dynamic range of peptides identified was not altered (Supplementary Fig. S5). These results show that the Thunder polygon increases peptide identification without introducing a bias in the type of peptides identified, and its

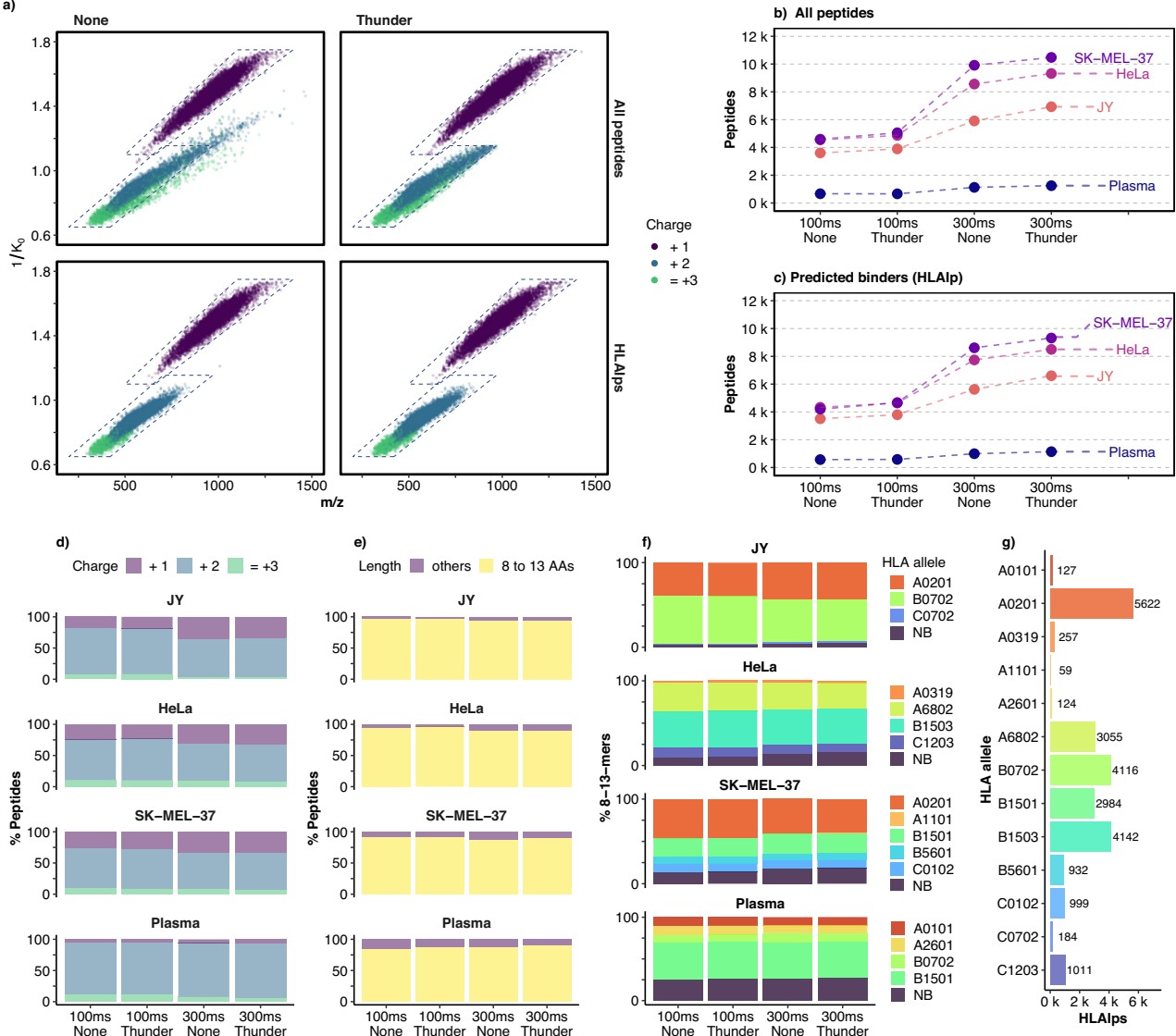

**Fig. 4 | Evaluation of the effects of the Thunder isolation polygon and DDA-PASEF optimization on HLAIp identification from samples with distinct peptide HLA-binding motifs. a** All peptides (top) and predicted HLA binders (HLAIps, bottom) identified across the $1/K_O$ vs m/z dimensions; the left panels show all the peptides identified without polygon for all samples, and the right panels all the peptides identified with the Thunder polygon; the dotted lines delimit the perimeter of the Thunder isolation polygon. **b** Total peptides identified from two injection replicates in each method. **c** Total HLAIps identified from two injection replicates in each method. **d** Proportion of peptides (considering modifications) identified in function of their charge state. **e** Proportion of peptides with 8–13 AAS (8-13-mers). **f** Proportion of 8-13-mers predicted to bind each of the HLA alleles of the sample, or none of them (non-binder, NB). **g** Total HLAIps in the whole dataset obtained from the four samples and four methods in function of the HLA allele with the highest binding affinity.

beneficial effects are more pronounced for the 300 ms method. This is likely due to the enhanced IMS separation and MS sensitivity provided by the longer TIMS time and the HS mode. As a result, more fragmentation events can be included per cycle, and low-abundant ions are more likely to be fragmented. To use instrument time most efficiently on ions representing putative HLAIps, the thunder polygon minimizes the fragmentation of chemical noise and other peptides.

Subsequently, we evaluated the effect of the DDA-PASEF optimization. Compared to their 100 ms counterpart, the 300 ms methods resulted in 64.2 to 120% more peptides (Fig. 4b). The charge distribution shifted towards more singly charged ions at 300 ms (Fig. 4d), and the proportion of 8-13-mers marginally decreased in the cell lines but not in plasma (Fig. 4e). From the 100 ms to 300 ms methods, the proportion of singly charged ions almost doubled for JY (from 19 to 35%), increased considerably for HeLa (from 25 to 32%) and SK-MEL-37 (from 28.5 to 35.5%), and only marginally for plasma (from 5.5 to 6.5%). The proportion of predicted non-HLA binders among the 8-13-mers increased slightly (by 0.5 to 5%) but did not change the proportion of predicted binders for each HLA allele (Fig. 4f). Although the improved peptide coverage comes at the cost of marginally increasing the detection of non-specific co-enriched peptides, the 300 ms methods still increased the number of HLAIps identified by 61.1 to 105% compared to the corresponding 100 ms methods (Fig. 4c. Moreover, the 300ms_Thunder method resulted in most HLAIps across all methods and samples types. When compared to 100ms_None, the 300ms_Thunder provided a total increase of 87.4 to 125% in HLAIps depending on the sample, thus in average doubling the immunopeptidome coverage.

The resulting combined dataset contains peptides covering thirteen HLA alleles with diverse sequence motifs, as shown in Supplementary Fig. S6 for the 9-mers. The number of peptides covering each allele varies from 59 for HLA-A11:01, found in the plasma sample only, to 5622 for HLA02:01, from JY and SK-MEL-37, (Fig. 4g), providing valuable information to evaluate their IMS vs. m/z distributions (Supplementary Fig. S7). HLAIps with motifs dominated by hydrophobic amino acids resulted in a large proportion of peptides at the top section of the polygon and thus a higher proportion of singly charged peptides. On the contrary, motifs dominated by basic residues showed fewer peptides at the top section due to a lower proportion of singly charged peptides. For instance A68:02 and HLA02:01 included 47.7% and 47.3% singly charged peptides, respectively, while B15:03, included only 13.8% (Supplementary Fig. S8).

In summary, these results show that the enhanced coverage of Thunder-DDA-PASEF is primarily due to the optimized DDA-PASEF settings. Still, the Thunder isolation polygon provides a further boost without sacrificing the identification of diverse HLAIps regardless of their allele-specific sequence motifs. Combining these two factors, the optimized (300ms) Thunder-DDA-PASEF doubled the immunopeptidome coverage across a large diversity of HLA alleles, compared to the Standard (100ms) DDA-PASEF with extended $1/K0$ range and also including singly charged peptides.

### Thunder-DDA-PASEF outperforms state-of-the-art MS immunopeptidomics methods

To compare the performance of Thunder against other state-of-the-art methods for immunopeptidomics making use of the singly-charged ions, we analyzed aliquots of a JY HLAIps sample using Thunder-DDA-PASEF in a timsTOF Pro-2 and using DDA in an Orbitrap Exploris 480. Although the LC and columns differed between the instruments ("Methods" section), the gradient duration (47 min) and total LC time were similar (1 h). In the Exploris, we acquired data without FAIMS (wo-FAIMS) or using gas phase fractionation (GPF) with FAIMS either switching between three compensation voltages (CVs, −20, −50, −70) during the acquisition (FAIMS-3CVs) or using single CVs per acquisition from −10 to −80 in steps of 10 (FAIMS-8CVs), as described in ref.

17. We acquired triplicate injections of 10 million cell equivalents in each method but only in a single replicate for each CV of the FAIMS-8CVs experiment. In addition, we compared our results to recent reports profiling the JY HLAI-ligandome (Fig. 5). Demmers[41]. used DDA on an Orbitrap Fusion Lumos Tribrid with electron-transfer/higher-energy collision dissociation (EThcD) fragmentation, and a 90 min LC gradient. They excluded singly charged ions since EThcD is only suitable for charge states >+1. We reprocessed the results of three injection replicates in PEAKS XPro. Pak et al. employed a data-independent acquisition (DIA) method in an Orbitrap Q-Exactive MS and a 110 min gradient. We downloaded the results of JY HLAIps identified using a multi-sample DDA-derived spectral library ("BigLib") compiled from 190 measurements (DIA BigLib)[33].

Thunder-DDA-PASEF resulted in the most identified peptides (12,639 HLAIps in total; Fig. 5a); thus higher than the Exploris wo-FAIMS (3.7-fold), FAIMS-3CVs (5.6-fold), FAIMS-8CVs (1.8-fold), Demmers (1.5-fold), and Pak (2.4-fold). The low performance of the FAIMS-3CVs may be due to the high cycle time (>3 s), resulting in a low duty cycle. Notably, the FAIMS-8CVs required almost three times more instrument time and sample consumption than Thunder but provided only half the number of HLAIps. Moreover, Thunder covered 60% of the identifications resulting from combining all the results (19,397 HLAIps; Fig. 5a) and provided the highest proportion of exclusively-identified HLAIps (24.5%; Fig. 5b). Demmers obtained the most similar coverage compared to Thunder-DDA-PASEF[42] and provided 18.7% exclusive identifications, confirming the complementarity of EThcD for immunopeptidomics experiments[42].

To evaluate the performance of the methods for identifying singly charged HLAIps, we plotted their charge distribution (Fig. 5c). The FAIMS-3CVs resulted in the highest proportion of HLAIps exclusively identified as singly-charged (36.9%), followed by Thunder (30.0%), the Exploris wo-FAIMS (27.0%), and Pak (24.8%). In the combined dataset, 24.8% of all the JY HLAIps identified were only detected as singly charged (Fig. 5c). Notably, in only three injections, Thunder-DDA-PASEF detected 91.9% of all the singly charged peptides in the data set, and 67.7% were exclusive (Fig. 5d). This indicates that the ion mobility separation is not essential to identify singly charged peptides. However, Thunder-DDA-PASEF contributed the most to singly charged peptides in the full dataset combining the timsTOF and Orbitrap results, due to the enhanced overall coverage achieved by the method.

In summary, these results indicate that Thunder-DDA-PASEF outperforms all the Orbitrap-based methods tested. In addition, Thunder-DDA-PASEF does not require a spectral library or many injections, resulting in a more straightforward approach and requiring less sample than library-based DIA and GPF DDA workflows.

### A workflow including an updated MS²Rescore with newly trained timsTOF MS²PIP model identifies more than 5700 HLAIps from 1 million JY cell equivalents

Several post-processing tools have shown improvements in immunopeptide identification by rescoring peptide spectrum matches (PSMs) based on characteristics disregarded in the initial search[16,34,43,44]. To improve the peptide identification, we implemented MS²Rescore (MS²R, v3.0.0b4)[16,29] into our workflow. Since peak intensity predictions rely heavily on the fragmentation method, instrument or labeling method[28,45], we trained a timsTOF-specific peak intensity model using in-house timsTOF data. In addition, it has been shown that prediction models trained on datasets including both, tryptic and non-tryptic peptides, can perform equally well for predicting the fragmentation of both types of peptides[16,34]. Therefore, we trained a MS²PIP[28] timsTOF fragmentation prediction model using data acquired at two different labs from immunopeptidomics (HLA class I), tryptic peptides, and elastase digest samples. In total, 241,104 peptides (including modifications) were used to train the model and a distinct set of 10,045

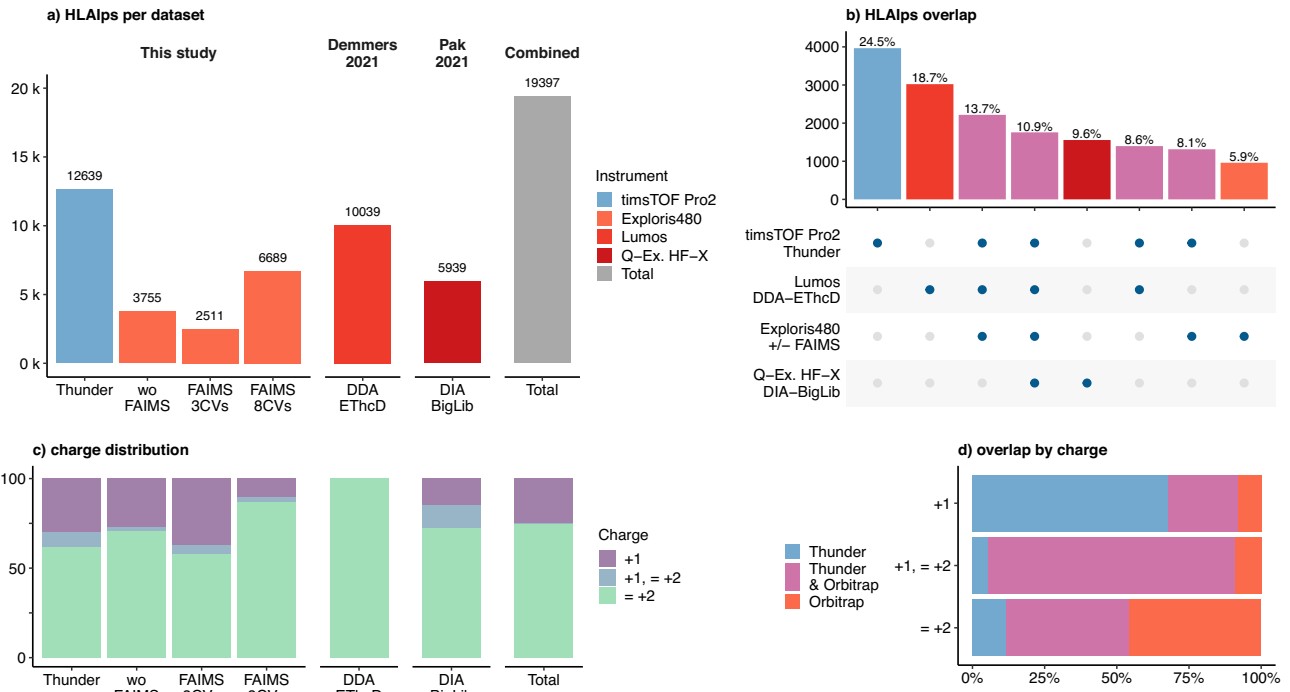

**Fig. 5 | Comparison of Thunder-DDA-PASEF to previous MS immunopeptidomics methods analyzing JY HLA1p-enriched samples.** In this study, aliquots of a JY HLAIp sample were acquired on a timsTOF Pro-2 using Thunder-DDA-PASEF (Thunder; *n* = 3 injection replicates) or on an Orbitrap Exploris 480 using DDA without FAIMS (wo-FAIMS; *n* = 3) or with FAIMS either switching between three compensation voltages (CVs: −20, −50, −70) during the acquisition (FAIMS-3CVs; *n* = 3) or using single CVs per acquisition from −10 to −80 in steps of 10 (FAIMS-8CVs; *n* = 1 each). In addition, two previously reported JY HLAIp-profiling data sets were included: Demmers 2021[41] using an Orbitrap Fusion Lumos in DDA with EThcD fragmentation (DDA EThcD; *n* = 3) and Pak 2021[33] using a Q Exactive (Q-ex.) HF-X in DIA and a big library for data analysis (DIA BigLib; *n* = 3). **a** Total HLAIps identified predicted to bind the JY HLA alleles (*rank* ≤ 2%). Colors indicate the instrument used. **b** Upset plot showing the number (barplot) and percentage (text) of HLAIps identified exclusively in each instrument or their combinations; the intersection matrix at the bottom indicates that the same peptides shown above (columns) were detected in the instruments (rows) highlighted with a blue dot. **c** Proportion of peptides (considering modifications) identified in function of their charge state exclusively as + 1, in multiple charge states including + 1 (1, ≤ +2) or only as multiply charged (≤ +2). **d** Percentage of peptides identified exclusively by Thunder-DDA-PASEF (blue), exclusively by Orbitrap instruments (red) or in both type of instruments (pink) in function of the charge state.

peptides to test it. Retention times were predicted with DeepLC[46]. In addition, to take advantage of the ion mobility separation, we added the peptide CCS predicted using the ionmob GRU predictor as a rescoring feature, as recently described[47].

Overall, compared to the HCD prediction model the results greatly improved using the timsTOF model (Supplementary Fig. S9), especially for the JY Ramp_20-59eV run and the HL60 immunopeptides, which utilized the IMS-dependent CE ramp. Here, median prediction performance increased, respectively for the JY and HL60 datasets, from 0.5453 and 0.5109 Pearson correlation coefficient (PCC) for the Orbitrap HCD model to 0.8651 and 0.8706 PCC for the timsTOF model

To assess the sensitivity of Thunder-DDA-PASEF and how it is affected by rescoring, we analyzed diverse injection amounts of JY HLAIps. We prepared aliquots of a JY-HLAIp pool at 20 and 2 million cell equivalents (Mce) and analyzed triplicate injections of 1, 2, 5, 10, 20, and 40 Mce in column using the 1 h gradient and optimized Thunder-DDA-PASEF. Then, we analyzed the data without (PEAKS) or with rescoring (+MS²R). We processed the files belonging to distinct injection amounts independently to avoid introducing a bias due to identification transfer from higher to lower cell inputs. The respective results were filtered at *FDR* ≤ 0.01.

First, to better understand the overall effect of rescoring, we evaluated diverse peptide characteristics by combining the results of all the injection amounts (Fig. 6a–d). In total, PEAKS identified 28,725 peptides, and rescoring boosted the identifications by 35.4%, resulting in 38,901 peptides (Fig. 6a). Rescored results shared 27,990 peptides with PEAKS, gained 10,911, and lost 735 (Fig. 6a). Although the majority

of lost peptides were singly charged (68.8%), this only accounted for 508 peptides. In contrast, rescoring gained 3,112 (28.5%) and retained 8420 (30.1%) singly charged peptides (Fig. 6b). The proportion of 8-13-mers was similar across the peptides shared (78%), gained (81%), and lost (70%) (Fig. 6c). Thus, 21,801, 8856, and 514 8–13-mers were shared, gained, and lost, respectively (Fig. 6d). Most of the lost peptides were not predicted to bind HLAI (59.1%) while most of the shared (84.6%) and gained (58.6%) peptides were predicted binders (Fig. 6d). These results show substantial overall improvement in peptide identification and indicate that MS²R did not introduce a bias towards a specific type of peptide.

Then, to evaluate the sensitivity of Thunder-DDA-PASEF without and with rescoring, we focused on the impact of injecting decreasing cell equivalents (Fig. 6e). PEAKS identified 4,749 ± 862 on average per injection of 1 Mce, increasing up to 16,169 ± 164 peptides at 40 MCe. MS²Rescore boosted peptide identification by more than 31% across all the cell inputs. Rescoring resulted in a statistically significant increase starting from 2 Mce. The identifications went from 6,581 ± 1,022 to 21,485 ± 135 for 1 Mce and 40 Mce, respectively. In addition, the median area of gained peptides was lower than for the shared and lost peptides (Fig. 6f), indicating that peptides with low intensity benefited the most from rescoring. Besides, rescoring improved the data completeness even for lower inputs, reaching 60.5% of peptides identified across all three replicates for 1 Mce and up to 70.9% for 40 Mce, compared to 46.5% and 58.8% without rescoring, respectively (Fig. 6g). Finally, we focused on the netMHCpan-predicted binders (HLAIps) (Fig. 6h). PEAKS without

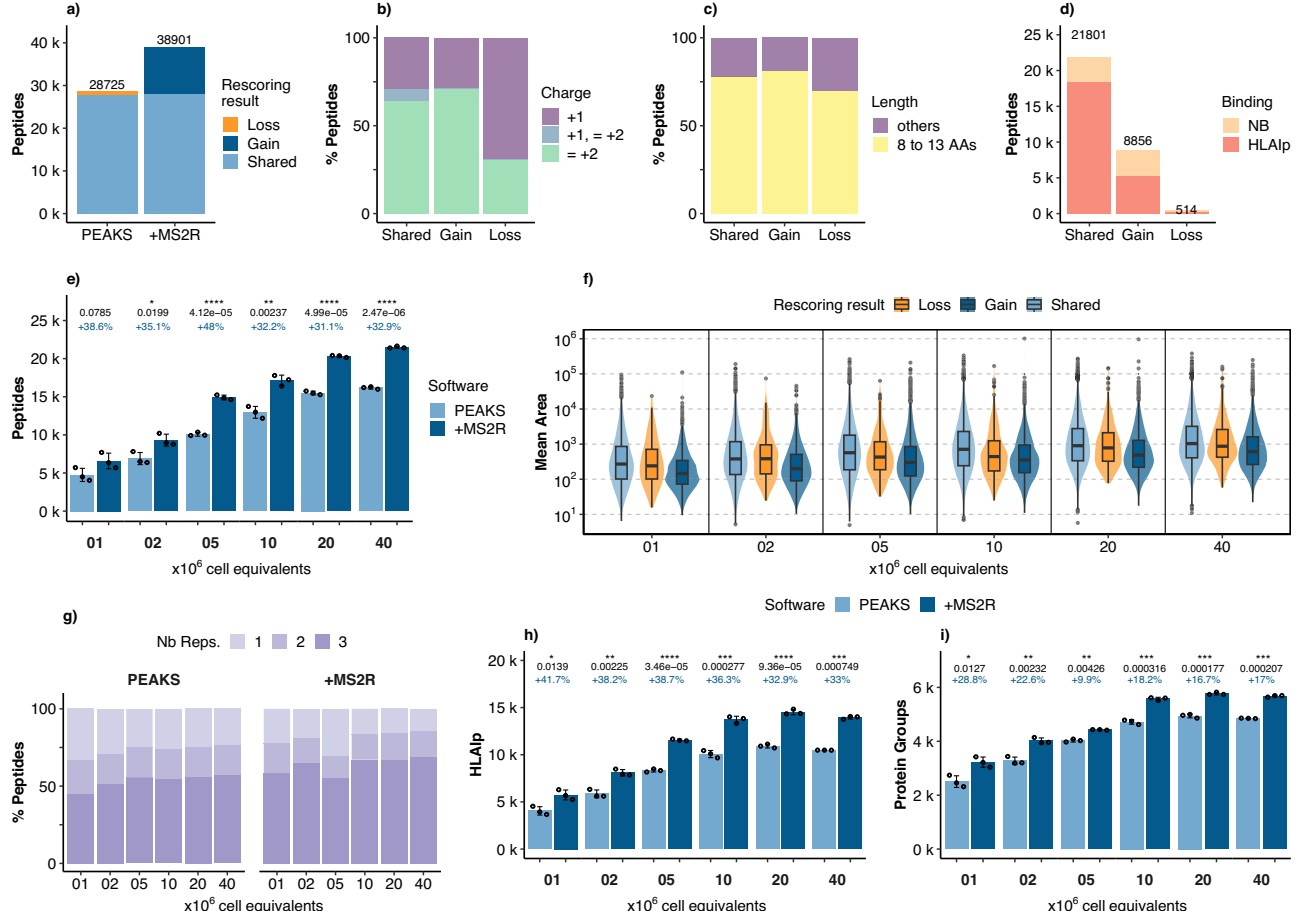

**Fig. 6 | Evaluation of the sensitivity of Thunder-DDA-PASEF and the effect of rescoring using the retrained MS²Rescore timsTOF immunopeptidomics model.** Injections of 1, 2, 5, 10, 20, and 40 million cell equivalents (Mce) of JY HLAIp samples were analyzed in triplicate using Thunder-DDA-PASEF. **a–d** Peptide characteristics of the whole dataset. **a** Peptides identified by PEAKS XPro alone (PEAKS) or by rescoring with MS²Rescore (+MS²R), colored in function of the rescoring result/impact. **b** Charge state distribution of peptides shared, gained or lost by rescoring. **c** Length distribution of peptides shared, gained or lost by rescoring. **d** Number of 8-13-mer peptides and the proportion predicted as HLA binders (HLAIp, $rank \leq 2\%$) or non-binders (NB, $rank > 2\%$) to the respective JY HLA types. **e–h** Peptide characteristics in function of sample input. **e** Total number of peptides identified by PEAKS alone or with rescoring (+MS²R). **f** Peptide area (mean of three replicates) distribution in function of rescoring result represented as violin and boxplots (center line, median; box limits, upper and lower quartiles; whiskers, 1.5x inter-quartile range). **g** Data completeness shown as the proportion of peptides identified in one, two, or all the three replicates. **h** Number of HLAIps identified by PEAKS alone or with rescoring (+MS²R); showing similar statistics as **e**. **i** Protein groups covered by the HLAIps identified. In **e**, **h**, **i** points represent individual injections; bars and error bars show mean ± sd, percentages in blue correspond to the total rescoring gain (Totalgainpercentage = 100 × Rescore/PEAKS); black values indicate $p$ values from a two-sided $t$-test, and asterisks indicate statistical significance, ns: $p > 0.05$ (label not shown), *$p \leq 0.05$, **$p \leq 0.01$, ***$p \leq 0.001$, ****$p \leq 0.0001$.

rescoring identified 4049 ± 443 HLAIps from only 1 Mce and up to 10,920 ± 200 from 20 Mce, where the number of HLAIps plateaued. Using MS²Rescore resulted in 5738 ± 525 HLAIps from 1 Mce, and 14,516 ± 272 from 20 Mce. Thus, rescoring boosted the HLAIps identification by 41.7% to 33%, respectively, significantly increasing the number of HLAIps across all sample inputs. Rescoring also significantly increased the number of protein groups covered by HLAIps (Fig. 6i). Using MS²Rescore resulted in 3225 ± 191 HLAIps from 1 Mce and 5765 ± 48 from 20 Mce. Thus, rescoring also boosted the immunopeptidome protein coverage, providing HLAIps for proteins that would have been missed otherwise.

Altogether, these results confirm previous observations showing that data-driven rescoring improves the coverage of immunopeptides with a higher gain for low sample inputs and low-intensity peptides (refs. 16,34). In addition, data-driven rescoring also improved the data completeness. In summary, our workflow using Thunder-DDA-PASEF and PEAKS+MS²R provides an in-depth characterization of the HLA class I immunopeptidome, obtaining more than 5700 predicted HLAIps per injection of 1 million JY cell equivalents.

## Thunder-DDA-PASEF identified 16 spike HLAIps in JY and Raji spike-transfected cells

To show an application of our workflow for identifying disease-related immunopeptides, we characterized the HLA class-I immunopeptidome of B-lymphoblastoid JY, and Raji cell lines transfected with a segment of the SARS-CoV-2 spike protein (Supplementary Material and Methods S1). Although SARS-CoV-2 has not been reported to infect B-cells, they commonly express HLAI complexes at high levels. In addition, the peptide binding motifs of JY and Raji HLA alleles broadly differ. Thus, we selected these cell lines to maximize the coverage and HLA diversity of spike immunopeptides.

Thunder-DDA-PASEF using PEAKS+MS²R identified in total 22,501 peptides from JY and 28,429 peptides from Raji, comprising 78% of 8–13-mers, with a median length of 9 AAs (Fig. 7a and Supplementary Data S5). The reproducibility between biological replicates ranged between 36.4% and 63.1% 8-13-mers identified in all the samples of the same genotype and 58.9% to 82% regarding the proteins covered (Supplementary Fig. S10). The 8–13-mers included 79.5% binders for JY and 78.1% for Raji (Fig. 7b). The specificity of HLAIp enrichment in this experiment was lower than in the samples

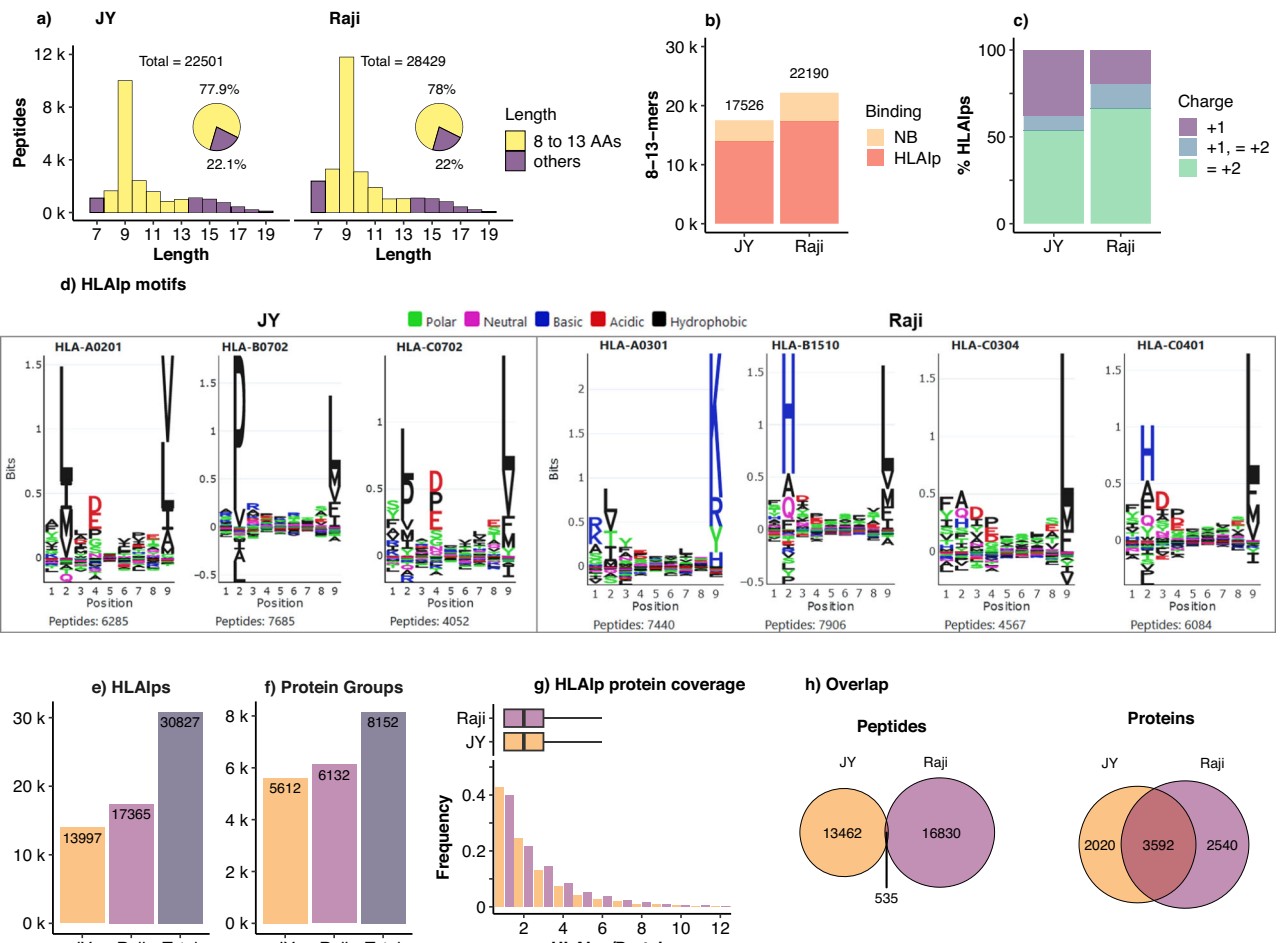

**Fig. 7 | HLA class I ligandome of JY and Raji cells employing Thunder-DDA-PASEF, combining wild type (WT) and spike-transfected cells (S1 and S2).** HLAIps were enriched from three cultures of each WT cell line and two different cultures of each transfected cell line and each sample was analyzed in triplicate. **a** Size distribution of total peptides identified from JY and Raji cells. **b** Number of 8-13-mer peptides identified in each workflow and the proportion predicted as HLA binders (HLAIp, *rank ≤ 2%*) or non-binders (NB, *rank > 2%*) to the respective HLA types. **c** Charge distribution for the predicted HLAIps. **d** Supervised clustering (GibbsCluster-2.0 via MhcVizPipe) showing the peptide sequence motifs corresponding to the specific allele motifs for JY and Raji HLAIps, respectively. **e** Total number of predicted HLAIps identified in JY, Raji, and in total. **f** Protein groups covered by the HLAIps identified. **g** Distribution of the number of HLAIps per protein group represented as boxplots (center line, median; box limits, upper and lower quartiles; whiskers, 1.5x interquartile range) (top) and histogram (bottom); y-axis cut-off at 12 for simplicity; thus, including 5569 and 6044 protein groups for JY and Raji, respectively; and excluding 0.77% and 1.43% of the protein groups (with 13–53 Binders/Protein) for JY and Raji, respectively. **h** Overlap of HLAI ligand peptides (left) and protein groups (right) between JY and Raji.

used for method optimization (>90%), primarily due to the higher contamination with larger peptides (22 and 22.5% vs. < 10%). Thus, we continued the downstream analysis on predicted HLA binders (Supplementary Data S6). A lower proportion of HLAIps was exclusively detected as singly-charged ions in Raji, compared to JY (33.5% vs. 46.1%, Fig. 7c). 9 This was due to the presence of basic amino acids at the anchor positions for Raji HLA allele HLA-B15 (Fig. 7d, right), including lysine or arginine at the C-terminus, resulting in a low proportion of singly charged peptides (Supplementary Fig. S11). In contrast, the anchor residues binding JY HLA alleles are dominated by apolar amino acids (Fig. 7d, left), resulting in higher proportion of singly charged ions in the most represented HLA allele, HLA-A02:01 (Supplementary Fig. S11).

In total, 13,997 and 17,365 HLAIps were detected in JY and Raji, respectively, summing up to 30,827 peptides (Fig. 7e). The HLAIps originated from a large variety of proteins, corresponding to 5,612 protein groups in JY, 6132 in Raji, and 8152 in total (Fig. 7f). Each protein group was crepresented by a median of two HLAIps and 75% protein groups by one to three HLAIps for both cell lines (Fig. 7g). As expected considering their HLA allotype differences, the JY and Raji immunopeptidomes were highly complementary, with only 1.7% of the peptides detected in both cell lines (Fig. 7h, left). Interestingly, the overlap of parent proteins was much higher since 44.1% of all the protein groups were covered by the ligandomes of the two cell lines (Fig. 7h, right). A gene ontology (GO) enrichment analysis using GOrilla[48] indicated a significant over-representation (*FDR≤0.001*) of proteins involved in essential processes, such as the metabolism of nucleic acids (GO:0090304), macromolecule biosynthesis (GO:0034645), macromolecule localization (GO:0033036), and regulation of the cell cycle (GO:0022402) (Supplementary Fig. S12 a–c and Supplementary Data S7). Thus, the cell lines presented complementary peptides for these same crucial proteins.

Then, we focused on the transfected SARS-CoV-2 spike protein and the GFP reporter included in the construct (Fig. 8a–d). Importantly, peptides from these proteins were only detected in the transfected cells and not in the wild-type cells. Three GFP-derived HLAIps were identified in JY and six in Raji cells (Fig. 8b), serving as a control for successful antigen processing of the transfected constructs. Five spike HLAIps were identified in JY and twelve in Raji (Fig. 8a) across a large dynamic range corresponding to four orders of magnitude (Fig. 8c). While the Raji spike HLAIps were distributed across the whole

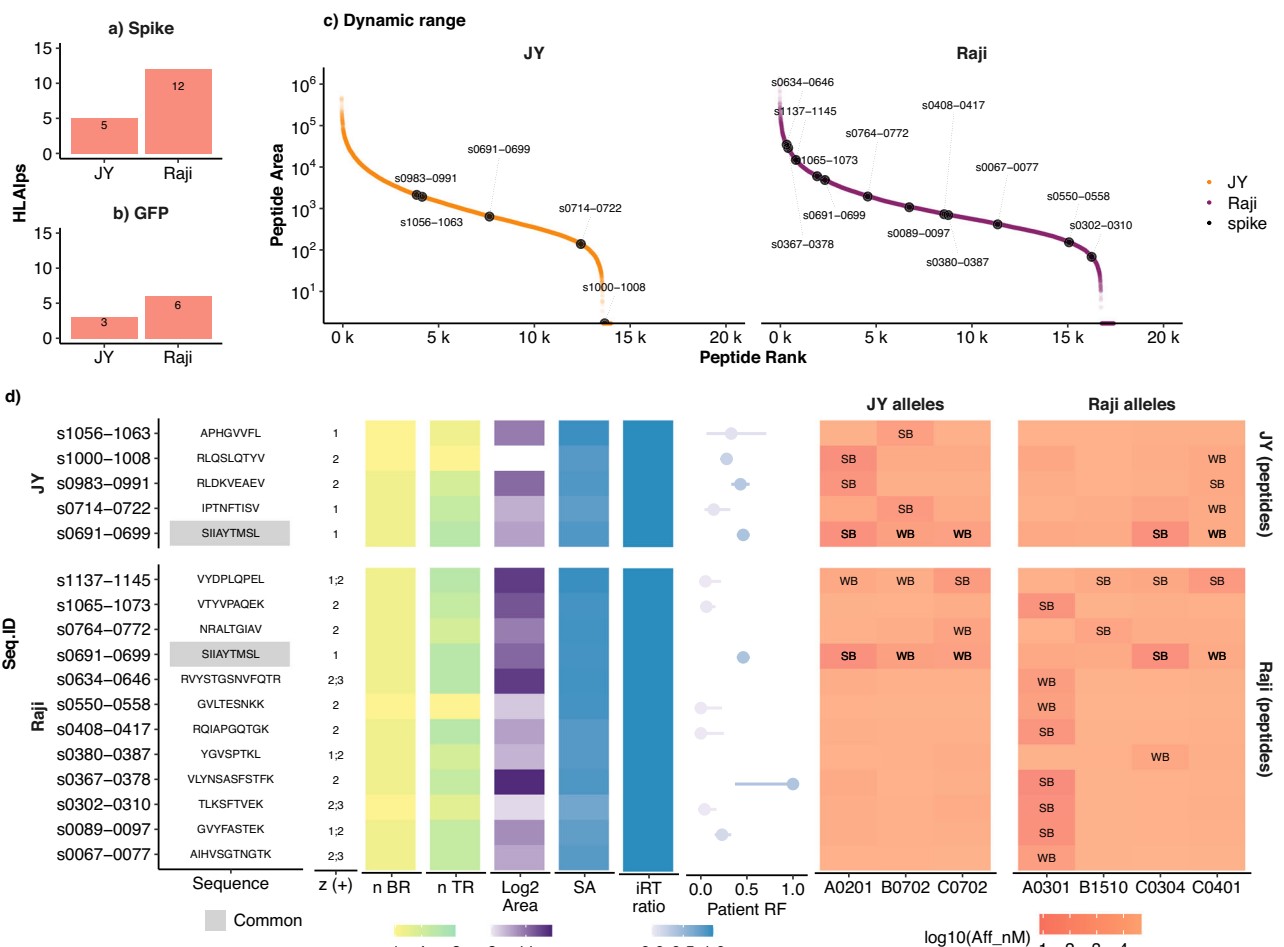

**Fig. 8 | Spike HLA class I binder peptides (HLAIps) identified in JY and Raji transfected cells. a, b** Count of protein-specific HLAIps predicted binders (HLAIp, *rank* ≤ 2%) for spike (**a**) and the reporter GFP (**b**). **c** Peptide peak area distribution of the spike peptides (black dots) and all the HLAIps identified in JY (orange) and Raji (purple). **d** Characteristics of spike HLAIps identified in JY (top) and Raji (bottom) transfected cells. From left to right: sequence code name indicating their position within the protein sequence (s[N-terminus]-[C-terminus], e.g., s0691-0699 for SIIAYTMSL); sequence, with common peptides highlighted in gray; charge state (number of H+); the number of biological replicates (BR) and technical replicates (TR) where the peptide was identified; Log2 of the peptide peak area; Spectral

Angle (SA) comparing the fragmentation spectrum of the endogenous peptide against synthetic peptides[34,73] calculated employing the Universal Spectrum Explorer (USE)[49]; indexed retention times (iRT) ratio (endogenous/synthetic); Immune Epitope Database and Analysis Resource (IEDB)[50] immune response frequency (RF) = proportion of subjects with positive immune response in B-cell or T-cell assays (dots = response frequency, lines = 95% confidence interval (CI) range, color scale = lower 95% CI, empty = not reported), relative to the total number of subjects tested for the corresponding peptide; binding affinity to JY and Raji HLA alleles predicted by NetMHCpan-4.1[30] as strong-binders (SB, *rank* ≤ 0.5%) or weak-binders (WB, 0.5% < *rank* ≤ 2%).

dynamic range, they were mainly in the middle to low range for JY. The sequence and characteristics of the spike HLAIps are shown in Fig. 8d and detailed in the Supplementary Data S8. The nomenclature in Fig. 8c, d denotes identified spike HLAIps (e.g., SIIAYTMSL[s0691-0699]) both by peptide sequence and position (N- to C-terminus) in the full-length spike protein.

In addition to showing a *FDR* < 0.6% (mokapot *q value*) and a posterior error probability (PEP) or *localFDR* < 6%, the spike HLAIps were assessed based on the number of identifications across biological and technical replicates (n BR, n TR; Fig. 8d, yellow to green scales), and by comparing their fragmentation spectral angle (SA) similarity[49] and indexed retention times (iRT) against synthetic peptides (Fig. 8d, blue scales). The mirrored spectra comparisons are shown in Supplementary Fig. S13. To maximize the identification confidence, we only report spike HLAIps detected in at least three injection replicates or with an SA ≥0.8. Only three of the 16 peptides did not fulfill both criteria, but all the peptides reported showed an iRT ratio endogenous/synthetic ≥0.99. Among these three peptides, GVLTESNKK[s0550-0558] from Raji and RLQSLQTYV[s1000-1008] from JY were identified in only one

injection replicate, but their SA were 0.92 and 0.85, respectively. Besides, although peptide TLKSFTVEK[s0302-0310] showed a low SA (0.71), it was identified in three injection replicates. The low SA may be due to low-quality fragmentation spectra resulting from low abundant ions since it was near the lowest end of the dynamic range (Fig. 8c). The peptide NSASFSTFK[s0370-0378] was detected in Raji and predicted to weakly bind HLA-A03:01 but was not included in the final report since it did not pass the selection criteria (*SA* = 0.79, detected in 2 injections).

To further assess the potential translational relevance of these peptides, we compared them to previously published spike HLAIps, and we evaluated their immune response frequency (RF) in patient-derived samples using published data downloaded from the IEDB. The RF is defined as the proportion of subjects with positive immune response in B-cell or T-cell assays (Fig. 8d, dots and bars with blue scale). Twelve of the spike HLAIps have been reported in the IEDB[50] (October 17, 2023) and seven of them showed a RF of at least 10% and up to 100%, indicating they are possibly immunogenic (Fig. 8d, dot range plot). Potential challenges of LC-MS immunopeptidomics are also exemplified here, since only one of the 16 spike HLAIps had been

previously reported by MS (SIIAYTMSL[s0691-0699])[44]. In addition, four of the peptides were not reported in the IEDB (AIHVSGTNGTK[s0067-0077], YGVSPTKL[s0380-0387], RVYSTGSNVFQTR[s0634-0646], NRALTGIAV[s0764-0772]). Interestingly, some spike HLAIps were predicted to bind to the HLA alleles of both cell lines, but only SIIAYTMSL[s0691-0699] was identified in both cell JY and Raji (Fig. 8d, orange scale). Once more, this highlights the need for direct validation of in silico-predicted HLA class I binders.

In summary, we report 16 spike peptides identified with high stringency and confidence, which are predicted to bind HLA class I in two cell lines expressing different HLA alleles. Accordingly, this analysis confirms the HLA presentation of many peptides previously reported to be capable of eliciting a T-cell response but also provides more potential spike HLAIps. While the presentation and immunogenicity of the peptides in a clinical setting remains to be validated, this shows the capabilities of Thunder-DDA-PASEF for identifying potential HLA class I-restricted immunogenic targets.

## Discussion

Here, we present Thunder-DDA-PASEF, an optimized LC-MS method for HLAIp immunopeptidomics compatible with 47 min gradients. We took advantage of the IMS separation in the timsTOF Pro-2 MS to semi-selectively fragment singly or multiply charged ions with the expected IMS vs m/z distribution for HLA class I peptide ligands (8-13 amino acids long). Then, we further optimized the resulting HLAIp-tailored method for enhanced TIMS resolution and MS sensitivity. Adapting the DDA-PASEF settings contributed the most to the improvements in peptide coverage (64.2–120%), and the HLAIp-tailored Thunder isolation polygon provided a further increase of 5–15%. Altogether, compared to the 100 ms method without polygon this doubled the identification of HLAIps across samples with diverse HLA alleles. In addition, rescoring the results with our MS²PIP timsTOF fragmentation prediction model further improved HLAIp identification by 33 to 41.7% and improved data completeness by 10%. Since fragmentation prediction was similarly accurate for non-tryptic and tryptic peptides, we expect similar improvements for proteomics experiments, but this was out of the scope of this project. Collectively, Thunder with PEAKS +MS²R resulted in $5738 \pm 525$ HLAIps from only one million JY cell equivalents and reached up to $14{,}516 \pm 272$ from 20 million.

Compared to other immunopeptidomics methods in Orbitrap instruments, Thunder-DDA-PASEF in the timsTOF Pro-2 provided the highest number of JY HLAIps (Fig. 5), followed by Demmers, which showed complementarity, probably due to the EThcD fragmentation[41]. However, Demmers et al. identified 10,039 HLAIps by analyzing three injections of 167 million cell equivalents each, using a 90 min gradient. In contrast, our method resulted in 12,639 HLAIps from three injections of only 10 million cell equivalents each, using a 47-min gradient. Since biological and sample preparation differences may also influence these results, we also analyzed the same sample in an Exploris 480 based on methods published by Klaeger et al.[17]. The method using extensive FAIMS-assisted gas phase fractionation (GPF) including singly charged peptides required eight injections (FAIMS 8CVs) but only identified 6889 HLAIps. In comparison, Thunder-DDA-PASEF provided twice the number of peptides in only three injections. Although FAIMS GPF can also be programmed to screen more than one CV per run, FAIMS 3CVs yielded even fewer peptides than the analysis without FAIMS (2511 vs. 3755 HLAIps, respectively). Moreover, unlike FAIMS GPF, where optimal CVs must be selected for each HLA allele, Thunder-DDA-PASEF can readily screen a large diversity of HLAIp types without further adaptation (Fig. 4). Therefore, Thunder-DDA-PASEF outperforms the methods tested regarding immunopeptidome coverage, analysis time, sample input requirements, and ease of deployment.

The importance of including singly charged peptides in HLAIp immunopeptidomics experiments has been previously proven and is becoming a common practice in the field[17,32–35], but it requires instrument-specific adaptations that are worth highlighting. An example using timsTOF instruments for immunopeptidomics was published by Feola et al.[39]. However, the method employed by the authors included the standard isolation polygon and restricted the TIMS range to 0.6–1.3 $1/K_0$, effectively excluding the singly charged ions since they are eluted at > 1.3 $1/K_0$ (Supplementary Fig. S1). As a result, the proportion of singly charged peptides detected in JY samples was below 1%, while we detected up to 46% JY HLAIps (Fig. 7c). Our study shows the importance of extending the TIMS range and adapting the isolation polygon to cover the large population of singly charged HLAIps. Removing the polygon and allowing the fragmentation of singly charged peptides already enables the identification of singly charged HLAIps (Fig. 2). Nevertheless, using a tailored isolation polygon improves duty cycle and peptide identification, as observed using the HLAIp-tailored Thunder polygon in combination with the 300 ms method (Fig. 4a, c).

Additional adjustments could further improve the identification of immunopeptides. Considering the high coverage obtained from only one million cell equivalents (4049 HLAIps from JY), the sample preparation method could be down-scaled to reduce sample requirements and transferred to a high-throughput format to take advantage of the short analysis time required (1 h), for instance, using microfluidics systems[39,51] or plate-based protocols[52]. Here, we aimed for a method compatible with a wide range of HLA alleles but it could be refined for exceptional cases where a specific HLA type with a distinct IMS vs. m/z profile is analyzed. For instance, allele-specific polygons could be created using the profiles shown in Supplementary data (Figs. S8 and S11d) or from a preliminary analysis of their sample of interest without an isolation polygon. In addition, adaptations may be needed to screen modified HLAIps. For instance, phosphopeptides have lower CCS values and thus higher $1/K_0$ than their unmodified counterparts[53]. Until now, few studies have analyzed modified immunopeptides such as phospho-immunopeptides due to the large amounts of sample required for the double enrichment (up to $1 \times 10^9$ cells)[51]. However, this may become possible with improved methods such as Thunder-DDA-PASEF. Although our study is limited by not addressing these points here, they could be the subject of future research by the growing international immunopeptidomics community.

The onset of the ongoing SARS-CoV-2 pandemic has fueled the discovery of antigen candidates for vaccination, employing in silico prediction algorithms, genetic screens, or peptide library T-cell response assays. Even though immunogenicity testing of hypothesized vaccine candidates yielded some positive outcomes (reviewed in refs. 20–22), direct evidence of HLA peptide presentation relies mainly on direct identification by LC-MS.

Here, we provide a list of 16 spike HLAIps identified with high confidence (FDR < 0.6%) in HLAIp-enriched samples, validated against synthetic peptides (SA >=0.8 and iRT ratio >= 99.9%) and predicted to bind at least one of the 7 HLA alleles in JY or Raji (Fig. 8d). Since we focused on optimizing the MS method and evaluating its performance, a limitation of our study is that we did not study the antigen presentation on SARS-CoV-2 infected cells or patient samples. However, the approach transfecting cells expressing high levels of HLA combined with the high coverage of Thunder-DDA-PASEF resulted in the most extensive MS-based panel of SARS-CoV-2 spike HLAIps so far, since only ten peptides had been reported by MS before. In addition, seven of those peptides have been reported in the IEDB[50] with a positive response in B-cell or T-cell assays in 10 to 100 % of the human subjects tested, suggesting they could be immunogenic. Although these peptides remain to be further evaluated for their diagnosis or therapeutic value, this shows that Thunder-DDA-PASEF can confirm predicted immunopeptides and also discover more potential targets.

The list of spike HLAIps detected in our study complements previous reports. For instance, Weingarten-Gabbay et al.[24] analyzed the HLA class I immunopeptidome of SARS-CoV-2-infected human HEK293T cells and lung A549 cells transfected to express the virus

entry factors ACE2 and TMPRSS2. After fractionating the HLAIp-enriched peptides by high pH RP, they analyzed the samples in an Orbitrap Exploris 480 with CV −50 and −70, and identified 36 SARS-CoV-2 HLAIps. They reported seven spike peptides, including four detected in HEK293T cells, which are predicted to bind HLA alleles also present in JY (HLA02:01 and B07:02). Three of those peptides (GLITLSYHL, GPMVLRGLIT, MLLGSMLYM) could not be detected in our transfectant system since they originate from variants in the 2019-nCoV/USA-WA1/2020 isolate (NCBI accession number: MN985325.1) used by Weingarten-Gabbay et al., which are not present in the plasmid we used (Wuhan-Hu-1 isolate, GenBank accession number QHD43416.1,[54,55]). The remaining peptide (NLNESLIDL) is shared between both sequences, but we could not detect it in JY. This could be due to differences in the antigen processing machinery or deviations between the transgenic and viral expression of the protein.

In addition, our results confirm SIIAYTMSL$^{s0691-0699}$ as a multiallelic spike antigen. This peptide was present in the data from Weingarten-Gabbay[24], but it was only identified when Xin et al. 2022 reanalyzed the data using a deep learning algorithm[44]. This peptide is particularly interesting since it is predicted to bind multiple HLA alleles in JY (HLA-A02:01, B07:02, and C:0702) and Raji (HLA-C03:04, C04:01), but also other HLA supertype representatives (HLA-A26:01, B08:01, HLA-B39:01). In addition, the IEDB reports an RF of 0.39 to 0.53[50]. Collectively, this information indicates that SIIAYTMSL$^{s0691-0699}$ is a multi-allelic HLAI-presented peptide and suggests that it could be immunogenic.

In summary, Thunder-DDA-PASEF using PEAKS+MS²Rescore enables an in-depth coverage of HLAIps, outperforming state-of-the-art methods for antigen discovery and direct validation of immuno-peptides hypothesized by non-MS methods. This opens opportunities to dig deeper into the immunopeptidome in our search to discover specific antigens to target infectious diseases and cancer.

## Methods
### Ethics statement
We confirm that this study complied with all relevant ethical regulations regarding the use of human study participants and was conducted according to the criteria set by the Declaration of Helsinki. Human plasma was purchased from Zen-Bio (USA, SER-SPL). According to the provider, Human Source Plasma was collected via plasmapheresis on a Fenwal Aurora from a consented adult volunteer donor in the United States. The donor signed a US Food and Drug Administration (FDA) validated donor consent form. The consent form explicitly listed the intended use for non-clinical research and waived any rights generated from these research and commercial applications. Institutional Review Board (IRB) approval was obtained from Pearl Pathways, LLC. Additionally, the study adhered to guidelines for the responsible conduct of research and the protection of human subjects.

### Materials and substances
Analytical grade substances and reagents were used for sample preparation, and were acquired from Sigma Aldrich (Merck), unless otherwise stated. Water, acetonitrile, and formic acid were LC-MS grade products acquired from Carl Roth. LoBind tubes (Eppendorf) were employed to minimize sample loss.

### Cell culture and human plasma sample
Both suspension and adherent cell cultures were used to increase analyzed samples containing different HLA alleles. HeLa (CVCL_0030) whole-cell pellets were purchased from Ipracell (Belgium, CC-01-10-50). Human plasma was purchased from Zen-Bio (USA, SER-SPL, fresh frozen, in 4% sodium citrate (m/v)). JY (CVCL_0108) and SK-MEL-37 (CVCL_3878) cells were cultured in-house and both, suspension and adherent cell cultures, were harvested at 220 x g for 10 min and washed three times with 1x PBS prior counting and freezing at −80 °C until

further use. However, culture conditions and preparation for harvest were slightly different.

The human B lymphoblastoid cell line JY was purchased from ATCC and the human Burkitt lymphoma cell line Raji was obtained from the DSMZ-German Collection of Microorganisms and Cell Cultures. Both cell lines were maintained in RPMI1640 medium supplemented with 10 % FCS (Gibco (v/v)), 2 mM glutamine, 1 mM sodium pyruvate, 100 units/ml penicillin, and 100 µg/ml streptomycin.

The human melanoma cell line SK-MEL-37 was purchased from sigma-aldrich (SCC262). SK-MEL-37 cells were maintained in DMEM medium supplemented with 10% FCS (Gibco (v/v)), 2 mM glutamine, 1 mM sodium pyruvate, 100 units/ml penicillin, and 100 µg/ml streptomycin. For passaging and prior harvest, cells were dis-attached from the culture flasks incubating 10 min with 10 mM EDTA in PBS under gentle rocking.

HL60 cells (CCL-240) purchased from the American Type Culture Collection (ATCC) were cultured in-house in RPMI medium supplemented by 10% FBS (v/v) and 5% penicillin/ streptomycin (v/v). The cell suspension was harvested at 1500 rpm at 4 °C for 5 min prior counting and freezing at -80˚C overnight and -150˚C until further use.

### Cell transfection
The pcDNA3.1-SARS2-spike vector containing the full-length cDNA encoding for the SARS-CoV2 spike protein (GenBank accession number QHD43416.1) was obtained from Fang Li (Addgene plasmid #145032 ; https://www.addgene.org/145032/)[55]. The spike S cDNA was split into S1 (2016 bp) and S2 (1761 bp) subunits for cloning by PCR into the NheI and XhoI restriction sites from the multiple cloning site of the pcDNA3.1+P2AeGFP vector (Genscript). The following oligonucleotides (all purchased by Sigma) were used : GCAT GCT AGC ATG TCT CAG TGC GTG AAC CTG ACT ACT AGA ACC and GCAT CTC GAG ACG GCG AGC CCT CCT TGG GGA GTT GGT CTG GGT CTG for the S1 cDNA and GCAT GCT AGC ATG AGC GTG GCC AGC CAG TCC ATC ATC GCC TAC and GCAT CTC GAG AGC GGG AGC GAC CTG GGA TGT CTC GGT GGA G for the S2 cDNA cloning. To generate stable JY and Raji trans-fectants expressing either the S1 or the S2 protein fragments (Supplementary Material S1, Material and Methods), 2 million cells were exposed to 230 V and 500 µF in the presence of 10 µg plasmid DNA using the Bio-Rad Gene Pulser II. After electroporation, cells were cultured 24 h before starting G418 (Gibco) selection at a concentration of 400 µg/ml for JY cells and 800 µg/ml for Raji cells. G418-resistant and eGFP-expressing cells were selected by three rounds of screening using a FACS Aria (BD Biosciences) at the Core Facility of the Research Center for Immunotherapy (University Medical Center, Johannes Gutenberg University Mainz).

### Immuno-affinity purification of HLA peptide ligands
Sample preparation in Tenzer lab. HLA class I ligands were enriched by immunoprecipitation as described by[56] with modifications[57]. Briefly, in-house cultured JY, Raji or SK-MEL-37 cells were washed three times with PBS, harvested, flash-frozen, and stored at −80 °C until further preparation. For the HeLa pellet and Plasma commercial sources were used. For method optimization and SARS-CoV-2 spike HLAIp analysis 500 x 10⁶ cells, for analysis of diverse allelic cell lines and plasma 100x10⁶ and 5 mL were used, respectively. The cell pellets were thawed and lysed in a non-denaturant buffer (1% CHAPS in PBS (m/v)) aided by sonication. Immunoprecipitation was performed using the W6/32 antibody immobilized on CNBr-activated beads (Cytivia). The anti-panHLA Class I monoclonal antibody W6/32 (anti-HLA-A, -B, -C) was purchased from Hoelzel-biotech, and produced by Leinco Technologies (ref. H263). After overnight incubation, the beads were washed once with PBS and once with water before peptide ligands were eluted under acidic conditions (0.2% TFA (v/v)). Next, peptides were ultra-filtered using 10 kDa molecular weight cutoff (MWCO) filters (Vivacon 500, 10,000 MWCO Hydrosart) and then desalted by SPE on a

Hydrophilic-Lipophilic-Balanced sorbent (Oasis HLB 96-well µElution Plate, 2 mg Sorbent per Well, 30 µm, Waters Corp.), applying 35% ACN (v/v), 0.1% TFA (v/v) for elution. Finally, dried peptides were dissolved in 15 µL of water with 0.1% FA (v/v) for subsequent LC-MS/MS analyses.

Sample preparation in Carapito lab (HL60 samples). HLA class I ligands were enriched by immunoprecipitation as previously described[16]. The HB-95 hybridoma producing anti-panHLA Class I antibody W6/32 was purchased from ATCC and cultured in Panserin 401 serum-free medium (Pan Biotech). The purification of the antibody was done with the NGC Chromatography System (Biorad) using a HiTrap Protein G HP 1 mL column (Amersham Pharmacia). A total of $600 \times 10^6$ in house cultured HL60 cells were lysed with 20 mM Tris-HCl, 150 mM NaCl, 0.25% sodium deoxycholate (m/v), 1mM EDTA pH8, 0.2 mM iodoacetamide, 1 mM PMSF, Roche Complete Protease Inhibitor Cocktail, 0.5% NP 40 (v/v), PBS, pH 7,4. The lysate was centrifuged at $21,000 \times g$ for 30 min at 4°C. HLA-peptide-complexes were captured on CNBr-activated sepharose 4B beads (Cytivia) linked to W6/32 antibody. Following binding, beads were washed several times with 3 buffers (150 mM NaCl, 20 mM Tris-HCl, pH7.4; 400 mM NaCl, 20 mM Tris-HCl, pH 7.4; and 20 mM TrisHCl, pH 8.0) and bound complexes were eluted in 0.1M acetic acid. Eluted HLA peptides and the subunits of the HLA complexes were desalted using a C18 Macro Spin column (Harvard Apparatus) according to the manufacturer's protocol. Finally, HLA peptides were purified from the HLA-I complex after the elution with 25% ACN (v/v), 0.1% TFA (v/v). Samples were evaporated under vacuum and resuspended in water with 0.1% FA (v/v).

## Proteomics sample preparation

Whole cell lysates were also used to train the timsTOF MS$^2$PIP model. For protein extraction, HeLa cells were lysed using a urea-based lysis buffer (7 M urea, 2 M thiourea, 5 mM dithiothreitol (DTT), 2% CHAPS (m/v)). JY cell pellets were thawed and lysed in 1% CHAPS in PBS (m/v) as mentioned for the HLA peptide enrichment. Lysis was further promoted by sonication at 4°C for 15 min (30 s on/30 s off) using a Bioruptor device (Diagenode, Liège, Belgium). After cell lysis, protein concentration was determined using the Pierce 660 nm (for HeLa) or Pierce BCA protein assays (for JY, due to the CHAPS) according to the manufacturer's protocols (Thermo Fisher Scientific).

Protein digestion for whole-proteome samples. HeLa, and JY, were processed using filter-aided sample preparation (FASP) as detailed before (refs. 58, 59). In brief, lysates were loaded onto spin filter columns (Nanosep centrifugal devices with Omega membrane, 30 kDa MWCO; Pall, Port Washington, NY) and washed three times with buffer containing 8 M urea. Afterward, proteins were reduced and alkylated using DTT and iodoacetamide (IAA), respectively. After alkylation, excess IAA was quenched by the addition of DTT. Then, the buffer was exchanged by washing the membrane three times with 50 mM NH$_4$HCO$_3$ (pH 7.8) for trypsin, or TRIS Base (pH 8.5) for elastase. The proteins were digested overnight at 37°C using trypsin (Trypsin Gold, Promega, Madison, WI) or elastase (Elastase, Promega, Madison, WI) at an enzyme-to-protein ratio of 1:50 (m/m). After proteolytic digestion, peptides were recovered by centrifugation and two additional washes with 50 mM NH4HCO3. After combining peptide flow-throughs, samples were acidified with trifluoroacetic acid (TFA) to a final concentration of 1% trifluoroacetic acid (TFA (v/v)) and lyophilized. Lyophilized peptides were reconstituted in water with 0.1% formic acid (FA (v/v)) for LC-MS analysis.

## LC-MS/MS timsTOF Pro-2 (Tenzer lab)

NanoLC-MS analysis was performed using a nanoElute coupled to a timsTOF-Pro-2 mass spectrometer. Data was acquired using Compass Hystar (Bruker) versions between 4 and 5.1, and timsControl versions between 3 and 4.0.5 (Bruker). The desalted peptides were directly injected in a C18 Reversed-phase (RP) Aurora 25 cm analytical column (25 cm × 75 µm ID, 120 Å pore size, 1.7 µm particle size, IonOptics,

Australia) and separated using gradients increasing the proportion of phase B (ACN with 0.1% FA (v/v)) to phase A (water with 0.1% FA (v/v)). A 47 min gradient was used for the method development analyses, and a 110 min gradient for the analysis of spike-transfected cells and the synthetic peptides, as detailed in Supplementary Data S1. A Captive Spray source was used for ionization, with a capillary voltage of 1600 V, dry gas at 3.0 L/min, dry temperature at 180 °C, and TIMS-in pressure at 2.7 mBar. MS data were acquired in DDA-PASEF mode. Different MS parameters were evaluated during method development, as detailed in Supplementary Data S1. The JY and Raji spike-transfected data set was acquired using the optimized conditions described in the following lines. HLAIp IP-enriched, ultrafiltered, and desalted peptides were analyzed in three injection replicates each, using a volume of 1.5 µL/injection, equivalent to 33 million cells from the original sample. Peptides were separated in a 110 min. gradient from 2% to 37% of ACN with 0.1% FA (v/v). The MS was configured with the optimized Thunder-DDA-PASEF method, employing an HLAIp-tailored isolation polygon (Fig. 2), a 300 ms TIMS ramp, three MS2 frames/cycle, one cycle overlap, using the high-sensitivity mode (optimized detector voltages). The settings used for LC-MS are detailed in Supplementary Data S1 and the timsTOF Pro method is included as Supplementary Data S2.

## LC-MS/MS timsTOF Pro, for collision energy tests (Carapito lab)

Proteomics on Hela tryptic digests. Hela Pierce protein digest (Thermo Fisher Scientific, Rockford, US) was diluted in an aqueous solvent with 2% ACN (v/v) and 0.1% FA (v/v) to 100 ng/µL and 2 µL were injected for each run. A nanoElute LC system (Bruker, Billerica, US) coupled to a timsTOF Pro mass spectrometer (Bruker Daltonics) was used for DDA-PASEF data acquisition. Peptide extracts were separated using an Aurora 25 cm packed emitter column (Ionopticks) with a linear gradient of 2–37% B (2% ACN with 0.1% FA (v/v)) over 100 min at a flow rate of 0.2 nL/min. The dual TIMS configuration utilized a ramp time and accumulation time of 100 ms, resulting in a total cycle time of 1.17s. DDA-PASEF mode with 10 PASEF scans covered a mass range from 100 m/z to 1700 m/z with charge states set from 0 to 5+. The capillary voltage was set at 1500 V. Different ion mobility ranges, and corresponding collision energy ramps were used. Four different collision energy ramps were used for peptide fragmentation: linear ramps from 52-20, 59-20, and 32–60 eV were used with an ion mobility range from $1.6 \, to \, 0.6 \, Vs/cm^2$, while the fourth ramp used a linear collision energy ramp from 59-20 eV and an ion mobility range from 1.3 to 0.85 Vs/cm$^2$.

Immunopeptidomics on HL60 samples. A nanoElute LC system (Bruker, Billerica, US) coupled to a timsTOF Pro mass spectrometer (Bruker Daltonics) was used for DDA-PASEF data acquisition. Data was acquired using Compass Hystar (Bruker) versions between 4 and 5.1, and timsControl versions between 3 and 4.0.5 (Bruker). The peptide extract equivalent to $60 \times 10^6$ cells was injected and separated using an Aurora 25 cm packed emitter column (Ionopticks) with a gradient of 2–37% B (2% ACN with 0.1% FA (v/v)) over 100 min at a flow rate of 0.2 nL/min. The dual TIMS configuration utilized a ramp time and different accumulation times (100, 166, or 200 ms). DDA-PASEF mode with either 6 or 10 PASEF scans covered a mass range from 100 m/z to 1700 m/z with charge states set from 0 to 5+. The capillary voltage was set at 1500 V. Four ion mobility ranges (1.6−0.6, 1.7−0.6, 1.75−0.65 and 1.7−0.65 Vs/cm$^2$), two collision energy ramps (59−20 and 55−10 eV) and threshold intensity (2500 and 1500) were used.

## LC-MS/MS Orbitrap Exploris 480

An aliquot of the same sample of JY HLAIps analyzed in the nanoElute - timsTOF Pro-2 was also analyzed in an Ultimate 3000 LC coupled to an Orbitrap Exploris 480 (Thermo Fisher Scientific). Peptides were separated on an Ultimate 3000 LC system at 300 nL/min using a Waters Analytical HSS-T3 column (75 µm × 25 cm, 1.8 µm particle size) and a linear gradient from 4-35% Solvent B (90% ACN with 0.1% FA (v/v)) over

44 min, from 35–90% under 1 min and flushed at 90% with 400 nL/min for 5 min. Data was acquired without FAIMS (wo-FAIMS) or using FAIMS-based gas phase fractionation (GPF) accordingly to methods previously introduced by Klaeger et al.[17] and described in the paragraphs below. For all the experiments, MS1 spectra were collected over a scan range of 350–1700 m/z at a resolution of 60,000, monoisotopic peak detection was set to peptide, and relax restrictions were enabled. For data-dependent MS/MS, precursors were isolated using a 1.1 m/z window and collected until either a normalized AGC target of 50% or a maximum injection time of 100 ms (CV −10 and −20) or 120 ms (CV −30 to −80) was reached. Fragmentation was achieved by applying 30% HCD collision energy and recorded at 15,000 resolution.

For the FAIMS-8CVs experiment, the CV-based peptide screening was performed in individual injections from CV −10 to −80 V, applying steps of −10 V, keeping the CV constant during each acquisition (hence, in 8 injections). FAIMS was set to standard resolution in all the methods. For CV −10 and −20 V, MS1 spectra were collected for precursors, including charge states +1 to +4 until either 100% normalized AGC target or 50 ms maximum injection time was reached (scan-based method: 10 scans/duty cycle). The intensity threshold was set to $1 \times 10^3$ and the dynamic exclusion time to 10 s. From CV −30 to −80 V, MS1 spectra were collected on precursors with charge states +2 to +4 until either 100% normalized AGC target or 25 ms maximum injection time was reached (cycle time-based method: 1.5 s cycle time/duty cycle). A precursor fit filter was set to a 50% fit threshold and 1.4 m/z fit window with an intensity threshold of $1x10^4$, with a dynamic exclusion time of 10 s.

For the FAIMS-3CVs experiment, the FAIMS was switched between three compensation voltages (CVs, −20, −50, −70) within each acquisition. MS Settings detailed above for CV −20, −50, and −70 were added as individual experiments in a single method. All other settings were as described above. The analysis of samples without FAIMS included the same MS settings as described for CV −10 to −20 V but removing FAIMS from the method.

## External datasets

The following publicly available JY HLAI-ligandome data sets were downloaded for comparison against Thunder-DDA-PASEF. Feola et al.[39], timsTOF Pro raw data was downloaded (PRIDE ID: https://proteomecentral.proteomexchange.org/cgi/GetDataset?ID= PXD000394PXD000394) and processed with PEAKS XPro using the same parameters as in our experiments (see following section). For Pak et al.[33] Q Exactive HF-X data, the list of peptides identified in JY cells by the authors using the "BigLib" was used. Demmers et al.[41], raw Orbitrap Fusion Lumos Tribrid data of HLA class I ligandome analyses of JY cells cultivated at 37 °C was downloaded (ProteomeXchange ID: https://proteomecentral.proteomexchange.org/cgi/GetDataset?ID=PXD022930PXD022930) and processed with PEAKS XPro with the settings described in the following section.

## Peptidomics database search and rescoring

Data analysis was performed in PEAKS XPro (v10.6, build 20201221). A complete description of the settings used is included in Supplementary Data S1, tab PEAKS_XPro. The protein database was composed of the UniProtKB (Swiss-Prot) reference proteomes of *Homo sapiens* (Taxon ID 9606, downloaded 02/Feb/2020), Epstein-Barr virus (strain GD1, Taxon ID 10376, downloaded 06/Feb./2022), GFP from *Aequorea victoria* (P42212), and SARS-CoV-2 (Taxon ID 2697049, downloaded 10/March/2021), as well as the SiORF1 reported by[60,61], supplemented with a list of 172 possible contaminants. For database searches, protein in silico digestion was configured to unspecific cleavage and no enzyme. Up to two variable modifications per peptide were allowed, including Methionine oxidation, cysteine cysteinylation, and Protein N-terminal acetylation. The settings for timsTOF and Orbitrap data are summarized in the paragraphs below, and further detailed in Supplementary Data S1.

To avoid introducing a bias due to the transfer of peptide identifications, files were processed in independent projects for each sample type (cell line or plasma), LC-MS method, or cell equivalents injected.

For Orbitrap data, Raw LC-MS files were loaded with the configuration as Orbi-Orbi with CID fragmentation for Exploris 480 data, or Orbi-Trap with EThcD for Lumos data (Demmers dataset), both as DDA. Mass tolerance was set to 5 ppm for MS1 and 0.02 Da for MS2.

For timsTOF data, the option timstof_feature_min_charge (in file PEAKSStudioXpro\algorithmpara\feature_detection_para.properties) was set to 1 to allow the identification of singly-charged features. Raw LC-MS files were loaded with the configuration for timsTOF DDA-PASEF data with CID fragmentation. Mass tolerance was set to 15 ppm for MS1 and 0.03 Da for MS2. Identification rescoring. Spectra were exported in MGF format and identifications in mzIdentML format, including decoys and without any score filter ($-10lgP \geq 0$ for peptides and proteins). Identifications were then rescored using an updated version of MS²Rescore[16,29] (v3.0.0b4) with the timsTOF model described in the next subsection. The Spectrum ID regex pattern was set as "(index=[0-9]*)", the MS2 mass accuracy tolerance was 0.03 Da, DeepLC was used for retention time prediction[46], and the ionmob GRU predictor[47] was used for CCS prediction.

For downstream analyses, peptide identifications and the respective protein coverage were evaluated using the -peptide.csv-report exported from PEAKS at peptide $FDR \leq 0.01$, or the combination of this file with the output of MS²Rescore filtered at $mokapotqvalue \leq 0.01$ In these files, only one charge state is reported for each peptide. Thus, the information regarding peptides identified at multiple charge states was extracted from the PSM results filtered at similar thresholds. A threshold of protein $-10lgP \geq 20$ was used, but this does not affect the peptide or PSM reports. Although PEAKS can provide de novo peptide candidates, only peptides identified by database search were reported across all the experiments.

## timsTOF peptide fragmentation prediction model

MS²Rescore integrates the machine learning prediction of retention and fragmentation peak intensity using DeepLC[46] and MS²PIP[28,62], respectively, with the semi-supervised machine learning-based FDR calculation of Mokapot[63]. As peak intensity predictions are heavily reliant on the fragmentation method, instrument, or labeling method, the predictions of the Orbitrap HCD model of MS²PIP[45] do not perform very well on timsTOF data. Therefore, we trained a timsTOF-specific peak intensity model using in-house timsTOF data from two different labs. This included digested peptides of JY (trypsin and elastase) and HeLa (trypsin), and HLA class I immunoprecipitation-enriched peptides of JY, HeLa, SK-MEL-37 (all previous from Tenzer lab), and HeLa (trypsin) and HL60 samples using multiple CE settings(Carapito lab).

To train the models, all identifications from both labs and with different collision energy settings were combined whereafter for each unique peptidoform in terms of sequence, charge, and modification the highest scoring PSM was retained in the training data. All the remaining 251,149 peptides (including modifications) were converted to an MS²PIP feature vector and were used to train XGBoost(v1.6.2)(Chen & Guestrin, n.d.) prediction models for both B and Y ions separately. Before training, 10,045 peptides (including modifications) were set aside to use as evaluation set for the final model, ensuring that the datasets were represented homogeneously. For the optimization of the hyperparameters, the Hyperopt (v0.2.7)(Hyperopt Documentation, n.d.) python package was used and combined with 6-fold cross-validation. Boosting rounds were set to 300 max with an early stopping of 10 rounds. All the hyperoptimization rounds were logged to weights and biases (Supplementary Data S9, interactive version available in https://wandb.ai/arthur_declercq/Final%20timstof%20model%20training/reports/MS-PIP-timsTOF-prediction-models--Vmlldzo1NjQxNTMw). The optimal hyperparameters were then used to train the final model on all the training data with the evaluation set as evaluation set. To test the overall

performance of the trained models, a different dataset was used. As, in immunopeptidomics, similar peptidoforms may occur for the same HLA alleles, we performed an extra step to remove all peptidoforms from the test set, even though these were different runs. The test set also contained runs with distinct collision energies for evaluation and were compared in performance to the 2021 Orbitrap HCD immunopeptidomics MS²PIP model.

## Experiment design

For method development, pooled samples of IP-enriched HLAIps from wild type cells or human plasma (as described above) were used. For the final JY and Raji data set, the IP protocol was used to enrich the HLAIps from three cultures of each WT cell line (JY_WT, and Raji_WT) and two different cultures of each transfected cell line (JY_S1, JY_S2, Raji_S1, and Raji_S2). In every experiment, each sample was analyzed in three LC-MS injection replicates.

## Data analysis and statistics

HLA alleles and binding prediction was performed using NetMHCpan-4.1[30] and sequence clustering with GibbsCluster-2.0[64], both via MhcVizPipe (v0.7.9)[31]. Since the human plasma sample was not HLA-typed, we previously obtained the possible matching alleles across twelve supertype representatives using MHCMotifDecon (ref. 40), and selected the top 4 for the binding prediction as mentioned above. The allotypes used for prediction were as follows: JY (HLA-A0201, B0702, C0702), HeLa (HLA-A68:02, A03:19, B15:03, C12:03), SK-MEL-37 (HLA-A0201, A1101, B1501, B5601, C0102), and commercial human plasma (predicted as HLA-B0702, B1501, A0101, A2601).

Data analysis, statistical analysis and visualization were done mostly using in house R scripts[65]. The main R packages used were as follows; the statistical difference was assessed by two-sided *t*-test using *ggpubr* (v. 0.4.0)[66]; plots were generated using *ggplot2* (v. 3.4.0)[67]; Venn plots with *ggvenn* (v. 0.1.9)[68]; upset plots with *ggupset* (v. 0.3.0)[69]; and sequence logos were generated using ggseqlogo[70].

Gene ontology (GO) enrichment analysis was performed using GOrilla[48]. Spectral mirrored plots and calculations were obtained from the Universal Spectrum Viewer (USE)[49] by comparing the spike peptide spectra acquired from transfected cells against spectra obtained from synthetic peptides based on the similarity Spectral Angle (SA). Spectra and respective transition lists were neither pre-filtered (e.g., selecting transitions) nor modified prior USE submission.

## Reporting summary

Further information on research design is available in the Nature Portfolio Reporting Summary linked to this article.

## Data availability

The mass spectrometry immunopeptidomics and proteomics data have been deposited to the ProteomeXchange Consortium[71] via the jPOSTrepo partner repository[72] with the dataset identifiers https://proteomecentral.proteomexchange.org/cgi/GetDataset?ID=PXD040385 PXD040385 for ProteomeXchange and https://repository.jpostdb.org/entry/JPST002044 JPST002044 for jPOSTrepo. Data from JY immunopeptidomics used for training was previously published[47] and can be accessed with the dataset identifiers https://proteomecentral.proteomexchange.org/cgi/GetDataset?ID=PXD043026 PXD043026 for ProteomeXchange and https://repository.jpostdb.org/entry/JPST002158 JPST002158 for jPOSTrepo. Data from Carapito lab have been deposited to the ProteomeXchange repositories https://proteomecentral.proteomexchange.org/cgi/GetDataset?ID=PXD046535 PXD046535 for HL60 immunopeptidomics, and https://proteomecentral.proteomexchange.org/cgi/GetDataset?ID=PXD046543 PXD046543 for HeLa tryptic proteomics files. A list of datasets, main LC-MS parameters, and their repository locations is included as Supplementary Data S10. Protein sequences

were downloaded from UniProtKB (Swiss-Prot), as detailed in the Methods subsection Peptidomics database search and rescoring, and the FASTA databases are also included in the corresponding repositories.

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

## Acknowledgements

We would like to acknowledge Lucas Kleinort (HI-TRON, Mainz, Germany) for his technical assistance and contributions to sample preparation, Kristina Marx (Bruker, Bremen, Germany) for the fruitful scientific discussions, Kevin Kovalchik (Immatics, Tuebingen, Germany) for his help with MhcVizPipe, and David Teschner (Institute of Computer Science, Mainz, Germany) for adapting ionmob to be implemented in MS²Rescore. We acknowledge the support of the flow cytometry core facility, the mass spectrometry core facility and the sequencing core facility of the Research Center for Immunotherapy (FZI) at the University Medical Center Mainz, and the High Throughput Immunopeptidomics Platform of HI-TRON Mainz. The graphical abstract and Figure 1 were designed in part using images from Servier Medical Art. Servier Medical Art by Servier is licensed under a Creative Commons Attribution 3.0 Unported License (https://creativecommons.org/licenses/by/3.0/). This work was funded by the Deutsche Forschungsgemeinschaft (DFG, German Research Foundation) SFB1292 TP13 (H.S.) and TPQ01 (S.T.), the Federal Ministry of Education and Research (BMBF) (MSCoreSys, DIASyM, Fkz: 161L0218, 16LW0241K to S.T.) the Helmholtz-Institute for Translational Oncology Mainz (HI-TRON Mainz) - a Helmholtz institute by DKFZ, Mainz, Germany. D.G.Z., E.K., and S.T., acknowledge funding from Bundesministerium für Bildung und Forschung, (BMBF) as part of the National Research Node -Mass spectrometry in Systems Medicine- (MSCoreSys) [031L0217A/B]. A.D., L.M., and R.G. acknowledge funding from the Research Foundation Flanders (FWO) [1SE3722, G010023N, G028821N, 12B7123N]. L.M. acknowledges funding from the European Union's Horizon 2020 Programme (H2020-INFRAIA-2018-1) [823839], support from the Horizon Europe Project BAXERNA 2.0 [101080544], and funding from the Ghent University Concerted Research Action [BOF21/GOA/033]. C.C. acknowledges funding from the Agence Nationale de la Recherche via the French Proteomics Infrastructure (ProFI FR2048, ANR-10-INBS-08-03) and the Interdisciplinary Thematic Institute IMS (Institut du Médicament Strasbourg), as part of the ITI 2021-2028 supported by IdEx Unistra (ANR-10-IDEX-0002) for the PhD fellowship of J.B.R.

## Author contributions

Conceptualization: D.G.Z., D.A.S., U.D., H.S., and S.T. Methodology: D.G.Z., D.A.S., A.D., R.G., H.S., and S.T. Software: D.G.Z., A.D., R.G., and M.K.L. Validation: D.G.Z., D.A.S., J.B., A.D., R.G., A.H., and J.B.R. Formal analysis: D.G.Z., D.A.S., J.B., A.D., and M.K.L. Investigation: D.G.Z., D.A.S., J.B., A.D., R.G., E.K., A.P., A.H., and J.B.R. Resources: D.A.S., C.C., L.M., H.S., and S.T. Data curation: D.G.Z., D.A.S., J.B., A.D., R.G., M.K.L., A.H., J.B.R., and U.D. Writing - original draft: D.G.Z., D.A.S., J.B., and A.D. Writing - review & editing: all the co-authors. Visualization: D.G.Z., A.D., and M.K.L. Supervision: D.G.Z., D.A.S., R.G., C.C., L.M., H.S., and S.T. Project administration: D.G.Z., D.A.S., C.C., L.M., U.D., H.S., and S.T. Funding acquisition: D.A.S., H.S., S.T.

## Funding

## Competing interests

The authors declare no competing interests.

## Additional information

[1]Institute of Immunology, University Medical Center of the Johannes-Gutenberg University, Mainz, Germany. [2]Helmholtz Institute for Translational Oncology Mainz (HI-TRON Mainz) - A Helmholtz Institute of the DKFZ, Mainz, Germany. [3]German Cancer Research Center (DKFZ) Heidelberg, Division 191, Heidelberg, Germany. [4]VIB-UGent Center for Medical Biotechnology, VIB, Ghent, Belgium. [5]Department of Biomolecular Medicine, Ghent University, Ghent, Belgium. [6]BioOrganic Mass Spectrometry Laboratory (LSMBO), IPHC UMR 7178, University of Strasbourg, CNRS, ProFI - FR2048, Strasbourg, France. [7]Research Center for Immunotherapy (FZI), University Medical Center of the Johannes-Gutenberg University, Mainz, Germany.
✉e-mail: david.gomez-zepeda@dkfz-heidelberg.de; tenzer@uni-mainz.de

