## [Peer Review File · Nature Communications]

Reviewers' Comments:

Reviewer #1:

Remarks to the Author:

Summary: In this paper the authors present an optimised method for immunopeptidomic analysis using a TIMSTOF mass spectrometer from Bruker. They take advantage of the ion mobility separation to focus the instrument on the sequencing of HLA peptides by expanding the proteomics setting to include singly charged species. It is widely accepted that in immunopeptidomics sequencing singly charged species adds value to the subsequent peptide ID rate. They report that including these species enhances the identification rate in immunopeptidomics and that this can be done using ion mobility rather than precursor filtering.

My general comment:

The importance of inclusion of singly charged species in immunopeptidomics mass spectrometry experiments is not a novel finding and reported widely for many instrument types. However, the sole use of the ion mobility device to focus the sequencing depth in the region where these peptides are found is of interest to the community as these type of instruments become more common in labs carrying out immunopeptidomics. This method reiterates the importance of optimising your mass spectrometer for immunopeptidomics and not proteomics. It is overall very useful to the community but could benefit from some more rigorous testing and data analysis as suggested below. In my opinion the authors sometimes over-state the improvements, and cherry pick evidence/datasets to support there statements rather than rigorously testing/investigating the methods performance. In general I support the publication but want to see it fully road tested.

Section specific comments

2.2

It is not fully investigated by the authors how the ion mobility thunder windows alter the result from having no windows, and just expanding the ion mobility range from the standard proteomics workflow. The authors show there is no improvement here in total identifications, but that less MS2 are acquired and without the window you identify larger peptides, and these are from contaminants. I'd like to know more about the costs of using the polygon window approach. It is important to know the nature HLA peptides missed by including such a filter, are the filtered-out peptides predicted binders, are post-translationally modified forms missed because they have different CCS behaviour? Additionally, how do these results compare to simple charge-state screening without the windows as implemented by most mass spectrometers?

The authors should include some comparisons to the none-polygon protocol so that we can see its effects on HLA allotype, binding affinity and dynamic range.

Are certain alleles that carry more charge (2/3+) sacrificed in order to allow singly charged species does this lead to bias in samples that contain peptides with different anchor residues...? How does this compare when you analyse samples with different HLA types, the authors should validate their results across a broader range of samples with different HLA types and/or in a synthetic peptide library consisting of a mixture of different alleles.?

Section 2.3

All the results are confined to the supplementary I think they mention some important optimisation steps that could be included in the main figure here, especially considering that these changes appear to have a more of an affect on ID rates than the polygon shape changes on which the paper is based upon, however it seems that these adaptation are not novel and are mentioned in other previously published work is this why they have been put in the supplemental?

Section 2.4

The authors use MS2 rescore a machine learning algorithm to improve their identification rates, the method is not novel or specifically developed for the data generated by this instrument and their observations recapitulate what has been seen for this machine learning approach. Potentially, the authors could improve the approach for singly charged ions which seem to be less well rescored compared to multiply charged species. Do the authors have a synthetic library they could

acquire (or more samples with different HLA backgrounds) with their new method and provide this as a resource for training better algorithms? They could also use this to validate the sensitivity of their method and explore if any allelic bias occurs when they alter their methods.

Section 2.5

The authors make a series of comparisons between the data they generate here and other datasets. How they choose the sets is unclear they seem to consist of three datasets that use very different methods for different purposes.

For example the work by Feola develops a microfluidic device for high sensitivity immunopeptidomics and uses a lot less cell input, these factors alone limit the depth of their analysis and therefore the 720 % increase is not unexpected and doesn't really demonstrate the value of their new method... Whilst of use, I believe the comparison shows that including singly charged species by TIMS is complementary to other approaches. The most appropriate comparison is with Demmers et al., 2021.. Where the inclusion of singly charged ions is off.

Section 2.6

Can the authors comment on the overall reproducibility in this experiment?

Was the binding affinity/netMHC the same as in the method development section, how reproducible was this?

Here they choose two cell lines with distinct HLA types it would be good to see how the polygon on/off might affect the overall detection of the different peptides (see my comments re section 2.2).

Line 249: Why compare it to Pak et al., whilst the more fair comparison would be to Demmers et al.,? I am not sure how this demonstrates the capacity of the method. I am sure it does have the capacity, but can it be better demonstrated?

Did you expect the same peptides when they have different alleles?

Can the authors elaborate on how this analysis represents a resource for further exploitation, do they identify peptides that could be targets for immunotherapy, for example with the new method is the depth of coverage of potential/known tumour antigens greater than before?

Section 2.7

Can the authors justify how peptides presented by B cells are clinically relevant to SARS-CoV2 infection? The transfection is meant to mimic infection, are B cells relevant could you not of transfected Epithelial cell lines?

Can you please put these peptides in context with the other studies, was the lack of overlap with other studies due to overall depth or HLA type?

Line 293: this sentence doesn't make sense to me perhaps elaborate.

Section 3 Discussion.

Line 311: In my understanding Thunder is a sound not a shape.

I think it is misleading, they should rephrase the discussion to highlight that standard proteomics methods miss an important part of the immunopeptidome, that ionises in the singly charged CCS space. They have used TIMS filtering and reshaped the proteomics polygon to account for this... etc..

Line 331: how can you achieve targeting 9-12 mers when you don't know the charge state, please explain or omit.

Reviewer #2:

Remarks to the Author:

The manuscript "Thunder-DDA-PASEF enables high-coverage immunopeptidomics and identifies HLA class-I presented SarsCoV-2 spike protein epitopes" by David Gomez-Zepeda et al. reported new immunopeptidomic methodology by tims-TOF system. In this manuscript, the authors optimized default tims-TOF DDA workflow for immunopeptidomics as "Thunder-DDA-PASEF" by

fitting the IM dimension for class I immunopeptides. Taking information of retention time and MS2 peak intensity predictions into identification process by combining the MS2Rescore algorithm, the immunopeptidome coverage was further improved. The authors focused on singly-charged immunopeptide that have not been fully analyzed by previous immunopeptidomics methodologies, and by thunder-DDA-PASEF, the authors were successful to largely expand the identification of singly-charged immunopeptides. In the end, the authors used this thunder-DDA-PASEF to identify the viral protein-derived immunopeptides by using artificial models of viral spike protein expressing JY and Raji cells to show the potential of this methodology. In a situation where new tools for analyzing large proteomic data, including immunopeptidomics, are being reported one after another, further validation and explanation regarding the plausibility of peptide identification is necessary to accept the authors' assertion that Thunder-DDA-PASEF contributes to the depth of immunopeptidomics and is particularly advantageous for the identification of singly charged immunopeptides.

Major Comments:

1). About the database search by PEAKS XPro, the authors described in the main text that they filtered the proteins by $-10\lg P \geq 20$ (page 14, line 428). While they didn't mention about the setting of the "number" of unique peptides for protein identification. Did authors also set the number of unique peptide (0, 1, 2, 3..., etc) as " $-10\lg P \geq 20$ and (number) \geq of unique peptides" for protein identification by PEAKS XPro? If not, the default setting by the software should have applied "0" for the number of unique peptides. Under this condition, the protein list by PEAKS XPro may include the proteins identified by using only "de novo only" peptides. Therefore, if the authors claim that they performed "database search", that requires the peptide spectrum match against the fasta sequence in database. If the search setting of protein filter was " $-10\lg P \geq 20$ and '1' \geq of unique peptides", my concern is no longer necessary. While the authors mentioned about the protein identification in Figure 5d, it's better to confirm this point. Please check the list of proteins used in this manuscript were really "database-searched" proteins. And then clarify the setting for the protein filter and describe it in main text.

Further, while de novo sequencing by PEAKS software has an advantage in exploratory identification for peptides, it has also been reported that the actual false identification, even under the same FDR setting, is higher in PEAKS by comparing the other search engines. (Assaf Kacen et al., Nature Biotechnology, <https://doi.org/10.1038/s41587-022-01464-2>, Supplementary Figure 1). Though I will mention about my concerns about the validation for identified spike protein-carrying peptides in detail below, I believe that the peptide identified by PEASK de novo system, even using a database search, requires certain caution for its true identification or not. Therefore, the appropriate validation is necessary and should be performed.

2). To date, dozens of MS2PIP version are available. The detailed information can be found at the MS2PIP server website (<https://iomics.ugent.be/ms2pip/>). The authors described in the main text (page 14, line 431) that they used Immuno-HCD MS2PIP for MS2Rescore system. While the authors mentioned that they acquired their mass data by CID collision in the main text (page 13, line 419). From my best knowledge, the collision by CID induces only primary cleavage while the HCD induces multiple cleavage after primary cleavage. Thus, even if the peptide sequences are identical, the spectral patterns between HCD and CID collisions become different. Actually, as far as I know, in one vendor's prediction system, HCD and CID collisions are independently handled and therefore, the data acquired by CID collision is not applicable for HCD-based prediction system. I'd like to know this is really feasible to use HCD model for CID collision data. In Figure 4a, the authors used publicly deposit data that uses ETHCD, this again, another different collision type-based data for comparison. Is it really applicable to used HCD model for non-HCD data? The authors should clarify the validity and describe the rationale why they can use HCD-based MS2PIP model for non-CID collision data.

3). The authors used artificial model of SARS-Cov-2 antigen discovery in this study. The antigen identification can be largely affected by the abundance of source proteins/peptides, while there was no description of how much of source spike peptides (S1 & S2) are expressed in transfectants except for the number of GFP-derived immunopeptide identification. The pcDNA3.1 system the authors used in this manuscript is an over-expression plasmid with GFP and authors cloned transfectants from each cell lines (JY & Raji) by FACS. During this process, the authors should have been able to establish a few clones for Spike-segment 1(S1) and S2 as well. Since one of the

claims by the authors for "Thunder-DDA-PASEF" is "sensitive", it's better to clarify about the identification efficiency by comparing the low copy and high copy of S1 and S2 transfectants. In addition, SARS-Cov-2 should infect to the adherent respiratory cells, not the floating B cell lymphoma cell lines. I'd like to know why the authors chose JY and Raji as model cells for SARS-Cov-2 model. Please rationale and mention why they used JY and Raji in this manuscript. Further, is it possible to obtain the MOI-known SARS-Cov-2 infected cells for Thunder-DDA-PASEF validation? Because cDNA-derived linearized peptides and the naturally translated viral spike proteins may be different in 3D structures and PTMs that can also affect the intracellular processing of immunopeptides. Lacking the lysine for ubiquitination and the partial expression that ignore the domain structures of source proteins can induce unnatural cleavage by that never reflect the intrinsic enzymatic activity. For this reason, it would be better to specify that whether antigens obtained from artificial models can also be identified in actual specimens (i.e., whether T cells responsive to the candidate sequence can be detected in the patient is another story).

4). It was kind of laborious work to connect the main text and corresponding data from main figures and supplementary materials because the necessary information was not organized well in one place, and thus my apologies in advance if I understood the data incorrectly, while I think there are typographical error or the mishandled data in supplementary material 10 regarding about the validation of peptide identification by USE.

In Figure 6d, there is one shared spike protein-carrying peptide (SIIAYTEMSL, $z = 1$) both from JY and Raji. In here, Figure 6d describes this peptide is singly-charged one. But in the Supplementary material 10, there were 2 MS2 for SIIAYTEMSL ($z = 1$) and SIIAYTEMSL ($z = 2$). Please check this point and correct or explain the data accordingly.

Further, I'm skeptical about the MS2 spectra used in the bottom for these two (SIIAYTEMSL of $z = 1$ and $z = 2$) that these are identical. The $z = 1$ and $z = 2$ spectra can be distinct from each other, while these two seems exactly the same. I think the processing by USE validation can include some kind of errors because not only for this, I found a few descriptions of "Procit CE: XX" on top spectra, that made me wonder the authors really used their raw file derived data for top spectra. Could you check and explain if I misunderstood this description? As such, the Figure 6d and the Supplementary material 10 are connected but really laborious to comprehend. It may be better to reorganize these materials for easy reading.

5). One of the advantages of "Thunder-DDA-PASEF" by the authors in this manuscript, is efficient identification of singly-charged immunopeptides that have not been paid so much attention to date. Still, the verification for this critical main claim by the authors is, I'm afraid but I must say, insufficient.

As examples were mentioned above, at least it seemed to me, the important MS2 spectra validation was not performed properly.

And more in detail, there were 9 Singly-charged immunopeptides derived from S1 and S2 model samples out of 18 validated peptides. While one (AIHVSGTNGTK) was obviously a false discovery because of the unmatched spectra against the predicted MS2 by USE. The authors mentioned that they included this peptide for validation because of the multiple identification from analyses while they didn't mention which of these peptides are the true discovery and false discovery. Applying the scoring system for validation, it is important to set the threshold like "more than score XX is valid and regarded as true discovery". Unfortunately, the authors didn't clarify this point, so it is very unclear whether the authors consider which of these peptides are the true discoveries or the false discoveries. The authors used USE that provides the score of SA and PCC, still it appears that the authors paid not so much consideration to SA. If not, isn't it better to use other calculation system, like dotp score Sykyline, or any other systems. I couldn't find what is the valid score for PCC by USE, but if you judge the identification only by PCC, still, you need to clarify what is the valid score for true identification and apply this score for further data validation. In main text, the authors described that the $PCC \geq 0.85$ as high-confidence identifications, while the Supplementary material 10 includes peptides that has PCC below 0.85 (ex., RQIAPGQTGK & SIIAYTMSL ($z = 2$)). As such, there is unmatched criteria between the main text and the data. Please set the threshold and clarify this point.

And the spectra comparison, if you simply inject more volume of synthetic peptides for Thunder-DDA-PASEF, can't you have better spectra for synthetic-APHGVVFL peptide that lacks b4 ion from the spectrum?

Some of those spike protein-carrying peptides were found as multiple charge state, like $z = 1$ and $z = 2$. How were the results of USE for those different charge state from same peptide? If the authors claim that the Thunder-DDA-PASEF has advantage in the identification of singly-charged immunopeptides, (as a matter of fact, the increased immunopeptidome by Thunder-DDA-PASEF seemed largely by increased singly-charged immunopeptide identification apparently, from main Figure 3a & 3b), the validation for this specie has to be done more thoroughly. There are 5 peptides only found by $z=1$ (Figure 6d). While the validation score for these 5 peptides are not high enough (if you set the threshold 0.91 like dotp score by Skyline system, only 1 candidate left since the data for SIIAYTMSL is currently compromised). And I would say, the spectra NSASFSTFK ($z=1$), the intensity of b and y ion series are obviously different. So this may be a false discovery. Still the PCC score is ranging from 0.85~0.86. Thus the setting $PCC > 0.85$ as high-confidence identifications is not acceptable, at least to me. Since it seems that the bottom spectra used for these peptides were from predicted spectra, I think it is better to acquire the data by using cognate synthetic peptide to show convincing "true" identification. Please consider to add more validation analyses and reconstruct the Supplementary material 10 accordingly.

Response to Reviewers (revision 1, *Nature Communications*)
Thunder-DDA-PASEF boosted by MS²Rescore enables high-coverage immunopeptidomics and identifies HLA class-I presented SarsCoV-2 spike protein epitopes

Table of Contents

I.	Introductory comments and repository access codes.....	2
II.	Main modifications for method validation	4
III.	Responses to comments of Reviewer #1	8
IV.	Responses to comments of Reviewer #2.....	16
V.	References	27

I. Introductory comments and repository access codes

I.1. General comment to Reviewers

Dear Reviewers,

We thank you for the time and effort dedicated to evaluating our manuscript and for the constructive feedback. In addition, we sincerely appreciate the kind words about the interest and value of our work. We have included answers to your questions in this document and updated the documents in the submission file. We have applied the modifications suggested and extended our explanation on key points highlighted by your questions. This has resulted in significant improvements in the revised version of the manuscript and also the supplementary data. In this document, we answered your questions and we provide a description of the actions taken to respond to your concerns. We believe that the manuscript is up to the standards of *Nature Communications*, and we hope you will agree with us and recommend our work for publication. In any case, we would be happy to address any further questions or suggestions.

In their summaries, the Reviewers correctly highlighted most of the key points of our manuscript. In both cases, adapting the isolation polygon to include singly charged peptides was mentioned as the main improvement. However, we also optimized other essential MS parameters to increase the immunopeptidome coverage. The comments of the Reviewers helped us realize that we gave too much emphasis to the singly charged peptides in our previous version of the manuscript. Thus, we adapted the text to mitigate this point and further highlight additional advantages of our MS method and the overall workflow. Importantly, in this new version, we include a retrained version of MS²Rescore optimized for timsTOF Pro HLA_p data. In addition, we show how Thunder-DDA-PASEF can be used to identify candidate antigens of interest that could be further studied for developing therapies.

Both Reviewers raised questions regarding the performance of Thunder compared to other immunopeptidomics methods and suggested extending the method validation. Thus, in Section II of this manuscript, we have compiled a description of the main modifications done in the manuscript to address these concerns.

I.2. Repository access credentials

Raw data and search results have been deposited to ProteomExchange and are accessible with the following Reviewer login credentials. In addition, we included a List of datasets and their repository locations in Supplementary Material S11.

Data from Tenzer lab:

Identifiers: PXD040385 for ProteomeXchange and JPST002044 for jPOST

URL: <https://repository.jpostdb.org/preview/209791232665415e51504c9>

Access key: 3041

The files of the JY immunopeptidomics data used to train MS2PIP were recently published¹ and are already publicly available with the dataset identifiers PXD043026 for ProteomeXchange and JPST002158 for jPOSTrepo.

HL60 immunopeptidomics data from Carapito lab:

URL to log in: <https://www.ebi.ac.uk/pride/login>

Project accession: PXD046535

Username: reviewer_pxd046535@ebi.ac.uk

Password: t0tCB5Cy

HeLa tryptic proteomics data from Carapito lab:

URL to log in: <https://www.ebi.ac.uk/pride/login>

Project accession: PXD046543

Username: reviewer_pxd046543@ebi.ac.uk

Password: KFdT3vsx

II. Main modifications for method validation

II.1. (Q2c, Q8a, Q8b, Q12, Q22a) How does Thunder-DDA-PASEF compare to other immunopeptidomics methods? and how did the authors select the methods and data sets for comparison?

We have extended the comparison of Thunder-DDA-PASEF against other methods by acquiring data from aliquots of a same sample in an Orbitrap Exploris 480. In addition, we removed the comparison against the Feola *et al.* dataset (see paragraph below “Removal of Feola dataset”). The results are detailed in the manuscript, section “Thunder-DDA-PASEF outperforms state-of-the-art MS immunopeptidomics methods” (L248-285). To avoid introducing a bias due to differences in the prediction performance, we did not rescore the results of any of the data sets compared. In summary, Thunder-DDA-PASEF provided the highest number of JY HLAIs, followed by Demmers *et al.*, which showed complementarity, probably due to the use of EThcD fragmentation. Although the HLAIp-tailored polygon is not essential for identifying singly charged peptides, it enables the efficient fragmentation of peptides with the expected size, resulting in an efficient usage of instrument cycle time. We would like to highlight that the enhanced coverage provided by Thunder-DDA-PASEF is also due to the TIMS separation and high sensitivity provided by the timsTOF Pro2 MS and boosted by the optimized parameters (300 ms TIMS ramp, 3 MS/MS frames, high sensitivity mode).

The paragraphs below further explain our rationale behind the method and public dataset selection.

- a) **Direct comparison against Exploris 480:** The expression of HLA proteins and the characteristics of the peptides presented differ extensively between cell lines (and tissues). Thus, we decided to evaluate Thunder against other methods profiling the JY HLA-ligandome. To obtain a direct comparison, we acquired data from aliquots of a sample using our method and DDA in an Orbitrap Exploris 480 with and without FAIMS-based gas-phase-fractionation (GPF). We selected this method because Klaeger *et al.* recently showed that GPF increases the HLA-peptidome coverage, and the required instrumentation was available in our laboratory.
- b) **Comparison against public datasets:** In addition, to compare Thunder against results from other established immunopeptidomics laboratories, we included the datasets of Demmers *et al.* (EThcD-DDA) and Pak *et al.* (DIA-BigLib). We selected these two data sets since they represented the deepest characterizations of the JY HLA-ligandome available in the literature to our knowledge. To avoid introducing a bias due to differences in the prediction performance, we did not use MS2Rescore to reprocess any of the results shown in this comparison.
- c) **(Q8.b) Removal of Feola dataset:** In the previous version, we included the comparison against Feola *et al.* because the authors also used a timsTOF Pro instrument to profile the JY HLA-ligandome. The number of cell equivalents injected was 5-fold higher than in our study (50 vs. 10 million). Thus, we initially thought that this was a fair comparison. However, sample preparation

is also critical for the sensitivity of immunopeptidomics experiments, and the starting material used by Feola *et al.* was 10-fold lower than ours (50 vs. 500 million cells). In addition, the authors restricted the TIMS range to 0.6 – 1.3 1/K0, effectively excluding the singly charged ions eluted at > 1.3 1/K0 (Supplementary Fig. S1). Thus, we agree with the Reviewers that the methods and results are not directly comparable and we have removed this dataset from the main manuscript and included it only in the Supplementary (Fig. S1.)

II.2. (Q3d, Q8c, Q22b, Q33) How does Thunder-DDA-PASEF compare to other MS methods relying on charge-state screening?

To evaluate the value of Thunder-DDA-PASEF for identifying singly charged ions, we performed several comparisons: we extended the comparison with the HLAIp-tailored polygon and without any isolation polygon in the timsTOF Pro2. In addition, we compared the detection of peptides in function of their charge state in Orbitrap instruments as described under point I.1 and section 2.5 in the manuscript. In summary, similar proportions of singly charged peptides can be identified by relying on charge-state screening. Still, Thunder-DDA-PASEF provides a deeper profiling of the HLA-ligandome overall. Although the HLAIp-tailored polygon is not essential for identifying singly charged peptides, it enables the efficient fragmentation of peptides with the expected size, resulting in an efficient usage of instrument cycle time.

- a) **Comparison against Orbitrap data.** High proportions of singly charged peptides were also detected in the Orbitrap data sets (with or without FAIMS) mentioned above. Nonetheless, the optimized Thunder-DDA-PASEF outperformed the Orbitrap analysis regarding total HLAIs identified and contributed with the most singly charged peptides in the combined data set (Lines 278-282, Fig 5d). In addition, these results validate the identification of singly charged peptides by diverse instruments and methods.
- b) **(Q8.c) Absence of singly charged ions in the Demmers dataset.** We want to clarify that Demmers *et al.* (2021) did not include the fragmentation of singly charged peptides because it is intrinsically incompatible with ETHcD due to the electron transfer. However, it provides complementary fragmentation patterns and improves the identification of peptides with higher charge states². Thus, we included it in the method comparison for the abovementioned reasons.

II.3. (Q5) The authors should validate their results across a broader range of samples with different HLA types.

To better understand the effects of the HLAIp-tailored Thunder isolation polygon and the DDA-PASEF optimization across distinct HLA-binding peptide motifs, we analyzed samples with diverse alleles using methods without and with the HLAIp-tailored isolation polygon (Thunder), combined with the original (100ms_None and 100ms_Thunder) and optimized DDA-PASEF settings (300ms_None and 300ms_Thunder) (also see Manuscript Section 2.4). Compared to 100ms_None, the 300ms_Thunder

provided a total increase of 87.4 to 125% in HLAIPs depending on the sample, thus, on average, doubling the immunopeptidome coverage.

This experiment also highlighted that, while the polygon is not essential to identify singly charged peptides, it improves the number of HLAIPs identified combined with the optimized parameters at 300 ms TIMS ramp. Compared to the 300 ms method without polygon, the HLAIP-tailored (Thunder) polygon improved the identification of HLAIPs from 5.1 to 16.5% in all the four sample types tested, covering 17 distinct HLA alleles. Although this is a modest increase compared to the effect of the optimized DDA-PASEF settings, it is substantial for immunopeptidomics experiments.

II.4. (Q7.a, Q25) To improve the rescoring performance, the authors should retrain the prediction models for timsTOF CID data using samples with different HLA backgrounds or synthetic peptides.

Reviewer 2 highlighted that models trained on immunopeptidomics HCD data may not provide precise predictions for CID data. Reviewer 1 suggested acquiring data from more samples with different HLA backgrounds to train MS2PIP. To improve the prediction accuracy, we trained an MS²PIP **timsTOF model** from diverse tryptic and non-tryptic samples: JY trypsin digest, HeLa trypsin digest, JY elastase digest, HLA immunopeptidomics samples from JY, HeLa, SK-MEL-37, and HL60 cell lines. In total, 241,104 peptides (including modifications) were used to train the model and a distinct set of 10,045 peptides to test it. The immunopeptidomics data covers 17 diverse HLA alleles. In addition, in a recently published project, we developed a model to predict collision cross section (CCS) based on tryptic peptides, phosphopeptides, and JY HLAIPs. The model (GRUpredictor), developed as part of the tool ionmob, showed the benefit of helping discern true from false hits during rescoring. Thus, we have now included it as a feature in the mokapot rescoring in the latest version of MS2Rescore. Altogether, this has resulted in a substantial improvement in the rescoring performance as described in section 2.6 and Supplementary Fig. S9.

In the previous manuscript version, we used the Immuno-HCD MS2PIP model presented by Declercq *et al.*³, increasing the identification of 8-13-mers by 16.7%. It was previously shown that a model trained on HCD-Orbitrap had a similar performance in predicting CID-TOF data (TripleTOF 5600+), at least for tryptic peptides⁴. Thus, we previously thought that the HCD-Orbitrap immunopeptidomics model would also translate correctly to timsTOF data. However, recent data indicated that HCD-Orbitrap data do not translate similarly for timsTOF instruments. We thank the Reviewers for highlighting this error and giving us the opportunity to fix it.

II.5. (Q7.b, Q27.a) The authors should further evaluate the sensitivity of Thunder-DDA-PASEF

To assess the sensitivity of Thunder-DDA-PASEF, we analyzed diverse injection amounts of JY HLAIs using Thunder-DDA-PASEF. To evaluate the effect of rescoring on the sensitivity, we compared the results without and with rescoring using the updated MS2Rescore. PEAKS without rescoring identified 3,328 +/- 405 (CV = 12.2%) HLAIs from only 1 Mce and up to 9,753.67 +/- 198 (CV = 2%) from 20 Mce. Using MS2Rescore resulted in 6323 +/- 737 (CV = 11.6%) HLAIs from 1 Mce, and 14,577 +/- 327 (CV = 2.2%) from 20 Mce. In summary, our workflow using Thunder-DDA-PASEF and PEAKS+MS2R provided an in-depth characterization of the HLA class I immunopeptidome, even from 1 million JY cell equivalents.

III. Responses to comments of Reviewer #1

III.1. Remarks to the Author:

Q1. *Summary: In this paper the authors present an optimised method for immunopeptidomic analysis using a TIMSTOF mass spectrometer from Bruker. They take advantage of the ion mobility separation to focus the instrument on the sequencing of HLA peptides by expanding the proteomics setting to include singly charged species. It is widely accepted that in immunopeptidomics sequencing singly charged species adds value to the subsequent peptide ID rate. They report that including these species enhances the identification rate in immunopeptidomics and that this can be done using ion mobility rather than precursor filtering.*

We thank Reviewer #1 for highlighting the key points of our method, the efficient selection, fragmentation, and identification of singly charged HLA peptides using the HLA-tailored isolation polygon.

III.2. My general comment:

Q2. *The importance of inclusion of singly charged species in immunopeptidomics mass spectrometry experiments is not a novel finding and reported widely for many instrument types. However, the sole use of the ion mobility device to focus the sequencing depth in the region where these peptides are found is of interest to the community as these type of instruments become more common in labs carrying out immunopeptidomics. This method reiterates the importance of optimising your mass spectrometer for immunopeptidomics and not proteomics. It is overall very useful to the community but could benefit from some more rigorous testing and data analysis as suggested below (a). In my opinion the authors sometimes over-state the improvements (b), and cherry pick evidence/datasets to support their statements rather than rigorously testing/investigating the methods performance (c). In general I support the publication but want to see it fully road tested (d).*

(a) We appreciate the kind words about the value of our work and Thunder-DDA-PASEF for the community and the suggestions for rigorously testing it.

(b) We have performed additional experiments to support our conclusions better and mitigated the interpretation.

(c) We have extended the comparison of Thunder-DDA-PASEF against other methods. In addition, we used the method to profile the immunopeptidome of more cell lines and human plasma with 17 distinct HLA alleles. See Section III.3 in this document for details.

(d) By following the suggestions of the Reviewers, we are convinced that we have now rigorously tested Thunder-DDA-PASEF against other MS-immunopeptidomics methods.

III.3. Section specific comments

III.4. Section 2.2

Q3. *It is not fully investigated by the authors how the ion mobility thunder windows alter the result from having no windows, and just expanding the ion mobility range from the standard proteomics workflow. The authors show there is no improvement here in total identifications, but that less MS2 are acquired and without the window you identify larger peptides, and these are from contaminants. I'd like to know more about the costs of using the polygon window approach. It is important to know the nature HLA peptides missed by including such a filter, are the filtered-out peptides predicted binders (a), are post-translationally modified forms missed because they have different CCS behaviour? (b) Additionally, how do these results compare to simple charge-state screening without the windows as implemented by most mass spectrometers? (c)*

To better understand the effects of the HLAIp-tailored Thunder isolation polygon and the DDA-PASEF optimization across distinct HLA-binding peptide motifs, we analyzed samples covering 17 alleles using methods without and with the HLAIp-tailored isolation polygon (Thunder), combined with the original (100ms_None and 100ms_Thunder) and optimized DDA-PASEF settings (300ms_None and 300ms_Thunder). This is detailed in Section 2.4 of the manuscript.

(a) The Thunder isolation polygon limited the fragmentation and identification of peptides to the area within its boundaries. In contrast, no polygon resulted in a more extensive distribution of peptides across the $1/K_0$ vs. m/z dimensions. However, even in the acquisitions without polygon, 99.9% of the predicted HLA binders (HLAIps) were detected within the defined boundaries of the Thunder polygon, showing that it does not exclude peptides from any of the HLA motifs screened. Less than five HLAIs were detected without polygon outside of the boundaries defined by the Thunder isolation polygon. These could be captured by slightly expanding the polygon but are, in any case, negligible.

(b) Post-translational and other chemical modifications can indeed alter peptide ion mobility. Thus, this could place the peptides outside the isolation polygon in the $1/K_0$ vs. m/z space. However, this was not the case for the modifications included in our data processing since more than 99.9% of the HLAIs were detected within the boundaries of the Thunder isolation polygon. This remains to be evaluated for other modifications but it was not the objective of this project since we did not perform any PTM enrichment.

(c) Similar proportions of singly charged peptides can be identified by relying on charge-state screening, but Thunder-DDA-PASEF results in a deeper profiling of the HLA-ligandome overall. Thus, although the HLAIp-tailored polygon is not essential for identifying singly charged peptides, it enables the efficient fragmentation of peptides with the expected size, resulting in an efficient usage of instrument cycle time. As a result, the Thunder polygon provides a further increase of 5 to 16% more HLAIs identified in the optimized (300 ms) method than the same method without polygon. For details, see sections II.1 and II.2 of this document.

Q4. The authors should include some comparisons to the none-polygon protocol so that we can see its effects on HLA allotype, binding affinity and dynamic range.

We have extended the comparisons of the method without polygon to the HLAIp-tailored polygon with a more considerable diversity of samples covering more HLA alleles (see manuscript section 2.4 and Supplementary Figures S3 and S4). For details, see the answer to Q3.a and Section III.3 in this document.

Q5. Are certain alleles that carry more charge (2/3+) sacrificed in order to allow singly charged species does this lead to bias in samples that contain peptides with different anchor residues...? (a) How does this compare when you analyse samples with different HLA types, the authors should validate their results across a broader range of samples with different HLA types and/or in a synthetic peptide library consisting of a mixture of different alleles.? (b)

(a) The singly charged peptides form a distinct ion cloud in the $1/K_0$ vs. m/z space separated from the multiply charged peptides. In addition, these two ion clouds do not overlap in the ion mobility ($1/K_0$) dimension (see Figure 2 or Figure 3). Thus, the method does not sacrifice the fragmentation of multiply charged ions to fragment singly charged ions. Furthermore, we made sure the Thunder isolation polygon includes such ions. We evaluated this across four different samples and observed that the charge distribution remained constant across the methods without and with the Thunder polygon. Furthermore, none of the predicted multiply charged HLAIPs was detected outside the boundaries defined by the polygon.

(b) It was indeed necessary to expand the evaluation to a more considerable diversity of samples with distinct HLA types. Purchasing an extensive library of synthetic peptides was unfortunately not possible. Thus, we extended the evaluation of the method using four different samples covering 17 HLA alleles. See the answer to Q3 for details.

III.5. Section 2.3

Q6. All the results are confined to the supplementary I think they mention some important optimisation steps that could be included in the main figure here, especially considering that these changes appear to have a more of an affect on ID rates than the polygon shape changes on which the paper is based upon, however it seems that these adaptation are not novel and are mentioned in other previously published work is this why they have been put in the supplemental?

We agree with the Reviewer on the importance of the figure showing the MS optimization steps (formerly figure S2). Thus, we have included it in the main manuscript (now figure 3). We originally placed it in the supplementary material to simplify the main document. We presented this optimization at a webinar for users of Brukers instruments in 2021 (1 March), and at the ASMS Conference in 2021 (31 October-4 November). However this optimization on a timsTOF Pro instrument has not been published before. Feola *et al.* ⁵ (October 2021) also used a 300 ms TIMS ramp with 3 MS2 frames, but they did not explain

the rationale behind these decisions. Besides, they did not show any comparison against other methods in the timsTOF Pro or other instruments. In addition, the authors restricted the TIMS range to 0.6 – 1.3 1/K₀, effectively excluding the singly charged ions eluted at > 1.3 1/K₀ (Supplementary Fig. S1) and did not activate the high sensitivity mode. In contrast, we show step-by-step optimization, revealing that these adaptations are crucial to maximizing the identification of HLAIPs. In addition, we compared the results against other state-of-the-art methods.

III.6. Section 2.4

Q7. The authors use MS2 rescore a machine learning algorithm to improve their identification rates, the method is not novel or specifically developed for the data generated by this instrument and their observations recapitulate what has been seen for this machine learning approach. Potentially, the authors could improve the approach for singly charged ions which seem to be less well rescored compared to multiply charged species. Do the authors have a synthetic library they could acquire (or more samples with different HLA backgrounds) with their new method and provide this as a resource for training better algorithms? (a) They could also use this to validate the sensitivity of their method (b) and explore if any alleles bias occurs when they alter their methods (c) .

(a) Following the suggestion of the Reviewer, we trained a novel MS2PIP timsTOF model using data acquired with Thunder-DDA-PASEF from diverse tryptic and non-tryptic samples. For details, see Section II.4 in this document and Section 2.6 in the manuscript.

(b) Following the suggestion of the Reviewer, we implemented the new rescoring model to assess the sensitivity of Thunder-DDA-PASEF without and with rescoring by injecting 1 to 40 million cell equivalents of a JY HLAIP sample. For details, see Section II.5 in this document and Section 2.6 in the manuscript.

(c) We evaluated this across four different sample types covering thirteen HLA alleles (Fig. 4f). The proportion of 8-13-mers predicted to bind the corresponding HLA alleles was roughly similar without and with the polygon. More non-binders were identified in the optimized (300 ms) methods compared to the original configuration (100 ms), but this was due to the enhanced sensitivity provided by the LSA mode.

III.7. Section 2.5

Q8. The authors make a series of comparisons between the data they generate here and other datasets. How they choose the sets is unclear they seem to consist of three datasets that use very different methods for different purposes (a). For example the work by Feola develops a microfluidic device for high sensitivity immunopeptidomics and uses a lot less cell input, these factors alone limit the depth of their analysis and therefore the 720 % increase is not unexpected and doesn't really demonstrate the value of their new method... (b) Whilst of use, I believe the comparison shows that including singly charged species by TIMS is complementary to other approaches (c). The most appropriate comparison is with Demmers et al., 2021. Where the inclusion of singly charged ions is off (d).

(a) In the previous version of the manuscript, we compared our method to previously published JY HLAIP datasets also using a timsTOF Pro (Feola *et al.* 2021) or providing the highest coverage in Orbitrap instruments (Demmers *et al.* and Pak *et al.*). **(b)** To avoid comparing against a method using very low cell inputs for sample preparation, we removed the Feola *et al.* dataset from the comparison. **(c)** To further investigate the complementarity and overlap of Thunder-DDA-PASEF to other approaches, we added an *in-home* direct comparison against the Orbitrap Exploris 480 without or without gas phase fraction using FAIMS. **(d)** We agree that the most appropriate comparison was with Demmers *et al.*, where the singly charged ions were excluded from fragmentation since EThcD fragmentation results in the charge loss of fragments originating from singly charged peptides. For details, see Section II in this document (a & b in II.1; b in II.2).

III.8. Section 2.6

Q9. Can the authors comment on the overall reproducibility in this experiment?

This was already commented on in L346-350 and shown in Fig. S10.

Thunder-DDA-PASEF using PEAKS+MS2R identified in total 22,501 peptides from JY and 28,429 peptides from Raji, comprising 78% of 8-13-mers, with a median length of 9 AAs (Fig. 6a, Supplementary Material S6), as expected for HLAIPs. The reproducibility between biological replicates ranged between 36.4% and 63.1% 8-13-mers identified in all the samples of the same genotype and 58.9% to 82% regarding the proteins covered (Supplementary Fig. S10).

These numbers represent the combined reproducibility or variability of the biological samples, sample preparation, and acquisition. The reproducibility of acquisition itself was around 75% of peptides identified across the triplicates of injection, which is not uncommon for DDA analyzes. Thus, it is not surprising that the reproducibility decreases in such large datasets.

Q10. Was the binding affinity/netmhc the same as in the method development section, how reproducible was this?

The JY HLAIPs sample used for method optimization resulted in 94% of predicted binders out of 11,634 8-13-mers in the Thunder+MS2R analysis. In the experiment shown in Section 2.6, the predicted binding was 78.9% for the JY and 77.6% for Raji samples, including the WT and the S1 and S2 transfected cells. Thus, the HLAIP enrichment procedure was less specific on these samples. We recognize this as a disadvantage of this experiment. Therefore, we have commented on it as a limitation in the manuscript (L350-352). Many factors could have impacted the sample preparation reproducibility, such as using different antibody or agarose beads batches. This issue is well-known in the field and is not the subject of this manuscript, where we focused on optimizing the MS method. In addition, to compensate for biological and sample preparation variability, we prepared biological duplicates of each transfectant and triplicates of the WTs.

Q11. *Here they choose two cell lines with distinct HLA types it would be good to see how the polygon on/off might affect the overall detection of the different peptides (see my comments re section 2.2).*

We evaluated the IMS vs. m/z peptide distribution using five sample types (JY, HeLa, and SK-MEL-37 cells, and human plasma). We observed that the polygon does not exclude possible HLA binders, at least for all the 17 HLA types tested (Supplementary Fig. S7). For details, see answers to Q3 and Q5, and section 2.4 of the manuscript. In addition, we also show the IMS vs. m/z peptide distribution of HLAIs for Raji samples acquired with the Thunder polygon (Supplementary Fig. S11d).

Q12. *Line 249: Why compare it to Pak et al., whilst the more fair comparison would be to Demmers et al.,? I am not sure how this demonstrates the capacity of the method. I am sure it does have the capacity, but can it be better demonstrated?*

We agree with the Reviewer. In addition, the comparison to other methods is already covered in section four of the results. Therefore, we removed this phrase (copied below, phrase with strike-through) from the manuscript. For details, see Section II.1 in this document and Section 2.7 (Lines 360-363) in the manuscript.

Each protein group was represented by a median of 2 HLAIs per protein group and 75% of them with one to three peptides for both cell lines. ~~As a comparison, the DIA analysis of JY HLAIs by Pak provided a median of one HLAIs per protein (Pak, 2021).~~

Q13. *Did you expect the same peptides when they have different alleles?*

No, we expected to identify different HLAIs between JY and Raji cells. The complementarity of their immunopeptidomes was one of the reasons for selecting these cell lines. We have adapted the text to clarify these points (Lines 343-345).

Q14. *Can the authors elaborate on how this analysis represents a resource for further exploitation, do they identify peptides that could be targets for immunotherapy, for example with the new method is the depth of coverage of potential/known tumour antigens greater than before?*

We have removed the correspondent text from the Results section. The text was misplaced here since it should have been part of the discussion. To show the usability of data generated by Thunder-DDA-PASEF for training predictor models, we have now included the training of MS²PIP (MS/MS fragmentation) in this manuscript, and we recently published a CCS prediction model as part of the ionmob python package (GRUpredictor)¹, respectively.

III.9. Section 2.7

Q15. Can the authors justify how peptides presented by B cells are clinically relevant to SARS-CoV2 infection? The transfection is meant to mimic infection, are B cells relevant could you not of transfected Epithelial cell lines?

The Reviewers correctly indicated that it would have been more clinically relevant to perform this experiment using cells that are known to be infected by SARS-CoV-2. However, this was not possible then, and we opted for a strategy that would maximize the immunopeptidome coverage. Thus, we used B-cell-derived cell lines since they also express the required antigen processing and presenting machinery and high levels of HLA. We adapted the Results and Discussion sections to explain the selection of cells better and to mention this point as a limitation of our study (L342-345, L473-475).

Q16. Can you please put these peptides in context with the other studies, was the lack of overlap with other studies due to overall depth or HLA type?

We have extended the discussion to better put the peptides in context with previous studies (L483-501). The low overlap of HLA-Ips with other studies is due to the low number of previously reported sequences and cell lines used, since differences on the antigen processing and presentation machinery (e.g., the HLA alleles) can lead to different sets of peptides. Data available from MS-based profiling for the spike glycoprotein-derived SARS-CoV-2 HLA-I ligandome deposited on IEDB is limited to two publications^{6,7} describing nine epitope sequences. On top, Xin et. al. 2022 reanalyzed data from Weingarten-Gabbay and additionally defined “SIIAYTMSL” as a SARS-CoV-2 Spike glycoprotein-derived peptide⁸. Thus, our analysis provides a complimentary set of spike HLA-Ips that had not been previously reported and also confirms SIIAYTMSL. Due to the limited MS-data availability for SARS-CoV-2 HLA-I peptidome of the Spike glycoprotein, we also included a comparison against other assays (Response Frequency: B- and T-cell assays). Here, we found that seven of our reported peptides (APHGVVFL, RLQSLQTYV, RLDKVEAEV, IPTNFTISV, SIIAYTMSL, VLYNSASFSTFK, GYVFASTK) elicited immune responses in at least 10% of the subjects. Over half (4/7) of them are singly-charged.

We appreciate and acknowledge the concern of the Reviewer. While the lack of overlap with other MS-based studies certainly is affected by limited data availability, we still think that the considerable overlap (including many singly-charged peptides) with non-MS assays shows the value of the method. Besides, all our experimentally postulated, stringently filtered peptides were predicted to bind Raji and JY HLA alleles and finally validated with synthetic peptides. Eight of the sixteen postulated peptides were not previously described but validated with synthetic peptides for our experimental model. Although these peptides remain to be further evaluated for their diagnosis or therapeutic value, this shows that Thunder-*DDA*-*PASEF* can confirm predicted immunopeptides and also discover potential novel targets. Some limitations of our experimental model are described in the manuscript (L342-343, L473-475, L479-481).

Q17. Line 293: this sentence doesn't make sense to me perhaps elaborate.

HLA-C expression at the cell surface is generally lower than HLA-A and HLA-B. Several publications have shown that this results in a lower proportion of HLA-C-binders presented than HLA-A and HLA-B^{9,10}. We observed this same pattern in our data sets for the overall HLA ligandome. We initially thought it was important to mention it. However, this is not a new finding and does not provide useful information in the manuscript. Thus, we removed the corresponding text.

III.10. Section 3 Discussion.

Q18. Line 311: In my understanding Thunder is a sound not a shape.

We replaced the “thunder-shaped isolation polygon” with “HLA_p-tailored "Thunder" isolation polygon”. Although Thunder is indeed not a shape, we named the HLA_p-tailored polygon as “Thunder” to simplify the lecture across the manuscript. This was clearly stated now in L135-136.

Q19. I think it is misleading, they should rephrase the discussion to highlight that standard proteomics methods miss an important part of the immunopeptidome, that ionises in the singly charged CCS space. They have used TIMS filtering and reshaped the proteomics polygon to account for this.... Etc...

We adapted the initial paragraph of the discussion as the Reviewer suggested (L415-419).

Q20. Line 331: how can you achieve targeting 9-12 mers when you don't know the charge state, please explain or omit.

In the previous version of the Manuscript, we included this paragraph to discuss some limitations and possible improvements of the method. However, it was rather hypothetical. Thus, we have removed it and mentioned the limitations in other paragraphs.

To answer the question of the Reviewer, a more stringent isolation polygon could be designed based on previously acquired or predicted data. Although the peptides binding to different HLA alleles will differ, their behavior in the m/z vs. IMS space is similar, as shown by our profiling of several cell lines (Figure 4 and Supplementary Figure S3-4). Thus, this information can be used to find the edges for designing a more stringent isolation polygon.

IV. Responses to comments of Reviewer #2

IV.1. Remarks to the Author

Q21. *The manuscript “Thunder-DDA-PASEF enables high-coverage immunopeptidomics and identifies HLA class-I presented SarsCoV-2 spike protein epitopes” by David Gomez-Zepeda et al. reported new immunopeptidomic methodology by tims-TOF system. In this manuscript, the authors optimized default tims-TOF DDA workflow for immunopeptidomics as “Thunder-DDA-PASEF” by fitting the IM dimension for class I immunopeptides. Taking information of retention time and MS2 peak intensity predictions into identification process by combining the MS2Resocre algorithm, the immunopeptidome coverage was further improved. The authors focused on singly-charged immunopeptide that have not been fully analyzed by previous immunopeptidomics methodologies, and by thunder-DDA-PASEF, the authors were successful to largely expand the identification of singly-charged immunopeptides. In the end, the authors used this thunder-DDA-PASEF to identify the viral protein-derived immunopeptides by using artificial models of viral spike protein expressing JY and Raji cells to show the potential of this methodology.*

We thank Reviewer# 2 for highlighting the key aspects of our manuscript.

Q22. *In a situation where new tools for analyzing large proteomic data, including immunopeptidomics, are being reported one after another, further validation and explanation regarding the plausibility of peptide identification is necessary to accept the authors' assertion that Thunder-DDA-PASEF contributes to the depth of immunopeptidomics (a) and is particularly advantageous for the identification of singly charged immunopeptides (b).*

(a) To better demonstrate the performance of Thunder-DDA-PASEF, we have extended the comparison of Thunder-DDA-PASEF against other methods (see manuscript section 2.5). In addition, we used the method to profile the immunopeptidome of other cell lines and human plasma with diverse HLA alleles. See Section II.3, III.1 c) and III.3 in this document for details and manuscript section 2.4. By following the suggestions of the Reviewers, we are convinced that we have now rigorously tested Thunder-DDA-PASEF against other MS-immunopeptidomics methods.

(b) Our extended evaluation revealed that other methods can identify similar proportions of singly charged peptides as Thunder-DDA-PASEF. However, our method provides a significant improvement in the immunopeptidome coverage overall. Thus, we have adapted the text to show this better. For details, see Manuscript Section 2.5 (L274-282) and Section II.1 and II. 2 in this document.

IV.2. Major Comments:

IV.3. 1) [PEAKS peptide identification]

Q23. *About the database search by PEAKS XPro, the authors described in the main text that they filtered the proteins by $-10\lg P \geq 20$ (page 14, line 428). While they didn't mention about the setting of the "number" of unique peptides for protein identification. Did authors also set the number of unique peptide (0, 1, 2, 3..., etc) as " $-10\lg P \geq 20$ and (number) \geq of unique peptides" for protein identification by PEAKS XPro? If not, the default setting by the software should have applied "0" for the number of unique peptides (a). Under this condition, the protein list by PEAKS XPro may include the proteins identified by using only "de novo only" peptides. Therefore, if the authors claim that they performed "database search", that requires the peptide spectrum match against the fasta sequence in database. If the search setting of protein filter was " $-10\lg P \geq 20$ and '1' \geq of unique peptides", my concern is no longer necessary (b). While the authors mentioned about the protein identification in Figure 5d, it's better to confirm this point. Please check the list of proteins used in this manuscript were really "database-searched" proteins (c). And then clarify the setting for the protein filter and describe it in main text (d).*

We understand the concern of the Reviewer about peptides identified by *de novo* only and we agree that they should be further validated. For this reason, **we did not include any "de novo only" identification** either at the peptide or the protein levels. Accordingly, *de novo* only peptides are not considered in any of the peptide or protein counts.

(a) We used the default setting for "Unique peptides" (≥ 0). We have now included this information in the Supplementary Material S2a, tab PEAKS_XPro. **(b)** However, this setting **does not result in de novo only peptides in the peptide.csv file** used for all the downstream analyses. We verified this by confirming that all the entries in the peptide.csv files are assigned as "Found by PEAKS DB." In addition, we contacted the provider, Bioinformatics Solutions (BSI), and they confirmed that in PEAKS XPro *de novo*, only peptides are exported in a separate file, which we did not include in our evaluations. The exact answer from BSI is pasted below.

Question (DGZ): "When the number \geq of unique peptides is set to 0, does the peptide.csv report also include De novo-only peptides?"

Answer (BSI): "The number of unique peptides refers to whether a specific peptide sequence is present in one (unique) or more database proteins. When this number is set to ≥ 0 , the CSV export will not contain any de novo only peptides. All peptides will be matched to a DB protein. In order to view de novo only peptides, you will have to refer to the denovoOnly.csv or denovoOnlyAllCandidates.csv. These will not match any protein sequences in your database."

(c) The same peptide.csv file was used to report the number of proteins represented by immunopeptides. Since only peptides “Found By PEAKS DB” were used in all the datasets, the proteins reported are represented by at least one HLAp assigned by database search and none by de novo.

(d) The settings were already mentioned in the Methods Section, subsections “Peptidomics database search and rescoring” and “Data analysis and statistics,” and further described in the Supplementary Material (formerly S2a, now S2). Since splitting the information may have been an issue, we now compiled it in the subsection “Peptidomics database search and rescoring” and highlighted it by starting the paragraph with the phrase “**Data export for downstream analyses**” (in bold). In addition, we further clarified the thresholds used to export the results and explicitly mentioned that *de novo* only peptides were not used.

Q24. *Further, while de novo sequencing by PEAKS software has an advantage in exploratory identification for peptides, it has also been reported that the actual false identification, even under the same FDR setting, is higher in PEAKS by comparing the other search engines. (Assaf Kacen et al., Nature Biotechnology, <https://doi.org/10.1038/s41587-022-01464-2>, Supplementary Figure 1). Though I will mention about my concerns about the validation for identified spike protein-carrying peptides in detail below, I believe that the peptide identified by PEASK de novo system, even using a database search, requires certain caution for its true identification or not. Therefore, the appropriate validation is necessary and should be performed.*

As mentioned in Q23, we agree with the concern of the Reviewer regarding the need to validate peptides identified by *de novo* sequencing. Thus, **we did not include any de novo only identification.**

Current methods for FDR evaluation in database searches are indeed suboptimal for immunopeptidomics. To compensate for this, we trained a timsTOF model of MS2PIP and used it for identification rescoring (see Section II.4 in this document).

IV.4. 2) [MS2PIP rescoring]

Q25. *To date, dozens of MS2PIP version are available. The detailed information can be found at the MS2PIP server website (<https://iomics.ugent.be/ms2pip/>). The authors described in the main text (page 14, line 431) that they used Immuno-HCD MS2PIP for MS2Rescore system. While the authors mentioned that they acquired their mass data by CID collision in the main text (page 13, line 419). From my best knowledge, the collision by CID induces only primary cleavage while the HCD induces multiple cleavage after primary cleavage. Thus, even if the peptide sequences are identical, the spectral patterns between HCD and CID collisions become different. Actually, as far as I know, in one vendor’s prediction system, HCD and CID collisions are independently handled and therefore, the data acquired by CID collision is not applicable for HCD-based prediction system. I’d like to know this is really feasible to use HCD model for CID collision data.*

The Reviewer highlighted that models trained on immunopeptidomics HCD data may not provide precise predictions for CID data. Further evaluation indicated that the HCD models don't perform well for timsTOF CID data. To improve the prediction accuracy, we trained an MS²PIP **timsTOF model** from diverse tryptic and non-tryptic samples. For details, see Manuscript Section 2.6, 4.10 and Section II.4 in this document.

Q26. In Figure 4a, the authors used publicly deposit data that uses EThcD, this again, another different collision type-based data for comparison. Is it really applicable to used HCD model for non-HCD data? The authors should clarify the validity and describe the rationale why they can use HCD-based MS²PIP model for non-CID collision data.

To avoid introducing a bias due to differences in the prediction performance, we did not rescore the results obtained by Thunder when comparing against other publicly deposited data and own acquisitions on other instrument types that use HCD. Similarly, we did not rescore the Orbitrap results.

IV.5. 3) [spike immunopeptidome]

*Q27. The authors used artificial model of SARS-Cov-2 antigen discovery in this study. The antigen identification can be largely affected by the abundance of source proteins/peptides, while there was no description of how much of source spike peptides (S1 & S2) are expressed in transfectants except for the number of GFP-derived immunopeptide identification. The pcDNA3.1 system the authors used in this manuscript is an over-expression plasmid with GFP and authors cloned transfectants from each cell lines (JY & Raji) by FACS. During this process, the authors should have been able to establish a few clones for Spike-segment 1(S1) and S2 as well. Since one of the claims by the authors for "Thunder-DDA-PASEF" is "sensitive", it's better to clarify about the identification efficiency by comparing the low copy and high copy of S1 and S2 transfectants **(a)**. In addition, SARS-Cov-2 should infect to the adherent respiratory cells, not the floating B cell lymphoma cell lines. I'd like to know why the authors chose JY and Raji as model cells for SARS-Cov-2 model. Please rationale and mention why they used JY and Raji in this manuscript **(b)**.*

(a) The experiment suggested by the Reviewer would have been valuable to evaluate the relationship between number of transfectant copies and HLA I peptide ligands presented. However, we think that it would not be the best strategy for evaluating the sensitivity of a LC-MS method. Since the presentation of HLA Ips is regulated by a large diversity of factors beyond protein expression, this would result in confounding factors unrelated to the MS sensitivity. Therefore, to assess the sensitivity of Thunder-DDA-PASEF and how it is affected by rescoring, we analyzed diverse injection amounts of JY HLA Ips (see Section II.5 in this document and Manuscript Section 2.6). In addition, Thunder-DDA-PASEF identified peptides covering dynamic range of 4.5 orders of magnitude across diverse samples (Manuscript Section 2.4).

(b) As also mentioned in the answer to Q15, the Reviewers correctly indicated that it would have been more clinically relevant to perform this experiment using cells that are known to be infected by SARS-CoV-2. However, this was not possible then, and we opted for a strategy that would maximize the immunopeptidome coverage. Thus, we used B-cell-derived cell lines since they also express the required antigen processing and presenting machinery and high levels of HLA. We adapted the Results and Discussion sections to explain the selection of cells better and to mention this point as a limitation of our study (L342-345, L473-475).

Q28. Further, is it possible to obtain the MOI-known SARS-Cov-2 infected cells for Thunder-DDA-PASEF validation? (a) Because cDNA-derived linearized peptides and the naturally translated viral spike proteins may be different in 3D structures and PTMs that can also affect the intracellular processing of immunopeptides. Lacking the lysine for ubiquitination and the partial expression that ignore the domain structures of source proteins can induce unnatural cleavage by that never reflect the intrinsic enzymatic activity (b).

(a) Analyzing SARS-CoV-2 infected cells or patient samples would indeed be clinically relevant. However, such an experiment cannot be performed in our laboratory due to biosafety regulations. Besides, the objective was to optimize and validate the MS immunopeptidomics methods. We characterized the SARS-CoV-2 spike immunopeptidome also to provide possible valuable targets, which then can be further studied by other specialized laboratories. To avoid misinterpretations, we have mitigated our discussion regarding these peptides and highlighted that further validation is required.

(b) The 16 spike peptides detected in the immunoprecipitation-based immunopeptidomics experiments were predicted to bind the respective HLA alleles of JY and Raji. In addition, seven of the peptides have shown positive results in T-cell or HLA ligand assay, with a response factor from 10 to 100%, according to the IEDB. Although we cannot discard differences in the antigen processing mechanisms due to the expression via transfection, these results strongly support that they are HLA class I binders.

Q29. For this reason, it would be better to specify that whether antigens obtained from artificial models can also be identified in actual specimens (i.e., whether T cells responsive to the candidate sequence can be detected in the patient is another story).

We extended the discussion to compare the list of peptides identified by Thunder-DDA-PASEF to previous reports. Importantly, the peptide detected in both JY and Raji (SIAYTMSL) was also present in the data from Weingarten-Gabbay *et al.* (2021)¹¹, but it was only identified when Xin *et al.* (2022)⁸ reanalyzed the data using a novel deep learning algorithm. Thus, this shows that peptides detected in the transgenic model can also be identified in infected cells.

It would be important to validate the presentation of the peptides identified using models that better represent the SARS-Cov-2 infection and, ideally, in patients. However, the scope of our study was focused on the optimization of the MS method. Thus, validating the therapeutic or diagnostic value of the spike HLAIs would be the subject of a follow-up study. We recognize this as a limitation of our research

and have mentioned it in the manuscript. Nevertheless, we report only peptides that were predicted to bind the Raji or JY HLA alleles, and we have now validated the identification of all of them using synthetic peptides (see answer to Q35). Moreover, in previous reports, at least seven of these spike HLAps peptides showed positive experimental results in T-cell, B-cell, or MHC-ligand assays (see answers to Q15 and Q28, and manuscript section 2.7).

IV.6. 4) [spike immunopeptides reported in supplementary material 10]

Q30. It was kind of laborious work to connect the main text and corresponding data from main figures and supplementary materials because the necessary information was not organized well in one place, and thus my apologies in advance if I understood the data incorrectly (a), while I think there are typographical error or the mishandled data in supplementary material 10 regarding about the validation of peptide identification by USE (b).

(a) We apologize for the confusion that arose browsing the considerable amount of supplementary information and data we provide. Since this amount of information was difficult to organize, we used the S1 pdf file to describe all the supplementary files provided. To guide the reader to the respective sections of the document, we included a table of contents with links at the beginning of the pdf. Furthermore, in the new version, we added a description of the content at the start of the document. To decrease the number of additional documents, we also included the supplementary graphs of the GOrilla analyses and mirror spectra in the main supplementary pdf document (S1). Unfortunately, including the large result tables in this way is impossible. Thus, they are included as Excel or zipped CSV files, and their content is described and indicated in the main supplementary pdf document (S1).

(b) Indeed, there were some typographical errors in the mirror spectra from USE provided in v1 of the manuscript, and we thank the Reviewer for highlighting them. USE correctly displays experimentally derived spectral information of singly charged peptides. However, up to now (13.10.2023), the GUI will label the mirror spectra with "Charge: +2" and "Precursor m/z: (m/z:2)" even if "Precursor Charge" is set to "1". Accordingly, we needed to manually correct the faulty, USE-imprinted information on singly-charged peptides but probably missed some. We contacted the developers about this problem (20.09.2023). They will try to fix it, but they suggested changing it manually for a quick workaround. Therefore, we also needed to manually correct the precursor charge and m/z for the singly charged peptides in the new USE figures. We did not perform any other modification that could affect the results or their interpretation.

Q31. *In Figure 6d, there is one shared spike protein-carrying peptide (SIAYTEMSL, z = 1) both from JY and Raji. In here, Figure 6d describes this peptide is singly-charged one. But in the Supplementary material 10, there were 2 MS2 for SIAYTEMSL (z = 1) and SIAYTEMSL (z = 2). Please check this point and correct or explain the data accordingly.*

The Reviewer correctly signaled some typographical errors in the plots of the first manuscript version. We now exclusively included mirror spectra comparing the data acquired from endogenous and synthetic peptides, and corrected the errors. Please see the answer to Q30b for more details.

Q32. *Further, I'm skeptical about the MS2 spectra used in the bottom for these two (SIAYTEMSL of z = 1 and z = 2) that these are identical. The z = 1 and z = 2 spectra can be distinct from each other, while these two seems exactly the same. I think the processing by USE validation can include some kind of errors because not only for this, I found a few descriptions of "Procit CE: XX" on top spectra, that made me wonder the authors really used their raw file derived data for top spectra. Could you check and explain if I misunderstood this description? As such, the Figure 6d and the Supplementary material 10 are connected but really laborious to comprehend. It may be better to reorganize these materials for easy reading.*

Thank you for signaling the errors in the plots of the first manuscript version. We now exclusively included mirror spectra comparing endogenous and synthetic peptides, and corrected the errors. Please see the answer to Q29 and Q30b for more details. In addition, we removed the mirror spectra plots from the main manuscript to avoid repetition, since they are shown in Supplementary Figure S12.

IV.7. 5) [validation of the method and the z=1 peptides]

Q33. *One of the advantages of "Thunder-DDA-PASEF" by the authors in this manuscript, is efficient identification of singly-charged immunopeptides that have not been paid so much attention to date (a). Still, the verification for this critical main claim by the authors is, I'm afraid but I must say, insufficient (b).*

(a) One of the advantages of Thunder-DDA-PASEF is using the HLAip-tailored which improves the efficiency of cycle time usage by restricting the fragmentation events to possible peptides of interest and decreasing the time spent on other ions (see Section 2.2, Figure 2 and Section 2.3, Figure 3). This benefits the identification of both singly and multiply charged peptides (see Section 2.4, Figure 4). Thus, we have modified the text in the manuscript to show better the overall improvements provided by Thunder-DDA-PASEF while still mentioning the singly charged peptides since it is important to remind the importance of adapting MS methods for immunopeptidomics to include singly charged peptides.

(b) We supported the identification of singly charged peptides by cross-instrument evaluations. See Manuscript Section 2.5 and Section II.1 plus II 2 in this document. In addition, we have extended the validation by analyzing the synthetic versions of all the spike HLAips reported (see Manuscript Section 2.7 and Supplementary Figure S13).

Q34. *As examples were mentioned above, at least it seemed to me, the important MS2 spectra validation was not performed properly.*

We have extended the validation by analyzing the synthetic versions of all the spike HLAps reported. In addition, we have corrected the errors highlighted by the Reviewer.

Q35. *And more in detail, there were 9 Singly-charged immunopeptides derived from S1 and S2 model samples out of 18 validated peptides. While one (AIHVSGTNGTK) was obviously a false discovery because of the unmatched spectra against the predicted MS2 by USE. The authors mentioned that they included this peptide for validation because of the multiple identification from analyses while they didn't mention which of these peptides are the true discovery and false discovery. Applying the scoring system for validation, it is important to set the threshold like "more than score XX is valid and regarded as true discovery". Unfortunately, the authors didn't clarify this point, so it is very unclear whether the authors consider which of these peptides are the true discoveries or the false discoveries. The authors used USE that provides the score of SA and PCC, still it appears that the authors paid not so much consideration to SA. If not, isn't it better to use other calculation system, like dotp score Sykyline, or any other systems. I couldn't find what is the valid score for PCC by USE, but if you judge the identification only by PCC, still, you need to clarify what is the valid score for true identification and apply this score for further data validation. In main text, the authors described that the $PCC \geq 0.85$ as high-confidence identifications, while the Supplementary material 10 includes peptides that has PCC below 0.85 (ex., RQIAPGQTGK & SIIAYTMSL ($z = 2$)). As such, there is unmatched criteria between the main text and the data. Please set the threshold and clarify this point.*

We appreciate and support the suggestion of the Reviewer to use SA instead of PCC and setting a clear threshold for validation. Therefore, we have established a threshold of $SA \geq 0.8$ or detection in at least three injections. In addition, all the peptides showed an iRT ratio endogenous/synthetic above 0.99. This is also clearly mentioned now in the manuscript (L382-393). With the updated stringency filters, NSASFSTFK was removed from the selection of reported spike peptides (L393-395).

To maximize the identification confidence, we only report peptides detected in at least three injection replicates or with an $SA \geq 0.8$. Only three of the 17 peptides did not fulfill both criteria, but all the peptides reported showed an iRT ratio endogenous/synthetic ≥ 0.99 . Among these three peptides, GVLTESNKK^{s0550-0558} from Raji and RLQSLQTYV^{s1000-1008} from JY were identified in only one injection replicate; their SA were 0.92 and 0.85, respectively. Besides, although peptide TLKSFTVEK^{s0302-0310} showed a low SA (0.71), it was identified in three injection replicates. The low SA may be due to low-quality fragmentation spectra resulting from low abundant ions since it was near the lowest end of the dynamic range (Fig. 8c).

Q36. And the spectra comparison, if you simply inject more volume of synthetic peptides for Thunder-DDA-PASEF, can't you have better spectra for synthetic-APHGVVFL peptide that lacks b4 ion from the spectrum?

We thank the Reviewer for the valuable input on this specific peptide. To explore this strategy, we re-injected the first pool of synthetic peptides and the remaining newly synthesized ones. Indeed, injecting a higher concentration of the synthetic peptide “APHGVVFL” substantially increased spectral overlap in the mirrored spectrum. We used these results in the new version of the manuscript and supplementary materials.

Q37. Some of those spike protein-carrying peptides were found as multiple charge state, like $z = 1$ and $z = 2$. How were the results of USE for those different charge state from same peptide?

Indeed, some peptides were detected with multiple charges. For example, the peptide “GVYFASTEK” occurs in the sample and the synthetic peptides with precursor charges +1 and +2. Alignment of the “GVYFASTEK” fragmentation spectra resulted in SAs of 0.8 and 0.91 for charge state(s) +1 and +2, respectively. We now included spectra for peptides detected as multiply charged precursors in the supplementary information.

In addition, we would like to highlight that PEAKS and MS2Rescore only report the charge state with the highest confidence in the peptide level results. Thus, we obtained the information on multiple charge states from the PSM files. We aimed to evaluate if some peptides could have been identified as multiply charged when the fragmentation of singly charged ions is disabled or not possible (e.g., as in EThcD, see Section II.2). Our evaluation in section 2.5 (method comparison) indicates that most of the peptides identified as singly charged are only identified in this state. This further emphasizes the importance of including the fragmentation of singly charged ions in immunopeptidomics methods,

Q38. If the authors claim that the Thunder-DDA-PASEF has advantage in the identification of singly-charged immunopeptides, (as a matter of fact, the increased immunopeptidome by Thunder-DDA-PASEF seemed largely by increased singly-charged immunopeptide identification apparently, from main Figure 3a & 3b), the validation for this specie has to be done more thoroughly.

Thunder-DDA-PASEF provides a high coverage of the HLA_Ip immunopeptidome. Although this is partly due to the inclusion of singly charged peptides, the enhancement is also due to the high sensitivity of the timsTOF Pro2 instrument and the optimizations explained in Manuscript Section 2.3. This has been further evaluated by comparing it to other methods, as described in Section II.1 and II.2 of this document, and Section 2.5 of the Manuscript. In addition, we have strengthened the validation of singly charged peptides as detailed in Section II.2 of this document. Furthermore, we have clarified across the document that the main advantages of Thunder-DDA-PASEF reside in the overall optimization and not only in the detection of singly charged peptides.

Q39. *There are 5 peptides only found by z=1 (Figure 6d). While the validation score for these 5 peptides are not high enough (a) (if you set the threshold 0.91 like dotp score by Skyline system, only 1 candidate left since the data for SIIAYTMSL is currently compromised (b)). And I would say, the spectra NSASFSTFK (z=1), the intensity of b and y ion series are obviously different (c). Still the PCC score is ranging from 0.85~0.86. Thus the setting PCC=>0.85 as high-confidence identifications is not acceptable, at least to me. Since it seems that the bottom spectra used for these peptides were from predicted spectra, I think it is better to acquire the data by using cognate synthetic peptide to show convincing “true” identification. Please consider to add more validation analyses and reconstruct the Supplementary material 10 accordingly (d).*

(a) We agree with the concern of the Reviewer regarding using strict scores for peptide validation. Therefore, we have established a threshold as detailed in Q35. **(d)** In addition, we supported the identification of singly charged peptides by direct comparison to Orbitrap data and acquired data from synthetic peptides for all the spike HLAIs.

(b) The SA is obtained from all the fragment ions detected, while Skyline usually calculates the dotp based on a few selected ions. Thus, setting a threshold a threshold at 0.91 SA would be too stringent. We consider that a SA threshold of 0.8 is more adequate for TOF data.

Skyline¹² pre-refines spectra, ranks product ions of peptide targets, and, most importantly, filters the transition lists (Settings > Transition Settings > Filter & Library). Thus, depending on these user-defined settings, Skyline typically chooses the three (default) to five most abundant product y-ions of a target peptide to calculate the dotp score. As we also clarified now in the Methods section (L734-738), we neither pre-filtered fragment spectra to contain only most-abundant ions nor excluded b-ions before submitting the ion lists to USE. Therefore, the dotp score calculated by Skyline from only a few ions is not directly comparable with the SA or PCC obtained in USE from all the matched fragmentation ions.

To show the effect of different pre-filters on spectral similarity scores, we include in this document a figure (Fig. R1, next page) where we compared the USE-calculated SA results (figures below) obtained using all the fragments (top) compared to a refined spectral table with only the best matching y-ions (bottom) for the spike peptide with the lowest SA (“TLKSFTVEK”). Relaxing restrictions and keeping the five best fitting y-ions propels peptide “TLKSFTVEK” from an SA of 0.71 to 0.88, rendering it a true discovery based on our new stringent criteria and almost reaching the dopt proposed by the Reviewer. To avoid introducing such bias, **we did not filter the fragmentation ions** in the spectral tables used to generate the mirrored spectra plots reported in the manuscript.

(d) NSASFSTFK showed the label Prosit on both sides of the spectra; this was indeed an unintentional error in the first manuscript version. We thank the Reviewer for highlighting it. For the revised version of the manuscript we only compared against synthesized peptides. In addition, with the updated stringency filters (detected in at least three replicates, SA >= 0.8), NSASFSTFK is no longer in our selection of reported spike peptides.

TLKSFTVEK all transitions (incl. b- and y-ions)

TLKSFTVEK best 6 transitions (y-ions only)

Fig. R1. mirrored spectra for TLKSFTVEK comparing data from Raji transfectant cells (top) and synthetic peptides (bottom).

V. References

1. Teschner, D. *et al.* Ionmob: a Python package for prediction of peptide collisional cross-section values. *Bioinformatics* **39**, (2023).
2. Mommen, G. P. M. *et al.* Expanding the detectable HLA peptide repertoire using electron-transfer/higher-energy collision dissociation (ET_hcD). *Proc. Natl. Acad. Sci.* **111**, 4507–4512 (2014).
3. Declercq, A. *et al.* MS2Rescore: Data-Driven Rescoring Dramatically Boosts Immunopeptide Identification Rates. *Mol. Cell. Proteomics* **21**, 100266 (2022).
4. Gabriels, R., Martens, L. & Degroeve, S. Updated MS²PIP web server delivers fast and accurate MS² peak intensity prediction for multiple fragmentation methods, instruments and labeling techniques. *Nucleic Acids Res.* **47**, W295–W299 (2019).
5. Feola, S. *et al.* PeptiCHIP: A Microfluidic Platform for Tumor Antigen Landscape Identification. *ACS Nano* **15**, 15992–16010 (2021).
6. Weingarten-Gabbay, S. *et al.* Profiling SARS-CoV-2 HLA-I peptidome reveals T cell epitopes from out-of-frame ORFs. *Cell* **184**, 3962-3980.e17 (2021).
7. Nagler, A. *et al.* Identification of presented SARS-CoV-2 HLA class I and HLA class II peptides using HLA peptidomics. *Cell Rep.* **35**, 109305 (2021).
8. Xin, L. *et al.* A streamlined platform for analyzing tera-scale DDA and DIA mass spectrometry data enables highly sensitive immunopeptidomics. *Nat. Commun.* **13**, 3108 (2022).
9. Kaur, G. *et al.* Structural and regulatory diversity shape HLA-C protein expression levels. *Nat. Commun.* **8**, 15924 (2017).
10. Demmers, L. C., Wu, W. & Heck, A. J. R. HLA Class II Presentation Is Specifically Altered at Elevated Temperatures in the B-Lymphoblastic Cell Line JY. *Mol. Cell. Proteomics* **20**, 100089 (2021).
11. Weingarten-Gabbay, S. *et al.* Profiling SARS-CoV-2 HLA-I peptidome reveals T cell epitopes from out-of-frame ORFs. *Cell* **184**, 3962-3980.e17 (2021).
12. MacLean, B. *et al.* Skyline: an open source document editor for creating and analyzing targeted proteomics experiments. *Bioinformatics* **26**, 966–968 (2010).

Reviewers' Comments:

Reviewer #1:

Remarks to the Author:

The authors have made considerable effort to improve the manuscript. It now reads very well, shows logical progression/development. They have extensively road tested their new LC-MS method, placed each development in context and not overstated the results whilst appreciating the limitations. I am happy to recommend it for publication.

Please see some minor comments below. most pertain to the method section which I may have missed in the first read, it requires some attention to detail to make it more repeatable and consistent.

General comments

Ref is missing line 296

Figure 6 d many of the gained peptides are none-binders can you comment on this?

Your method section needs a little attention to detail please review it carefully, as a methods paper you need to allow people to repeat your work. Methods can you give the ordering details for the MWCO filter you have used.

e.g.

Where did your antibodies come from?

Line 566 H₂O or water or (v/v)?

Please try and be consistent, someone will hopefully try and utilise your methods in their own research.

Line 576 has question marks where I think there should be references

Line 463 ref missing

Reviewer #2:

Remarks to the Author:

Thank you again for giving me the opportunity to review the revised manuscript by David Gomez-Zepeda et al. I have reassessed the revised article.

My concerns have been adequately addressed, especially about the learning dataset for MS2Rescore as well as the validation efforts of the identified epitopes by using synthetic peptides. Formerly misplaced items have now been properly organized and the labeling of figures has been also corrected. And the authors have included additional relevant data in the revised supplementary materials. I believe this review process have improved the quality of revised manuscript.

Towards the end, if I may add a few small points that the authors may consider:

1: About the dataset for retraining MS2Rescore

I assume the main focus of MS2Rescore is to improve the identification of immunopeptides from timsTOF (CID) data and I appreciate the authors efforts of retraining the algorithm in the revised manuscript. I believe using the appropriate learning dataset is critical to build such these prediction algorithms. While in revised manuscript, the authors used trypsin digested samples for MS2Rescore retraining. From my understanding, the trypsin-digested peptides and the immunopeptides are different in many ways, e.g., the C-terminus trimming, the amino acid length and the preference of amino acid hydrophobicity. So, I'd like to know why the authors used trypsin-digested sample data for retraining. My apology in advance if I seem to have skipped over the reason mentioned somewhere in the revised manuscript. Authors emphasized the potency of MS2Rescore in immunopeptidomics, still, since the retraining dataset includes tryptic digests, can it be also anticipated the better protein identification under MS2Rescore? If this is the case, it seems too humble not to mention about this point. Have the authors ever checked the impact of MS2Rescore in protein identification by timsTOF? I will appreciate the clarification why they include the data of trypsin-digested peptides to retrain the MS2Rescore, and, the impact of MS2Rescorin in protein identification. The future readers of this manuscript will also expect the explanation about

these points.

2: About the SARS-COV2 in the article title

I understand the desire of authors to include the impactful word "SARS-COV2" in the article title, which has had a significant impact on our world over the past few years, but it is somewhat misleading in terms of content. Readers would expect more physiological findings in the SARS-COV2 infectious disease from the current title.

Since the B cells, the authors used in this manuscript for SARS-COV2 study, present Class I immunopeptides from cross-presentation. This means the intracellular processing of SARS-COV2 in B cells may include a bit different property of immunopeptides from that of lung epithelial cells, the original host cells. The authors explained the reason of unavailability of using actual SARS-COV2 samples for their analyses and I can understand that should be tolerated. Still, as the authors say by themselves, this manuscript focuses on MS2Rescore development, so from my personal view, I feel that the misleading word SARS-COV2 should be omitted from the title.

I believe the authors' expertise in development of prediction algorithms is crucial in the current MS situation and I wish the authors the best of success with continuing their relevant work.

Response to Reviewers (revision 2, *Nature Communications*)

Thunder-DDA-PASEF enables high-coverage immunopeptidomics and is further boosted by MS²Rescore with a novel MS²PIP timsTOF fragmentation prediction model

Table of Contents

I.	Introductory comments and repository access codes.....	1
II.	Responses to comments of Reviewer #1	3
III.	Responses to comments of Reviewer #2.....	6
IV.	Other improvements provided by the authors	9
V.	References	10

I. Introductory comments and repository access codes

I.1. General comment to Reviewers

Dear Reviewers,

We thank you again for the time and effort dedicated to evaluating our manuscript and for the constructive feedback. In addition, we sincerely appreciate the kind words about the interest and value of our work. We have included answers to your questions in this document and updated the documents in the submission file. We have applied the modifications suggested and extended our explanation on key points highlighted by your questions. This has resulted in significant improvements in the revised version of the manuscript and also the supplementary data. In this document, we answered your questions and we provide a description of the actions taken to respond to your comments. Please note that we split some of the comments into shorter questions, without modifying the text, to facilitate answering one by one. In addition, we included a marked-up file to facilitate identifying the modifications.

We believe that the manuscript is up to the standards of *Nature Communications*, and we hope you will agree with us and recommend our work for publication. In any case, we would be happy to address any further questions or suggestions.

I.2. Repository access credentials

Raw data and search results have been deposited to ProteomExchange and are accessible with the following Reviewer login credentials. In addition, we included a List of datasets and their repository locations in Supplementary Material S11.

Data from Tenzer lab:

Identifiers: PXD040385 for ProteomeXchange and JPST002044 for jPOST

URL: <https://repository.jpostdb.org/preview/209791232665415e51504c9>

Access key: 3041

The files of the JY immunopeptidomics data used to train MS²PIP were recently published¹ and are already publicly available with the dataset identifiers PXD043026 for ProteomeXchange and JPST002158 for jPOSTrepo.

HL60 immunopeptidomics data from Carapito lab:

URL to log in: <https://www.ebi.ac.uk/pride/login>

Project accession: PXD046535

Username: reviewer_pxd046535@ebi.ac.uk

Password: t0tCB5Cy

HeLa tryptic proteomics data from Carapito lab:

URL to log in: <https://www.ebi.ac.uk/pride/login>

Project accession: PXD046543

Username: reviewer_pxd046543@ebi.ac.uk

Password: KFdT3vsx

II. Responses to comments of Reviewer #1

II.1. Remarks to the Author:

Q1. The authors have made considerable effort to improve the manuscript. It now reads very well, shows logical progression/development. They have extensively road tested their new LC-MS method, placed each development in context and not overstated the results whilst appreciating the limitations. I am happy to recommend it for publication.

Please see some minor comments below. most pertain to the method section which I may have missed in the first read, it requires some attention to detail to make it more repeatable and consistent.

Dear Reviewer #1, we are pleased to know that you agree with us on the overall improvements on the method evaluation, the novel fragmentation model, and the whole manuscript after the first round of revisions. We thank you again for all your previous and new suggestions which were highly valuable for us. We have implemented the minor corrections as detailed below.

II.2. General comments

Q2. Ref is missing line 296

Thank you for highlight the missing reference. We have added the corresponding reference:

R. Bouwmeester, R. Gabriels, N. Hulstaert, L. Martens, S. Degroeve, DeepLC can predict retention times for peptides that carry as-yet unseen modifications, Nat. Methods. 18 (2021) 1363–1369. <https://doi.org/10.1038/s41592-021-01301-5>.

Q3. Figure 6 d many of the gained peptides are none-binders can you comment on this?

Indeed, a large proportion of the gained peptides (41.4%) are predicted to be non-binders (Fig. 6d). In addition, 59.1% of the lost peptides were predicted binders. Since these proportions approach 50%, they indicate that MS²Rescore is not introducing a bias towards or against certain types of peptides. Since rescoring was more beneficial for peptides in the low range of the dynamic range (Fig. 6f), we believe the gained non-binders may originate from low abundant co-enrichments and eventual protein degradation, which are detected in higher numbers in the highest sample inputs.

Q4. Your method section needs a little attention to detail please review it carefully, as a methods paper you need to allow people to repeat your work.

a. *Methods can you give the ordering details for the MWCO filter you have used. e.g.*

We added the information about the MWCO filters and extended the text about the HLB desalting plates:

[...] Next, peptides were ultrafiltered using 10 kDa molecular weight cutoff (MWCO) filters (Vivacon 500, 10,000 MWCO Hydrosart) and then desalted by SPE on a Hydrophilic-Lipophilic-Balanced sorbent (Oasis HLB 96-well μ Elution Plate, 2 mg Sorbent per Well, 30 μ m, Waters Corp.). [...]

b. *Where did your antibodies come from?*

We added this information to the manuscript.

Tenzer Lab (L564-565).

[...] The anti-panHLA Class I antibody W6/32 (anti-HLA-A, -B, -C) was purchased from Hoelzel-biotech, and produced by Leinco Technologies (ref. H263). [...]

Carapito Lab (L572-575).

The HB-95 hybridoma producing anti-panHLA Class I antibody W6/32 was purchased from ATCC and cultured in Panserin 401 serum free medium (Pan Biotech). The purification of the antibody was done with the NGC Chromatography System (Biorad) using a HiTrap Protein G HP 1 mL column (Amersham Pharmacia).

c. *Line 566 H₂O or water or (v/v)?*

We corrected H₂O to water.

d. *Please try and be consistent, someone will hopefully try and utilise your methods in their own research.*

We agree with the Reviewer on the importance of providing all the information possible to facilitate the replication of our methods by other researchers. Indeed, that was one of our objectives when submitting this manuscript for publication. Thus, we had already included detailed descriptions of all the parameters used for data acquisition and analysis in the Supplementary Material (S2). However, we inadvertently missed some details about the sample preparation methodology.

To facilitate the replication of our methodology, we made the following modifications:

- We added the subsection “Materials and substances” at the beginning of the Methods sections (L515 – 518), describing the origin of the main non-biological Materials and substances
- We homogenized the description of percentage and ratios of mixtures by adding (v/v) or (m/v) after the substance name
- We homogenized the mentions of mixtures with added FA or TFA; e.g., “water with 0.1\% formic acid (FA (v/v))”.

e. Line 576 has question marks where I think there should be references

Thank you for highlight the missing references. We have added them:

J.R. Wisniewski, A. Zougman, N. Nagaraj, M. Mann, Universal sample preparation method for proteome analysis, *Nat Methods*. 6 (2009) 359–362. <https://doi.org/10.1038/nmeth.1322>

M. Sielaff, J. Kuharev, T. Bohn, J. Hahlbrock, T. Bopp, S. Tenzer, U. Distler, Evaluation of FASP, SP3, and iST Protocols for Proteomic Sample Preparation in the Low Microgram Range, *J. Proteome Res.* 16 (2017) 4060–4072. <https://doi.org/10.1021/acs.jproteome.7b00433>.

f. Line 463 ref missing

Thank you for highlight the missing reference. We have added it:

K. Ogata, C.-H. Chang, Y. Ishihama, Effect of Phosphorylation on the Collision Cross Sections of Peptide Ions in Ion Mobility Spectrometry, *Mass Spectrom.* 10 (2021) A0093–A0093. <https://doi.org/10.5702/massspectrometry.A0093>.

III. Responses to comments of Reviewer #2

III.1. Remarks to the Author

Q5. *Thank you again for giving me the opportunity to review the revised manuscript by David Gomez-Zepeda et al. I have reassessed the revised article.*

My concerns have been adequately addressed, especially about the learning dataset for MS2Rescore as well as the validation efforts of the identified epitopes by using synthetic peptides. Formerly misplaced items have now been properly organized and the labeling of figures has been also corrected. And the authors have included additional relevant data in the revised supplementary materials. I believe this review process have improved the quality of revised manuscript.

Towards the end, if I may add a few small points that the authors may consider: [Continues in next section]

Dear Reviewer #2, we are pleased to know that you agree with us on the overall improvements on the method evaluation, the novel fragmentation model, the spike peptide validation, and the whole manuscript after the first round of revisions. We thank you again for all your previous and new suggestions which were highly valuable for us. We have implemented the minor corrections as detailed below.

III.2. 1: About the dataset for retraining MS2Rescore

Q6. *I assume the main focus of MS2Rescore is to improve the identification of immunopeptides from timsTOF (CID) data and I appreciate the authors efforts of retraining the algorithm in the revised manuscript. I believe using the appropriate learning dataset is critical to build such these prediction algorithms. While in revised manuscript, the authors used trypsin digested samples for MS2Rescore retraining. From my understanding, the trypsin-digested peptides and the immunopeptides are different in many ways, e.g., the C-terminus trimming, the amino acid length and the preference of amino acid hydrophobicity. So, I'd like to know why the authors used trypsin-digested sample data for retraining. My apology in advance if I seem to have skipped over the reason mentioned somewhere in the revised manuscript (a). Authors emphasized the potency of MS2Rescore in immunopeptidomics, still, since the retraining dataset includes tryptic digests, can it be also anticipated the better protein identification under MS2Rescore? If this is the case, it seems too humble not to mention about this point. Have the authors ever checked the impact of MS2Rescore in protein identification by timsTOF? I will appreciate the clarification why they include the data of trypsin-digested peptides to retrain the MS2Rescore, and, the impact of MS2Rescorin in protein identification (b). The future readers of this manuscript will also expect the explanation about these points.*

(a) We agree on the importance of using the appropriate dataset to train prediction models. Therefore, we used a dataset including both tryptic and non-tryptic peptides, as already described in the previous version of the manuscript, and copied below. However, the comment of the Reviewer helped us realize that we had not mentioned the rationale behind including both types of peptides in the same model, contrary to how it was done in previous MS²PIP models. Therefore, we have adapted the results text accordingly (highlighted in red in the text below).

Results (293 – 299):

*Several post-processing tools have shown improvements in immunopeptide identification by rescoring peptide spectrum matches (PSMs) based on characteristics disregarded in the initial search²⁻⁵. To improve the peptide identification, we implemented MS²Rescore (MS²R, v3.0.0b4)^{2,6} into our workflow. **Since peak intensity predictions rely heavily on the fragmentation method, instrument or labeling method^{7,8}, we trained a new timsTOF-specific peak intensity model using in-house timsTOF data. In addition, it has been shown that prediction models trained on datasets including both, tryptic and non-tryptic peptides, can perform equally well for predicting the fragmentation of both types of peptides^{2,3}. Therefore, we trained a novel MS²PIP⁸ timsTOF fragmentation prediction model using data acquired at two different labs from immunopeptidomics (HLA class I), tryptic peptides, and elastase digest samples. In total, 241,104 peptides (including modifications) were used to train the model and a distinct set of 10,045 peptides to test it. Retention times were predicted with DeepLC⁹. In addition, to take advantage of the ion mobility separation, we added the peptide CCS predicted using the ionmob GRU predictor as a rescoring feature, as recently described¹.***

Methods (L718 – 721):

[..] Therefore, we trained a new timsTOF-specific peak intensity model using in-house timsTOF data from two different labs. This included digested peptides of JY (trypsin and elastase) and HeLa (trypsin), and HLA class I immunoprecipitation-enriched peptides of JY, HeLa, SK-MEL-37 (all previous from Tenzer lab), and HeLa (trypsin) and HL60 samples using multiple CE settings (Carapito lab).

(b) Thank you for this valuable suggestion. We added a plot showing the effect of MS²Rescore on the number of proteins covered by HLA Ips (Fig. 6i) and commented on it in the Results. Indeed, there is also a significant increase in the protein coverage. We also expect similar improvements for tryptic peptides in proteomics experiments since the fragmentation prediction was similarly accurate in the test dataset. However, we did not further evaluate this in this manuscript since it is focused on immunopeptidomics.

Results (L336 – 339):

[...] Rescoring also significantly increased the number of protein groups covered by HLAIs (Fig. 6i). Using MS²Rescore resulted in 3,225 ± 191 HLAIs from 1 Mce and 5,765 ± 48 from 20 Mce. Thus, rescoring also boosted the immunopeptidome protein coverage, providing HLAIs for proteins that would have been missed otherwise. [...]

Discussions (L431 – 433):

[...] Since fragmentation prediction was similarly accurate for non-tryptic and tryptic peptides, we expect similar improvements for proteomics experiments, but this was out of the scope of this project. [...]

III.3. 2: About the SARS-COV2 in the article title

Q7. I understand the desire of authors to include the impactful word "SARS-COV2" in the article title, which has had a significant impact on our world over the past few years, but it is somewhat misleading in terms of content. Readers would expect more physiological findings in the SARS-COV2 infectious disease from the current title. Since the B cells, the authors used in this manuscript for SARS-COV2 study, present Class I immunopeptides from cross-presentation. This means the intracellular processing of SARS-COV2 in B cells may include a bit different property of immunopeptides from that of lung epithelial cells, the original host cells. The authors explained the reason of unavailability of using actual SARS-COV2 samples for their analyses and I can understand that should be tolerated. Still, as the authors say by themselves, this manuscript focuses on MS²Rescore development, so from my personal view, I feel that the misleading word SARS-COV2 should be omitted from the title. I believe the authors' expertise in development of prediction algorithms is crucial in the current MS situation and I wish the authors the best of success with continuing their relevant work.

We agree with the suggestion of the Reviewer to change the title of the manuscript. Here, we focused on showing (1st) the development of Thunder-DDA-PASEF and (2nd) the update of MS²Rescore with a novel timsTOF MS²PIP fragmentation prediction model. Although the SARS-CoV-2 spike immunopeptidome was used as an example, we indeed did not include further physiological analyses. Thus, to better represent the content of the manuscript, we changed the title to:

Thunder-DDA-PASEF enables high-coverage immunopeptidomics and is further boosted by MS²Rescore with a novel MS²PIP timsTOF fragmentation prediction model

IV. Other improvements provided by the authors

Q8. *Line 735 reference was missing.*

We have added the corresponding reference:

O. Wagih, ggseqlogo: a versatile R package for drawing sequence logos, *Bioinformatics*. 33 (2017) 3645–3647. <https://doi.org/10.1093/bioinformatics/btx469>.

Q9. *MS2Rescore 3.0 citation*

Since the previous submission, a preprint describing MS2Rescore 3.0 has been posted to ChemRxiv. Thus, we added the corresponding reference in Lines 82, 98, 291, 762, and the footnote of Figure 1:

L.M. Buur, A. Declercq, M. Strobl, R. Bowmeester, S. Degroeve, L. Martens, V. Dorfer, R. Gabriels, MS2Rescore 3.0 is a modular, flexible, and user-friendly platform to boost peptide identifications, as showcased with MS Amanda 3.0, Preprint Available ChemRxiv. (2023) 1–23. <https://doi.org/10.26434/chemrxiv-2023-rvr9n>.

Q10. *In the label of figure 1d, the mention of MS²Rescore rescoring was missing*

We have modified the label as shown below (highlighted in red).

(d) Data analysis: Database search was performed in PEAKS XPro using unspecific cleavage. *After training a novel MS²PIP⁸ timsTOF fragmentation prediction model, peptide identification was rescored using MS²Rescore (MS²R, v3.0.0b4)^{2,6}. Data analysis was performed in R and predicted MHC-binding affinity was evaluated using NetMHCpan-4.1¹⁰ and GibbsCluster-2.0¹¹ through MhcVizPipe (v0.7.9)¹².*

Q11. *Other minor corrections*

Removed double spaces.

Q12. *The methods did not mention how files were organized into projects.*

We added the following text to the section Methods / Peptidomics database search and rescoring (L690 – 692).

To avoid introducing a bias due to the transfer of peptide identifications, files were processed in PEAKS projects and MS2Rescore processes containing only data from the same cell line (or plasma), LC-MS method, or cell equivalents injected.

V. References

1. Teschner, D. *et al.* Ionmob: a Python package for prediction of peptide collisional cross-section values. *Bioinformatics* **39**, (2023).
2. Declercq, A., Bouwmeester, R., Degroeve, S., Martens, L. & Gabriels, R. MS²Rescore: Data-driven rescoring dramatically boosts immunopeptide identification rates. *bioRxiv* 2021.11.02.466886 (2021).
3. Wilhelm, M. *et al.* Deep learning boosts sensitivity of mass spectrometry-based immunopeptidomics. *Nat. Commun.* **12**, 3346 (2021).
4. Li, K., Jain, A., Malovannaya, A., Wen, B. & Zhang, B. DeepRescore: Leveraging Deep Learning to Improve Peptide Identification in Immunopeptidomics. *Proteomics* **20**, (2020).
5. Xin, L. *et al.* A streamlined platform for analyzing tera-scale DDA and DIA mass spectrometry data enables highly sensitive immunopeptidomics. *Nat. Commun.* **13**, 3108 (2022).
6. Buur, L. M. *et al.* MS2Rescore 3.0 is a modular, flexible, and user-friendly platform to boost peptide identifications, as showcased with MS Amanda 3.0. *Prepr. available ChemRxiv* 1–23 (2023) doi:10.26434/chemrxiv-2023-rvr9n.
7. Gabriels, R., Martens, L. & Degroeve, S. Updated MS²PIP web server delivers fast and accurate MS² peak intensity prediction for multiple fragmentation methods, instruments and labeling techniques. *Nucleic Acids Res.* **47**, W295–W299 (2019).
8. Declercq, A. *et al.* Updated MS²PIP web server supports cutting-edge proteomics applications. *Nucleic Acids Res.* **51**, W338–W342 (2023).
9. Bouwmeester, R., Gabriels, R., Hulstaert, N., Martens, L. & Degroeve, S. DeepLC can predict retention times for peptides that carry as-yet unseen modifications. *Nat. Methods* **18**, 1363–1369 (2021).
10. Reynisson, B., Alvarez, B., Paul, S., Peters, B. & Nielsen, M. NetMHCpan-4.1 and NetMHCIIpan-4.0: Improved predictions of MHC antigen presentation by concurrent motif deconvolution and integration of MS MHC eluted ligand data. *Nucleic Acids Res.* **48**, W449–W454 (2021).
11. Andreatta, M., Alvarez, B. & Nielsen, M. GibbsCluster: Unsupervised clustering and alignment of peptide sequences. *Nucleic Acids Res.* **45**, W458–W463 (2017).
12. Kovalchik, K. A. *et al.* MhcVizPipe: A Quality Control Software for Rapid Assessment of Small- To Large-Scale Immunopeptidome Datasets. *Mol. Cell. Proteomics* **21**, 0–14 (2022).